# Effects of the impact angle on the coefficient of restitution in rockfall analysis based on a medium-scale laboratory test

Yanhai Wang[1], Wei Jiang[1, 2], Shengguo Cheng[2], Pengcheng Song[2], Cong Mao[2]

[1] Hubei Key Laboratory of Disaster Prevention and Mitigation (China Three Gorges University), Yichang, Hubei, 443002, People's Republic of China

[2] Department of Civil, Structural, and Environmental Engineering, University at Buffalo, Buffalo, NY, 14260, United States

*Correspondence to*: Wei Jiang (jiangweilion@163.com)

**Abstract.** The reliability of a computer programme simulating rockfall trajectory depends on the ascertainment of reasonable values for the coefficients of restitution, which typically vary with the kinematic parameters and terrain conditions. The effects of the impact angle with respect to the slope on the coefficients of restitution have been identified and studied using small-scale laboratory tests. To investigate whether the existing conclusion based on small-scale laboratory tests is valid when the test scale changes and the role of rotation in the effect of the impact angle on the coefficients of restitution, this study performed a medium-scale laboratory test using spherical limestone polyhedrons impacting concrete slabs. Free fall tests are conducted, and the velocities before and after the impact are obtained by a 3D motion capture system. The results comparison between our test and the existing small-scale tests verified that several general laws occur when accounting for the effect of the impact angle, regardless of the test scales and conditions. Increasing the impact angle will induce reductions in the normal coefficient of restitution $R_n$, the kinematic coefficient of restitution $R_v$ and the kinetic energy coefficient of restitution $R_E$, whereas it will lead to increases in the tangential coefficient of restitution $R_t$. The rotation plays an important role in the effect of the impact angle. A higher percentage of kinetic energy converted to rotational energy always induces a higher normal coefficient of restitution $R_n$ and a lower tangential coefficient of restitution $R_t$. As the impact angle decreases, the ratio between the rebound angle $\beta$ and the impact angle $\alpha$ increases, and the percentage of kinetic energy dissipated in rotation as the collision became higher. Considering that the effect of block shape and the detailed impact orientations are not involved in the present study, the test results are valid for trajectory simulation codes based on a lumped-mass model and can be referenced in the trajectory predication of spherical rocks impacting hard surfaces using a rigid body model.

## 1 Introduction

In mountain areas, rockfall is a frequent natural disaster that endangers human lives and infrastructure. Numerous examples of fatalities or infrastructure damage due to rockfall have been reported (Guzzetti, 2003; Pappalardo, 2014). Various protective measures, such as barrier fences, cable nets and rockfall shelters, have been widely used to reduce rockfall hazards. To ensure the efficiency of mitigation techniques, the motion trajectory of the rockfall must be estimated. The trajectory can

provide important information, such as the travel distances of possible rockfall events, the bouncing height and kinetic energy level of the rockfall at various positions along the slope.

Numerous algorithms have been developed to solve this problem, and the progress up to the end of the last century has been summarized by Dorren (2003) and Heidenreich (2004). Due to these efforts, computer simulation codes, such as RockFall (Stevens, 1998), CRSP (Jones et.al, 2000) and Stone (Guzzetti et.al, 2002), RAMMS::Rockfall (Christen et al, 2007), Rockyfor3D (Dorren, 2010) and Pierre (Valentin et al., 2015; Andrew and Oldrich, 2017), are developed to acquire motion information regarding rockfall. A main feature that allows one to distinguish between different rockfall trajectory codes is the representation of the objective rock. The first approach, a lumped-mass model, treats the rock as a single and dimensionless point and assigns all of the properties of the rock to that point. The second model is a rigid body model, which considers the rock as a body with its own shape and volume and accounts for all types of block movement, including rotation. Finally, a hybrid model adopts a lumped mass model to calculate the free fall of the rock and simulates other types of block motion using a rigid body model. In most codes, the trajectory of falling rocks was described as combinations of four types of motion: free fall, rolling, sliding and rebound. The rebound motion, a succession of rockfalls impacting the slope surface, is the least understood and the most difficult to predict of the four types of motion (Volkwein A et.al, 2011), which is controlled by the coefficients of restitution in computer simulation. Thus, the reliability of the estimation of the coefficient of restitution must be ensured.

## 1.1 Definition for the coefficient of restitution

The coefficient of restitution is a dimensionless value representing the ratio of velocities or energies of a boulder before and after it impacts the slope. Various definitions for the coefficient of restitution have been proposed in previous studies, but no consensus was reached on which definition is more appropriate for rockfall prediction. As shown in Fig. 1, when one boulder impacts the slope surface, the impact velocity $v_i$ can be resolved into a normal component $v_{ni}$ and a tangential component $v_{ti}$ according to the slope angle $\theta$. Then, the boulder leaves the surface with a rebound velocity $v_r$, which similarly has a $v_{nr}$ and a $v_{tr}$. The angular velocities of the boulder before and after impact are denoted as $\omega_i$ and $\omega_r$, respectively. The impact angle $\alpha$ and rebound angle $\beta$ are drawn in Fig. 1.

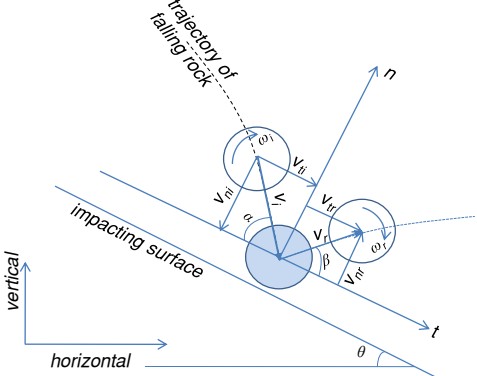

**Figure 1.** Related quantities adopted in the definitions for the coefficient of restitution

The normal and tangential coefficients of restitution are the most commonly used definitions, and the two coefficients of restitution are typically denoted as $R_n$ and $R_t$, respectively. The mathematical expressions of $R_n$ and $R_t$ are

$$R_n = v_{nr} / v_{ni}, R_t = v_{tr} / v_{ti} \tag{1}$$

Another common definition is the kinematic coefficient of restitution, $R_v$, representing the ratio between the magnitudes of the rebound and impact velocities:

$$R_v = v_r / v_i \tag{2}$$

This definition, originating from Newton's theory of particle collision, has been used by Habib (1976), Paronuzzi (1989) and other scholars. When $R_v$ is used in the trajectory predication, an assumption regarding the rebound direction is necessary to fully determine the velocity vector after impact.

In addition, the ratio of kinetic energies before and after impact is used to define the kinetic energy coefficient of restitution $R_E$, which is written as

$$R_E = E_r / E_i = (E_{rr} + E_{rt}) / (E_{ir} + E_{it}) \tag{3}$$

in which $E_i$ and $E_r$ are the kinetic energy before and after the impact, respectively. $E_{ir}$ and $E_{rr}$ are the rotational energy before and after the impact; $E_{it}$ and $E_{rt}$ denote the translational energy before and after the impact. $E_{ir}$, $E_{it}$, $E_{rr}$ and $E_{rt}$ are computed as

$$E_{ir} = 0.5 I \omega_i^2, E_{it} = 0.5 m v_i^2, E_{rr} = 0.5 I \omega_r^2, E_{rt} = 0.5 m v_r^2 \tag{4}$$

Here, $m$ is the mass, $I$ is the moment of inertia. $R_E$ can reflect the kinetic energy loss caused by the impact and has been used by Bozzolo and Pamini (1986), Azzoni et al. (1995) and Chau et al. (2002).

In these definitions, $R_n$ and $R_t$ become more popular in engineering practice for simplicity in computer simulation software. $R_n$ and $R_t$ are used conjointly and characterize the variation in the tangential and normal components of the boulder velocity, respectively. Given an impact velocity, the rebound velocity and direction can be completely determined using this definition without any further assumption. Therefore, $R_n$ and $R_t$ attracted the most attention in previous studies, and some typical values of $R_n$ and $R_t$ have been summarized (Agliardi and Crosta, 2003; Heidenreich, 2004; Scioldo, 2006).

**1.2 Previous studies on the effects of the impact angle on the coefficient of restitution**

Various techniques, such as laboratory tests (Buzzi et.al, 2012; Asteriou et.al, 2012), field tests (Dorren et.al, 2006; Spadari et al. 2012), back analysis of field evidence (Paronuzzi, 2009) and theoretical estimation (He et.al, 2008), have been used to determine the coefficient of restitution. Variations in the impact conditions, e.g., the material properties of both the rocks and slopes (Wu, 1985; Fornaro et.al, 1990; Robotham et.al, 1995; Richards et.al, 2001; Chau et.al, 2002; Asteriou et.al, 2012), the shape of the rocks (Chau et.al, 1999; Buzzi et.al, 2012), the roughness of the slope surface (Giani et.al, 2004) and the impact angle, influence the coefficient of restitution considerably. However, in those existing summaries for typical values, the coefficients of restitution were determined mainly accounting for the terrain conditions.

The impact angle, the angle between the directions of the impact velocity and the slope segment, is a kinematic parameter of the falling rock, indicating only that the terrain conditions involved in estimating the value of the coefficient of restitution

may be unreliable. Since Broili (1973) first identified this problem, numerous experiments have been performed to acquire a comprehensive picture of the effects of the impact angle. In situ tests are expensive and not suitable for statistical and parameter analysis; thus, existing studies have largely been performed in the laboratory. In some studies, the impact angle was referred to as the slope angle $\theta$ (or the impact surface angle) in free-fall tests. However, the impact surface angle is only another expression because the slope angle $\theta$ and impact angle $\alpha$ sum up to 90 ° under these conditions.

Wu (1985) conducted laboratory tests using rock blocks on a wooden platform and rock slope and suggested that there is a linear correlation between the impact surface angle and the mean value of the restitution coefficient. He proposed that increasing the angle of the impact surface causes the normal coefficient $R_n$ to increase regardless of the block mass and causes the tangential coefficient $R_t$ to decrease slightly.

Richards et al. (2001) executed free-falling tests considering different types of rock and slope conditions and established a correlation between the coefficient of restitution and the Schmidt hammer rebound hardness. The impact surface angle was added to the correlation to reflect its linear improvement effect on the normal coefficient $R_n$.

Chau et al. (2002) conducted experiments using spherical boulders and a rock slope platform, both made of dental plaster. The free-falling tests indicated that the normal coefficient increases with increases in the impact surface angle, whereas there was no clear correlation with the tangential coefficient.

Cagnoli and Manga (2003) studied oblique collisions of lapilli-size pumice cylinders on flat pumice targets and determined that the impact angle can influence the rebound angle, the kinetic energy loss and the ratios of the velocity components. The normal coefficient decreases as the impact angle approaches 90 °.

Asteriou et al. (2012) performed laboratory tests using five types of rocks from Greece. The results of the parabolic drop tests indicated that the kinematic coefficient of restitution $R_v$ was more appropriate than the normal coefficient of restitution for use in correlations with the impact angles. Then, the normal coefficient of restitution could be estimated accounting for the rebound–impact angle ratio.

Buzzi et al. (2012) conducted experiments using flat concrete blocks in four different forms and determined that a combination of low impact angle, rotational energy and block angularity may result in a normal coefficient of restitution in excess of unity.

James (2015) evaluated restitution coefficients using milled aluminium blocks and a planar wooden slope. Three different shapes of blocks were custom made, and the slope surface was carpeted. Both the first impact under free fall conditions and the series impacts during runout were recorded. It was observed that $R_n$ shows a positive correlation with increasing slope angle, while $R_t$ shows a negative correlation.

These efforts have highlighted the importance of the impact angle with regard to the coefficient of restitution. Most of the existing tests indicated that increasing the impact angle induces a reduction in the normal coefficient of restitution $R_n$ but an improvement in the tangential coefficient of restitution $R_t$. However, there are two issues that remain unsolved. First, whether the laws regarding the effect of the impact angle on the coefficient of restitution are influenced by the test scale is uncertain. Until now, the existing laboratory tests commonly captured the trajectory of small samples using a high-speed

video camera, which means that the existing results are based on small-scale laboratory tests. As Heidenreich (2004) noted, the mature similarity theory regarding the model test on the coefficient of restitution is still absent because the influence factors are much more than the material properties and sizes. Therefore, laboratory tests with larger scales should be performed to confirm the validity of the existing conclusions, which is beneficial for further interpreting the results of small-

scale laboratory tests. Second, in free fall tests, the rotation after impact plays an important role in kinetic energy dissipation of the falling block (Chau et al., 2002), and it was supposed to affect the variation of $R_n$ and $R_t$ (Broili, 1973; Cagnoli and Manga, 2003). However, few studies have been performed to reveal the effect of rotational motion on the coefficient of restitution through quantitative analysis. Whether a correlation occurs between the rotation and the effect of the impact angle deserves our attention and may offer some insights into the effect of the impact angle on the coefficients of restitution.

Hence, this study employs a 3D motion capture system and a special releasing device to perform a medium-scale laboratory experiment. Spherical polyhedrons made of limestone were selected as samples with a maximum diameter of 20 cm. The landing plate consisted of C25 concrete slabs. To address the effect of the impact angle, different inclined plate angles and releasing heights were used in free fall tests. The resulting coefficients of restitution, $R_n$, $R_t$, $R_v$ and $R_E$, were calculated, and their trends in terms of the impact angle were explored to provide a complete picture. Then, the results are compared with

three existing small-scale experiments to determine whether the test scale affects the law whereby the impact angle influences the coefficients of restitution. The percentage of the total kinetic energy before impact converted to rotational energy was investigated, and the role of rotation in the effect of the impact angle on the coefficient of restitution was analysed. For only spherical polyhedrons that are taken as the samples in this study, the test results may have some limitations.

**2 Laboratory investigation**

**2.1 Rock specimens and concrete slabs**

All falling rock specimens in this study were natural limestone from the China Three Gouges area and were customized in accordance with the required sizes. As shown in Fig. 2a, irregular artificial cutting facets constituted the surface of the specimens, and the edges were not smoothed; thus, the shape is called a spherical polyhedron in this study to distinguish it

from the standard sphere used in other research studies. To appraise the effect of rock size on the rebound characteristics, two different diameters were considered (D=10 cm and D=20 cm), with corresponding average masses of 1.2 kg and 10 kg. The C25 concrete slabs came from a prefabricated concrete factory. As shown in Fig. 2b, each concrete slab had dimensions of 120 cm×50 cm×15 cm. The mechanical properties of the materials adopted in the test are determined beforehand. The limestone has Young's modulus $E$=41 GPa, Poisson's ratio $v$=0.21 and Schmidt Hardness R=36.0. The concrete has $E$=28

GPa, $v$=0.20 and R=32.5.

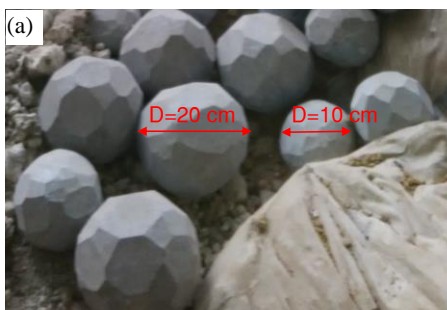
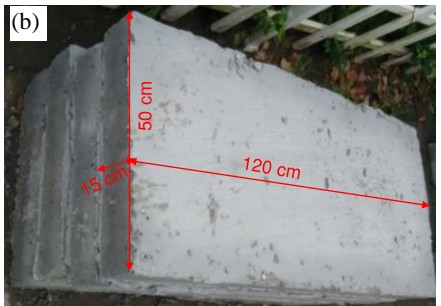

**Figure 2.** Materials used in this experiment: **(a)** Spherical limestone polyhedron with two diameters. **(b)** C25 prefabricated concrete slabs.

## 2.2 Testing apparatus

The apparatus used in this study consisted of a ramp, landing plate and release device (Fig. 3a). The ramp was built by compacting gravelly soil and had an inclined surface with planned angles produced by artificial excavation. Then, two concrete slabs were placed on the inclined surface to form the landing plate. One device was designed and manufactured specifically to catch and release specimens of various sizes. As shown in Fig. 3b, the device had four adjustable tongs at the bottom, which could grasp spherical blocks with diameters from 8 cm to 25 cm. A wireless receiver and electromagnetic relay were installed in the upper portion of the device, offering a wireless method of altering the tong status, grasping or loosing. The device could be connected to an indoor mobile crane using the top ring, which means that the device could go up and down by managing the crane.

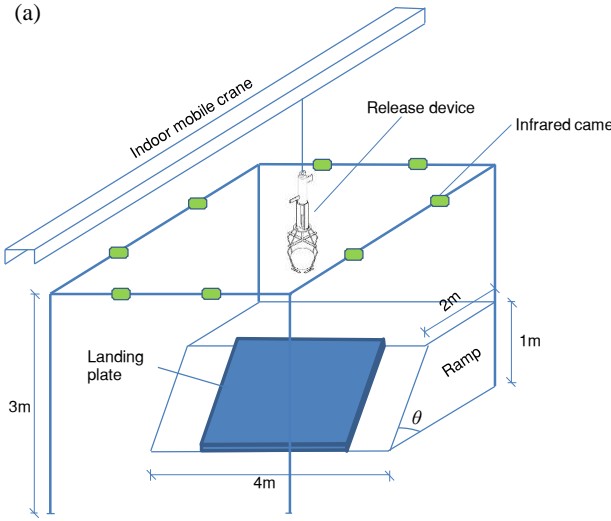
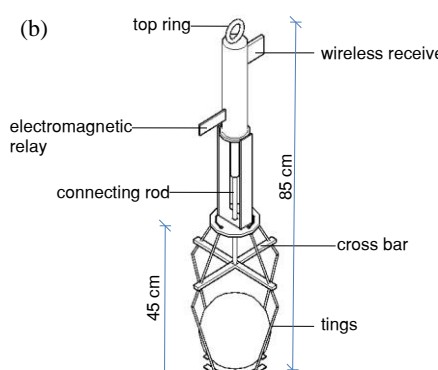

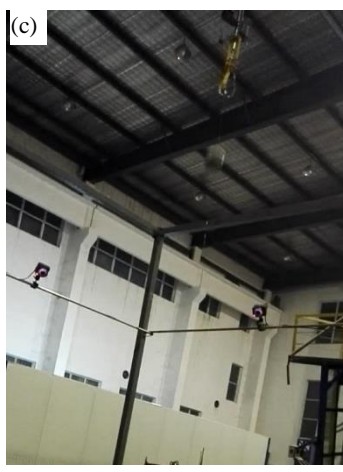
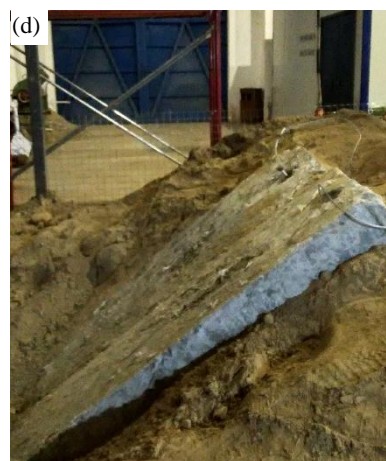

**Figure 3.** The testing apparatus in this study. **(a)** A general view of the apparatus. **(b)** A schematic representation of the specifically designed release device. **(c)** A falling sample after the wireless switch was turned on. **(d)** Slabs to be replaced for excessive damage.

A free-fall test was performed in the experiment, and the complete process of one test is as follows. First, when one spherical polyhedron is prepared to be tested, the tongs are adjusted to accommodate the polyhedron by moving the cross bar up and down. After the sample is in the tongs, the grasping state is selected. Then, the device hanging on the indoor crane is moved to the position above the landing plate and lifted to the planned height. Next, by operating the wireless switch, the tongs are loosened, and the sample begins to fall (Fig. 3c). Finally, the sample impacts the landing plate, and its motion is recorded. The surface of the concrete slabs becomes worn with successive impacts. Once the damage to the surface is excessive, as shown in Fig. 3d, the slabs used are replaced with new slabs.

### 2.3 Data acquisition

The spatial motion information of falling samples was obtained by the Doreal DIMS-9100(8c) Motion Capture System. This system has eight near-infrared cameras (see Fig. 4a) with an operating speed of 60 fps and can capture the spatial trails of markers attached to the surface of the sample, as shown in Fig. 4b. Then, the motion analysis programme provides the spatial motion information of the sample, e.g., its position and velocity. Finally, the coefficient of restitution can be calculated according to Eq. (1-4) for subsequent analysis.

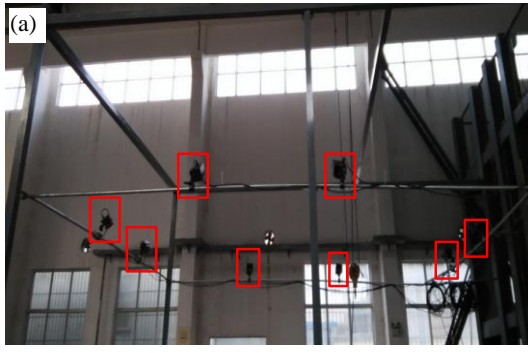
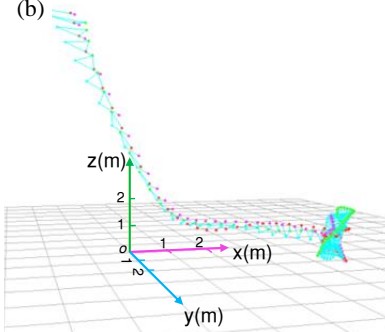

**Figure 4.** The data acquisition system: **(a)** 8 infrared cameras used to capture the trails, and **(b)** a motion analysis programme used to provide the spatial motion information.

## 2.4 Experimental program

Four different inclined angles $\theta$ of the landing plate (30°, 45°, 60° and 75°) were considered in this study to determine the effect of the impact angle on the coefficients of restitution. The impact angles are approximately related to the incline angle $\theta$ of the landing plate under free-fall test conditions. Limestone specimens were released at three different heights of 2.5 m, 3.5 m and 4.5 m upon the inclined concrete slabs. However, two tests do not necessarily have identical release conditions even if they have the same release height and use the same specimen because the positions on which the tongs catch the specimen may differ slightly in any two tests.

In Table 1, the release conditions of our experiment are presented in addition to the resulting impact velocities and angles, $R_n$, $R_t$, $R_v$, $R_E$ and the rebound angles. The inertia moment of the sample was approximated to a full sphere in the calculation of rotational energy. Before impact, the angular velocities did not exceed 3 rad/s, and the rotational energy of the rock only accounted for 0.01%-0.03% of the total kinetic energy in this study.

**Table 1.** Release conditions and results of our experiments. $D$ − the diameter of samples (cm); $\theta$ − the inclined angle of the landing plate (°); $H$ − the release height (m); $v_i$ − the impact velocity (m/s); $\alpha$ −the impact angle (°); $R_n$ − the normal coefficient of restitution; $R_t$ −the tangential coefficient of restitution; $R_v$ − the kinematic coefficient of restitution; $R_E$ − the kinetic energy coefficient of restitution; $\beta$ −the rebound angles (°). $N$ denotes the test numbers for one release condition.

| $D$ | $\theta$ | $H$ | $v_i$ Mean | $v_i$ Std Dev | $\alpha$ Mean | $\alpha$ Std Dev | $R_n$ Mean | $R_n$ Std Dev | $R_t$ Mean | $R_t$ Std Dev | $R_v$ Mean | $R_v$ Std Dev | $R_E$ Mean | $R_E$ Std Dev | $\beta$ Mean | $\beta$ Std Dev | $N$ |
|---|---|---|---|---|---|---|---|---|---|---|---|---|---|---|---|---|---|
| 10 | 30 | 2.5 | 6.50 | 0.11 | 57.25 | 1.41 | 0.42 | 0.09 | 0.80 | 0.03 | 0.56 | 0.04 | 0.36 | 0.05 | 38.31 | 6.21 | 4 |
|  |  | 3.5 | 7.75 | 0.06 | 57.21 | 1.44 | 0.38 | 0.04 | 0.92 | 0.08 | 0.59 | 0.04 | 0.39 | 0.04 | 32.88 | 3.94 | 3 |
|  |  | 4.5 | 9.06 | 0.16 | 58.27 | 1.33 | 0.39 | 0.03 | 0.88 | 0.04 | 0.57 | 0.01 | 0.37 | 0.02 | 35.49 | 3.23 | 3 |
|  | 45 | 2.5 | 6.41 | 0.17 | 39.42 | 1.24 | 0.53 | 0.04 | 0.82 | 0.09 | 0.72 | 0.05 | 0.57 | 0.07 | 28.47 | 3.52 | 4 |
|  |  | 3.5 | 7.72 | 0.09 | 39.07 | 0.66 | 0.52 | 0.02 | 0.78 | 0.08 | 0.69 | 0.06 | 0.52 | 0.08 | 28.48 | 2.85 | 3 |
|  |  | 4.5 | 8.76 | 0.10 | 39.77 | 0.20 | 0.58 | 0.07 | 0.82 | 0.06 | 0.67 | 0.04 | 0.50 | 0.05 | 30.47 | 3.76 | 3 |
|  | 60 | 2.5 | 6.51 | 0.21 | 25.06 | 1.21 | 0.76 | 0.06 | 0.69 | 0.03 | 0.70 | 0.03 | 0.59 | 0.04 | 27.17 | 1.77 | 3 |
|  |  | 3.5 | 7.94 | 0.16 | 25.76 | 0.91 | 0.88 | 0.11 | 0.70 | 0.12 | 0.74 | 0.08 | 0.64 | 0.10 | 32.13 | 8.13 | 3 |
|  |  | 4.5 | 8.80 | 0.17 | 26.99 | 1.62 | 0.53 | 0.17 | 0.70 | 0.02 | 0.67 | 0.02 | 0.51 | 0.03 | 20.92 | 5.70 | 3 |
|  | 75 | 2.5 | 6.47 | 0.16 | 12.79 | 1.71 | 1.00 | 0.11 | 0.53 | 0.03 | 0.56 | 0.02 | 0.44 | 0.02 | 23.35 | 3.16 | 3 |
|  |  | 3.5 | 7.91 | 0.08 | 10.06 | 0.73 | 1.11 | 0.27 | 0.65 | 0.19 | 0.67 | 0.17 | 0.58 | 0.21 | 18.65 | 7.38 | 3 |
|  |  | 4.5 | 9.12 | 0.20 | 9.76 | 1.21 | 1.68 | 0.30 | 0.49 | 0.08 | 0.57 | 0.04 | 0.45 | 0.05 | 31.24 | 9.73 | 3 |
| 20 | 30 | 2.5 | 6.47 | 0.20 | 57.69 | 0.75 | 0.40 | 0.01 | 0.87 | 0.03 | 0.58 | 0.02 | 0.38 | 0.02 | 35.77 | 1.49 | 3 |
|  |  | 3.5 | 7.80 | 0.11 | 58.09 | 1.44 | 0.31 | 0.03 | 0.89 | 0.11 | 0.54 | 0.03 | 0.34 | 0.02 | 29.41 | 4.18 | 3 |
|  |  | 4.5 | 8.89 | 0.19 | 58.02 | 1.63 | 0.37 | 0.03 | 0.85 | 0.05 | 0.55 | 0.04 | 0.34 | 0.05 | 34.90 | 3.15 | 4 |

| | | | | | | | | | | | | | | | | |
|---|---|---|---|---|---|---|---|---|---|---|---|---|---|---|---|---|
| *45* | *2.5* | *6.47* | *0.21* | *40.97* | *3.14* | *0.56* | *0.03* | *0.73* | *0.02* | *0.66* | *0.02* | *0.53* | *0.04* | *33.89* | *4.48* | *3* |
| | *3.5* | *7.96* | *0.05* | *40.05* | *1.55* | *0.49* | *0.05* | *0.70* | *0.04* | *0.62* | *0.04* | *0.46* | *0.06* | *30.54* | *3.03* | *3* |
| | *4.5* | *8.74* | *0.13* | *40.16* | *0.22* | *0.47* | *0.09* | *0.86* | *0.04* | *0.72* | *0.04* | *0.56* | *0.05* | *24.68* | *3.77* | *3* |
| *60* | *2.5* | *6.33* | *0.05* | *26.76* | *2.40* | *0.76* | *0.08* | *0.72* | *0.02* | *0.73* | *0.03* | *0.65* | *0.06* | *28.15* | *2.71* | *3* |
| | *3.5* | *7.57* | *0.09* | *24.96* | *0.38* | *0.74* | *0.13* | *0.66* | *0.07* | *0.68* | *0.05* | *0.56* | *0.07* | *27.64* | *5.60* | *3* |
| | *4.5* | *8.76* | *0.19* | *27.44* | *1.57* | *0.64* | *0.12* | *0.63* | *0.10* | *0.63* | *0.07* | *0.49* | *0.11* | *28.29* | *6.11* | *3* |
| *75* | *2.5* | *6.08* | *0.16* | *11.39* | *1.52* | *0.79* | *0.07* | *0.67* | *0.13* | *0.71* | *0.09* | *0.65* | *0.09* | *14.32* | *5.67* | *3* |
| | *3.5* | *7.30* | *0.29* | *11.56* | *2.05* | *1.08* | *0.21* | *0.64* | *0.16* | *0.69* | *0.11* | *0.59* | *0.15* | *19.58* | *5.83* | *3* |
| | *4.5* | *7.97* | *0.26* | *10.63* | *2.90* | *1.17* | *0.18* | *0.62* | *0.16* | *0.68* | *0.12* | *0.58* | *0.15* | *19.41* | *3.23* | *3* |

## 3 Analysis of the results and comparison with existing studies

### 3.1 Effect of the impact angle on the coefficients of restitution based on our tests

Although the mean values and standard deviations have been calculated in terms of various release conditions, data points are considered in this section to provide a broad perspective for an evaluation of the effect of the impact angle. Four different inclined angles of the landing plate ($\theta$=30°, 45°, 60° and 75°) induce four intervals of impact angles: 55°<$\alpha$<60°, 36°<$\alpha$<44°, 23°<$\alpha$<30° and 6°<$\alpha$<15°. The mean values of the coefficients of restitution are computed for the four intervals. In Fig. 5, solid lines are adopted to represent the mean values for samples with size D=10 cm, and dashed lines represent the mean values for size D=20 cm.

The effect of impact angle on the normal coefficient of restitution $R_n$ is shown in Fig. 5a. When the impact angle is smaller than 15°, the values of $R_n$ range from 0.709 to 1.989, and more than 60% of the values of $R_n$ are larger than 1.0. A larger impact angle tends to produce a smaller value of $R_n$ and reduce the discreteness. Initially, the solid line is above the dashed line, although the gap narrows with increasing impact angle. When the impact angle is larger than 30°, these two lines do not exhibit a clear difference. Therefore, small specimens are more likely to have a higher $R_n$ than large specimens with small impact angles, and the effect of the rock size on $R_n$ can be neglected when the impact angle is more than 30°.

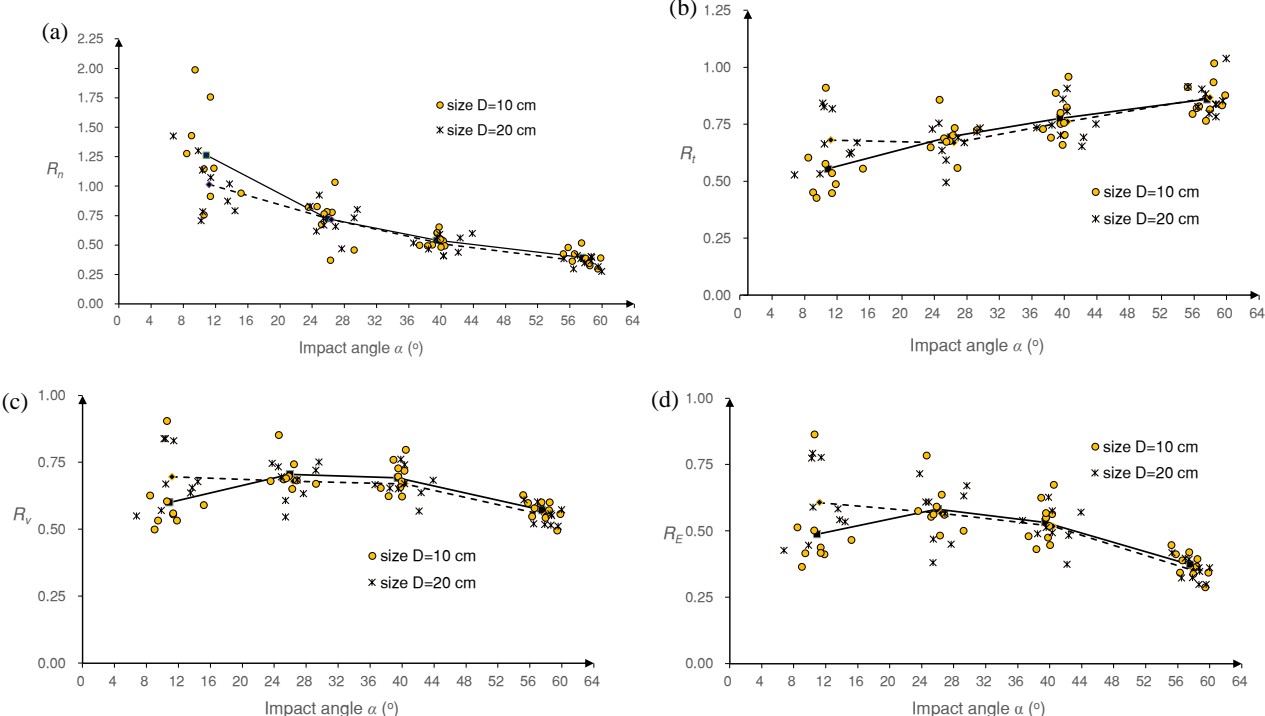

**Figure 5.** The variations of four coefficients of restitution, **(a)** the normal coefficient $R_n$, **(b)** the tangential coefficient $R_t$, **(c)** the kinematic coefficient $R_v$ and **(d)** the kinetic energy coefficient $R_E$, with the impact angle $a$ in this study, and the mean value lines for samples with size D=10 cm (solid lines) and D=20 cm (dashed lines).

As Fig. 5b shows, the impact angle has a different effect on the tangential coefficient of restitution $R_t$ compared to $R_n$. First, the discreteness of data points has not been reduced as the impact angle increases. Then, $R_t$ increases slightly with increasing impact angle. In the first impact angle interval, the solid line is below the dashed line, which implies that small specimens gain a lower $R_t$ than large specimens with small impact angles. Until the impact angle reaches 23 °, the two lines have no distinct difference to be distinguished. Overall, the mean value lines in Fig. 5b seem to accord with the linear correlation between $R_t$ and the impact angle $\alpha$ (Wu, 1985).

Furthermore, the kinematic coefficient of restitution $R_v$ versus the impact angle is plotted in Fig. 5c. As the impact angle increases, the peak values of $R_v$ of the four impact angle intervals decrease gradually, and $R_v$ becomes concentrated. However, the mean values present a more complicated trend in Fig. 5c. Except where the solid line rises from the first impact angle interval to the second, the mean values generally decline. The decline is tiny from the second impact angle interval to the third, while it is apparent from the third to the fourth. Taking the small gap between the mean lines into account, larger specimens obtain a small $R_v$ more easily than small specimens when the impact angle is more than 23 °.

Finally, the effect of the impact angle on the coefficient of kinetic energy restitution $R_E$ is illustrated in Fig. 5d. Similar to Fig. 5c, the peak values of $R_E$ of the four impact angle intervals decrease with increasing impact angle. However, the discreteness of data points does not disappear clearly until the fourth impact angle interval. The trends for the mean value of

$R_E$ are similar to $R_v$. However, the decline in the mean values of $R_E$ is more intuitive than $R_v$ from the second impact angle interval to the third. The gap between the mean lines of $R_E$ is narrower than $R_v$ with larger impact angles. Although some scholars (Chau 2002; Asteriou 2012) suggested that smaller impact angles induce less kinetic energy loss and higher $R_E$, the deduction may not be suitable for small impact angles in this study.

In addition to the effect of the impact angle on the four coefficients of restitution, another interesting phenomenon can be observed in Fig. 5. Two sizes are adopted in this experiment to evaluate the effect of rock size on the rebound characteristics. Except for a smaller impact angle, the gaps between the two mean lines in Fig. 5 are much smaller compared to the magnitudes of the restitution coefficients. Therefore, the four coefficients of restitution seem to be independent of the sample sizes in our test when the impact angle exceeds 23 °, which could be attributed to the test conditions. As Farin et al. (2015)

noted, the thickness of the impacted objective is an important factor in determining whether the coefficients of restitution change with the boulder size. When the impacted objective has a large thickness compared to the boulder size, the coefficient of restitution is independent of the boulder size. In this study, the impacted objective is concrete slabs fixed in the ground, which has enough thickness to eliminate the effect of rock size on the coefficients of restitution in case of large impact angles.

In addition, data points for all coefficients except $R_t$ become concentrated as the impact angle increases. In the first impact angle interval, the diversity of $R_n$ is clearly larger than the other three coefficients. However, when the impact angle exceeds 23 °, the lowest diversity occurs for $R_n$, and the second is for $R_v$. Various functions were considered to match data points, but no function can provide a correlation coefficient $R^2$ more than 0.60 in terms of $R_t$, $R_v$ and $R_E$ for all options considered. The power function provides the best $R^2$ in matching data points of $R_n$, which reaches 0.80. Therefore, the scaling law to describe

data points is abandoned in this study. Although Asteriou et al. (2012) suggested that $R_v$ was more suitable than $R_n$ for use in correlations with the impact angles, it is invalid in this study, which is caused by the variations in test conditions.

### 3.2 Comparison with existing small-scale experiments

In this section, the effect of the impact angle on the coefficient of restitution obtained in this study is compared with some existing small-scale experiments to determine the effect of the test scale. Tests conducted by Chau et al. (2002), Cagnoli and

Manga (2003), and Asteriou et al. (2012) are selected here for the data availability. The test conditions of those studies are provided in Table 2 in comparison with this study. This study mainly differs from the other studies in terms of the size and mass of the samples.

**Table 2. Test conditions of previous studies and our experiment**

|  | This study | Chau et al, 2002 | Cagnoli and Manga, 2003 | Asteriou et al., 2012 |
|---|---|---|---|---|
| Sample shape | Spherical Polyhedron | Sphere | Cylinder | Cubic |
| Sample material | limestone | dental plaster | pumice | marble/sandstone/marl |
| Landing plate material | concrete | dental plaster | pumice | marble |

| | | | | |
|---|---|---|---|---|
| Sizes (mm) | Diameter: 100/200 | Diameter: 18.35/60/60 | Basal diameter: 5.5 Length: 8.9 | Edge:20 |
| Mass (g) | 1200/10000 | 6.05/153.64/204.33 | 0.11 | 19.9/20.3/16.5 |
| Impact velocities (m/s) | 6.7-9.3 | 4-5.65 | 24.88 | 3 |
| Impact angles | 6°-60° | 20°-80° | 18º-74° | 16°-77º |
| Roughness of the target surface | uneven | flat | flat | smooth |
| Sharpness of the specimens | rear edges | no edges | rear edges | smooth edges |

Although previous studies imposed various test conditions, they provided references for us to evaluate the effect of the test scale. The effects of the impact angle on $R_n$, $R_t$, $R_v$ and $R_E$ provided by Chau et al. (2002), Cagnoli and Manga (2003), Asteriou et al. (2012) are plotted in Fig. 6 with this study. In Chau's results, the specimen with a mass of 204.33 g was selected because that mass is closest to those of our samples. In Asteriou's results, we chose a marble specimen as the reference because marble and limestone have nearly the same hardness. In the absence of detailed data, only the trend line results in the related literature are extracted and redrawn in Fig. 6 to make a comparison. Different line styles are adopted for trend lines in Fig. 6. The lines with data markers are the mean value lines, while those lines without data markers are fitting lines. Two lines, $R_v$ versus the impact angle $a$ for Cagnoli and Manga's test and $R_E$ versus the impact angle $a$ for Asteriou's test, are absent in Fig. 6 because the literature did not provide them. In addition, ①a and ①b are used to represent the results for D=10 cm and D=20 cm in this study, respectively.

Although the results of the previous studies and our tests are quite different when they are plotted in one figure, some general trends could be observed. First, all of the trends in $R_n$ versus the impact angle are consistent (see Fig. 6a): $R_n$ decreases with increasing impact angle. Asteriou's tests offered the maximum $R_n$ in most cases, which can be attributed to the lighter mass and lower impact velocities adopted in the tests. The small $R_n$ values in Cagnoli and Manga's tests were due to the weak strength of pumice whose damage upon impact dissipates kinetic energy, and the impact velocity in this test is much higher than the others. Compared to the other tests, our results produce the steepest descent at the beginning of the trend line. Linear function has been suggested to be used to describe the correlation of $R_n$ and the impact angle $\alpha$ in several reports (Wu, 1985; Richards et al., 2001), although we cannot offer a definitive conclusion that linear functions are the best choice. In Fig. 6a, the fitting curve ③ is a second-order polynomial, and the fitting curve ④ is a power function. As mentioned above, it is also found that the best correlation coefficient $R^2$ is provided by the power function when matching $R_n$ in this study.

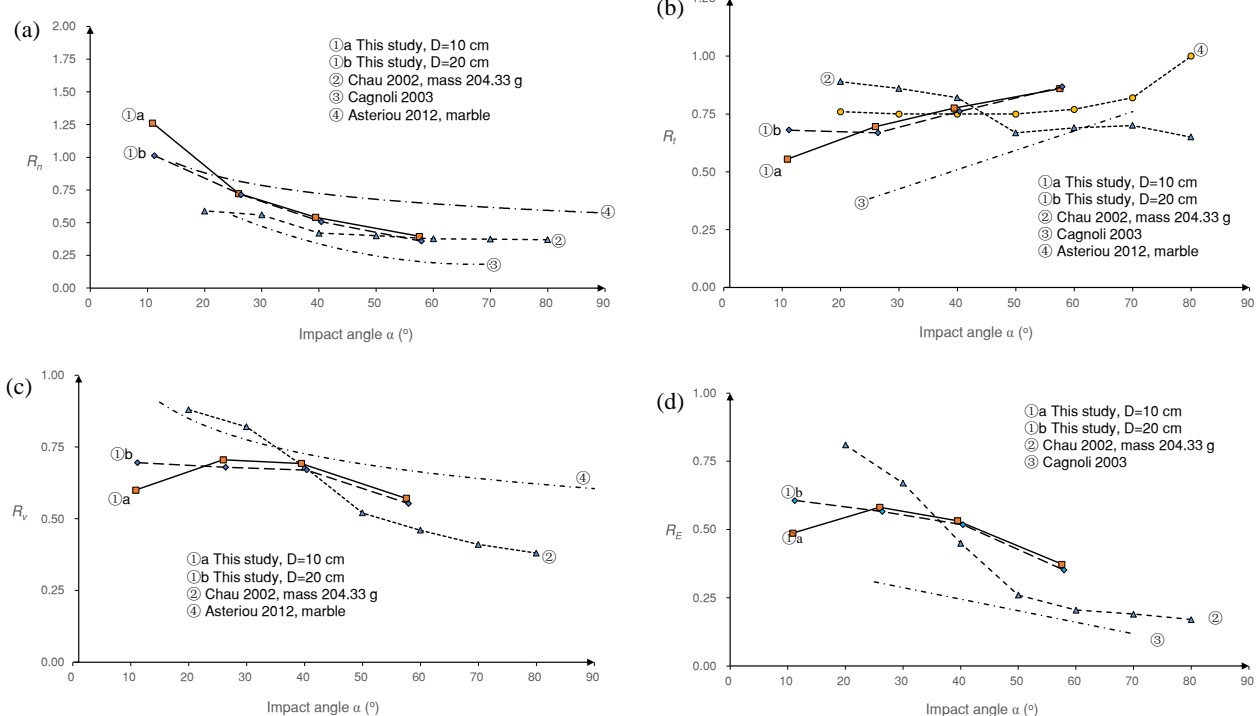

**Figure 6.** Results Comparison between this study and existing small-scale experiments in terms of the effect of the impact angle on the four coefficients of restitution: **(a)** The normal coefficient $R_n$, **(b)** the tangential coefficient $R_t$, **(c)** the kinematic coefficient $R_v$ and **(d)** the kinetic energy coefficient $R_E$. The mean value lines (lines with data markers) or the fitting lines (lines without markers) are adopted in accordance with the original literature (Chau et al., 2002; Cagnoli and Manga, 2003; Asteriou et al., 2012).

Next, Fig. 6b indicates that the trends of $R_t$ versus the impact angle are scattered. Except for Cagnoli's result, the $R_t$ obtained in the tests are all in the range from 0.5 to 1.0. Wu (1985) suggested that $R_t$ may decrease linearly with increasing slope angle $\theta$. In other words, $R_t$ may experience an improvement with increasing impact angle $\alpha$, which is in line with the results of all experiments except Chau's. Cagnoli and Manga matched $R_t$ using a linear function, which resulted in the fitting curve ④ in Fig. 6b. Therefore, the improvement effect of the impact angle $\alpha$ on $R_t$ is valid in most cases. In addition, if the impact angle is less than 45°, there is an apparent gap between Cagnoli and Manga's result and the other tests. Although the results of the other three experiments are different, the variation ranges of $R_t$ occur regardless of the test conditions.

Finally, the trends of $R_v$ and $R_E$ versus the impact angle are shown in Fig. 6c and 6d, respectively. Cagnoli and Manga's result is not involved in Fig. 6c for its absence, and for the same reason, Asteriou's result is not involved in Fig. 6d. Four unique trend lines are plotted in Fig. 6c, although $R_v$ exhibits a descending trend overall, which means that $R_v$ is reduced in most cases as the impact angle $\alpha$ increases. Similar to $R_v$, all experiments produce downward trend lines for $R_E$, except the initial ascent stage in line ①a, which implies that increasing the impact angle induces more kinetic energy dissipation. However, the trend lines in Fig. 6d are scattering. Clearly, the trend lines for $R_v$ and $R_E$ are more likely to be influenced by

the test conditions than $R_n$ and $R_t$. In Fig. 6c, the fitting curve ④ is a power function, and in Fig. 6d, the fitting curve ③ is a linear function. The difference in the trend lines is apparent for the listed experiments, and we cannot conclude which type of functions should be recommended to match $R_v$ and $R_E$.

In conclusion, various experimental conditions induce different results for $R_n$, $R_t$, $R_v$ and $R_E$, although there are certain trends that occur regardless of the test conditions. The normal coefficient of restitution $R_n$, kinematic coefficient of restitution $R_v$ and kinetic energy coefficient of restitution $R_E$ all decrease with increasing impact angle, while the tangential coefficient of restitution $R_t$ increases as the impact angle increases in most cases. A power function appears suitable for use in fitting data points of $R_n$, while its validity should be further verified by other studies.

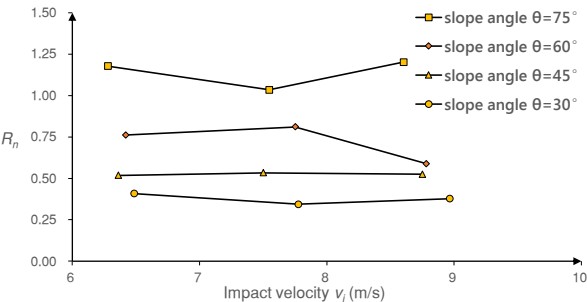

**Figure 7.** The mean value lines of the normal coefficient of restitution $R_n$ versus the impact velocity

In addition, Asteriou's test provided the highest trend lines of $R_n$ and $R_v$ in Fig. 6, while Cagnoli and Manga's test provided the lowest trend lines of $R_n$, $R_t$ and $R_E$. Asteriou's experiment was conducted using the lowest impact velocity in Table 2, while the highest impact velocity was adopted by Cagnoli and Manga. Cagnoli and Manga employed pumice, which has a much weaker strength compared to sample materials in other tests. Asteriou and Tsiambaos (2018) noted that $R_n$ reduces when increasing the impact velocity and increases as the material becomes harder, which partly accounts for the difference between Asteriou's and Cagnoli and Manga's test. However, we cannot make a definitive conclusion regarding which factor in Table 2 is the main reason for the magnitude difference in the coefficients of restitution between the tests compared. The tests differ from each other in multiple test conditions, as Table 2 lists; therefore, the estimation of the effect of one specific factor on the magnitude of the coefficient of restitution is unreasonable using their data together. To evaluate the effect of the impact velocity in this study, Fig. 7 plots the mean value of $R_n$ versus the impact velocity with different slope angles. No determined trend of $R_n$ appears for the limited variation range of the impact velocity.

## 4 Direction transitions of translational velocities and rotation

### 4.1 Direction transitions of translational velocities

Taking the ratio between the rebound angle and the impact angle $\beta/\alpha$ as a reference, the direction transition of the translational velocity versus the impact angle are illustrated in Fig. 8. Assuming that the falling rock is spherical and no

energy dissipation occurs during the collision, the rebound angle should theoretically be equal to the impact angle, which would result in the data points lying on the red line $\beta/\alpha=1$ in Fig. 8.

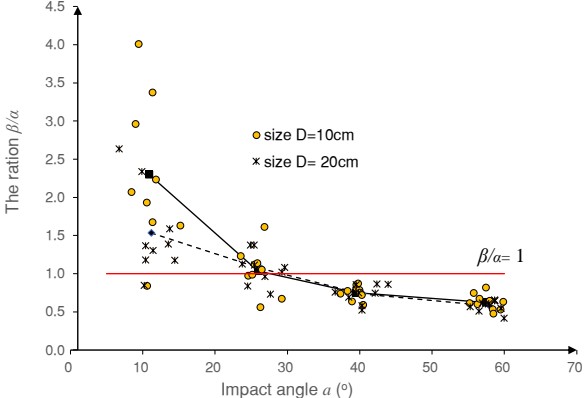

**Figure 8.** The variation of $\beta/\alpha$ with the impact angle and the mean value lines for samples with size D=10 cm (solid lines) and D=20 cm (dashed lines).

However, the test results are almost entirely located above the line in the first impact angle interval, and nearly 50% of the test results are above the line in the second interval. The data points are stably located below the line until the impact angle reaches 36°. As the impact angle increases, the ratio between the rebound angle and the impact angle $\beta/\alpha$ appears to be a clear reduction, and the discreteness of data points decreases. The mean values of $\beta/\alpha$ are still represented by a solid line for samples with size D=10 cm and a dashed line for size D=20 cm. The two mean values line have little difference from the second interval to the fourth. However, under small impact angle conditions, a smaller sample is more likely to have a larger $\beta/\alpha$ than a larger sample.

A rebound angle greater than the impact angle was also observed by Cagnoli and Manga (2003), which does not violate the energy dissipation rule. The experimental results presented in Sect. 3.1 demonstrated that in this study, the kinetic energy loss constituted 40-65% of the total kinetic energy for many data points in the first impact angle interval and constituted 35-55% in the third interval. Therefore, the ratio between the rebound angle and the impact angle cannot be directly used as a reference in estimating whether the energy loss level is high or low.

This phenomenon implies that the rebound motion may have an unexpected direction of translational velocity. Fig. 9 plots the direction transition of translational velocity caused by the impact, in which diagrams are individually drawn for four impact angle intervals. For a uniform expression, the landing plate is denoted as the bottom black line. Although the impact velocity directions are concentrated for each impact angle interval, the rebound velocity directions vary considerably. The variation for the interval 36°<$\alpha$< 44° is the smallest of all intervals.

(a)                      (b)

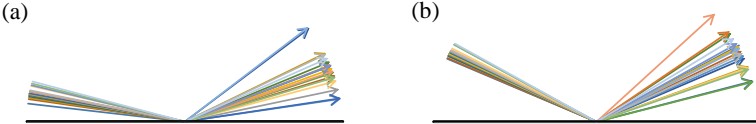

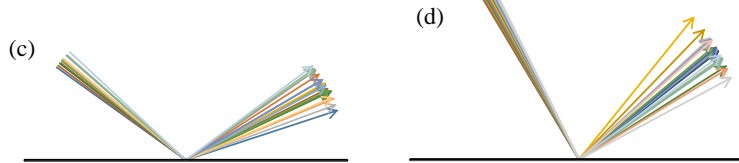

(c)         (d)

**Figure 9.** Direction transitions of translational velocities induced by impacts under different impact angle conditions: **(a)** 6 °≤α<15 °; **(b)** 23 °≤α<30 °; **(c)** 36 °≤α<44 °; **(d)** 55 °≤α<60 °.

## 4.2 The rotation caused by the impact

Except for the direction transition of translational velocity, the rotation is another significant consequence of the impact. Despite little rotation before impact, the samples experienced an observable rotation after impact in this study, and the angular velocities were recorded and involved in the calculation of the kinetic energy coefficient of restitution $R_E$ in Table 1. Considering that the magnitudes of kinetic energy before impact varied in this study, the percentage between $E_{rr}$ and $E_i$ is used to denote how much kinetic energy is dissipated in rotation after the impact. As mentioned in Sect. 1.1, $E_i$ is the total

kinetic energy before impact, and $E_{rr}$ is the rotation energy after impact.

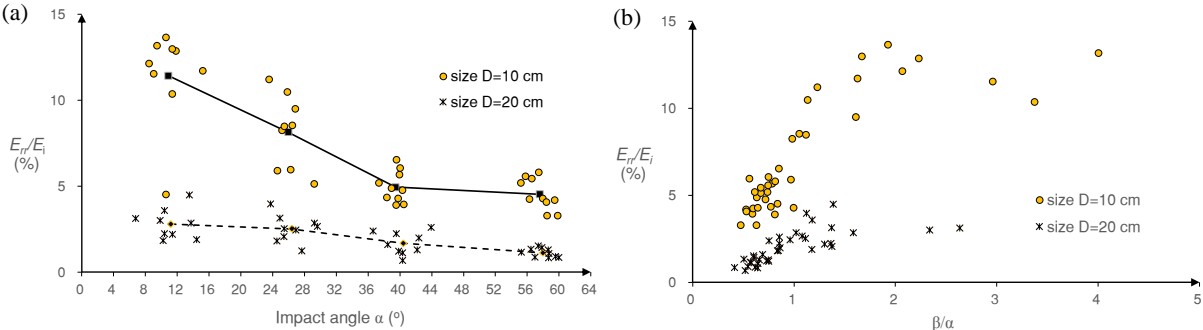

**Figure 10.** The effect of **(a)** the impact angle and **(b)** the ratio $\beta/\alpha$ on $E_{rr}/E_i$

Fig. 10a shows the effect of the impact angle on $E_{rr}/E_i$. A solid line represents the mean values for specimens with size D=10 cm, while a dashed line is for size D=20 cm. First, the difference is most remarkable between the two sample sizes in Fig.

10a. $E_{rr}/E_i$ ranges from 3.3% to 13.7% for size D=10 cm and ranges from 0.7% to 4.5% for size D=20 cm, which means that small samples are more likely to have a high $E_{rr}/E_i$ than larger samples. Next, $E_{rr}/E_i$ decreases as the impact angle increases. For size D=10 cm, $E_{rr}/E_i$ experiences a steep decline from the first impact angle interval to the third and then decreases gently to the fourth. For size D=20 cm, $E_{rr}/E_i$ has a gradual reduction over time, which may be attributed to its small variation range. Finally, the improvement of the impact angle results in more concentrated data points, although data points

for size D=10 cm are always more scattering than size D=20 cm. Both the impact angle and the sample size have an important impact on $E_{rr}/E_i$ in this study. In conclusion, larger samples are more likely to have a stead and small $E_{rr}/E_i$ than small samples, and a large impact angle leads to a small $E_{rr}/E_i$.

As mentioned in Sect. 4.1, the unexpected direction transition of translational velocity always occurs when the impact angle is small. The correlation between the direction transition of translational velocity and $E_{rr}/E_i$ is investigated using $\beta/\alpha$ (the ratio between the rebound angle and the impact angle) as a reference. Fig. 10b illustrates the trends for $E_{rr}/E_i$ versus $\beta/\alpha$. $E_{rr}/E_i$ increases as the ratio $\beta/\alpha$ increases. If $\beta/\alpha$ is smaller than 1.0, $E_{rr}/E_i$ are much concentrated, which ranges from 3.3% to

6.5% for size D=10 cm and ranges from 0.7% to 2.6% for size D=20 cm. With increasing $\beta/\alpha$, the data points become scattering. Moreover, the improvement of $E_{rr}/E_i$ appears to be terminated when $\beta/\alpha$ reaches a specific value. Therefore, we can conclude that a strong correlation occurs between $E_{rr}/E_i$ and $\beta/\alpha$. For a given impact angle, larger rebound angles indicate that more kinetic energy is converted to rotational energy during the collision.

## 4.3 The correlation between the coefficients of restitution and the rotation

The rotation plays an important role in energy dissipation during impact, especially for small samples. The percentage between the resulting rotational energy and the original total kinetic energy decreases as the impact angle increases. The correlation between $E_{rr}/E_i$ and the coefficients of restitution is investigated in this section to evaluate the effect of rotation. Fig. 11 plots the coefficients of restitution versus $E_{rr}/E_i$ for this study. The four coefficients fall into two categories according to their responses to $E_{rr}/E_i$. The first category includes $R_n$ and $R_t$, the two most commonly used coefficients of restitution,

which appears to be strongly correlated with $E_{rr}/E_i$. As $E_{rr}/E_i$ increases, $R_n$ increases but $R_t$ decreases, which verified Broili's deduction (1973). The rotation generated from impact results in an increased normal velocity and reduced tangential velocity. Furthermore, in case that more kinetic energy is converted to rotational energy during the collision, the collision yields a higher $R_n$ and lower $R_t$. In Fig. 11a and 11b, data points for two sizes are not mixed, which can be attributed to the effect of sample sizes on the magnitude of $E_{rr}/E_i$. In addition, the data points become more scattering with increasing $E_{rr}/E_i$. $R_v$ and $R_E$

belong to the second category. There is no remarkable correlation between them and $E_{rr}/E_i$, as shown in Fig. 11c and 11d, so $R_v$ and $R_E$ are independent of the rotation motion in this study. In conclusion, the improvement in the percentage of kinetic energy converted to rotational energy leads to a larger $R_n$ and a smaller $R_t$, while it has no distinct influence on $R_v$ and $R_E$.

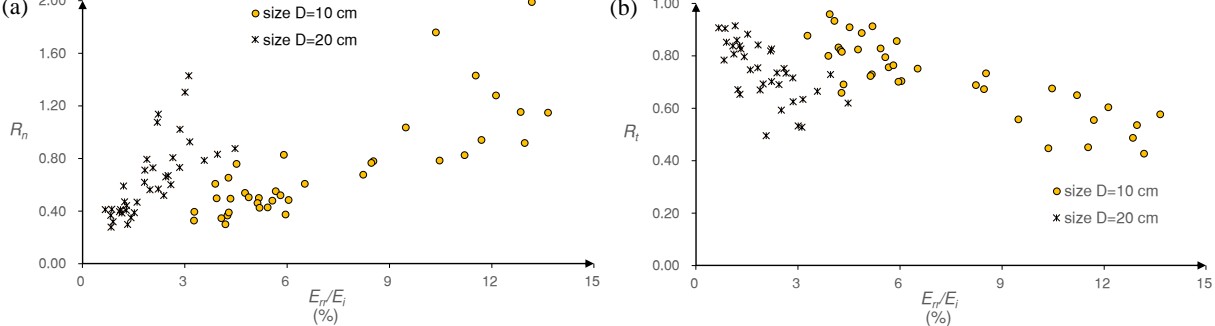

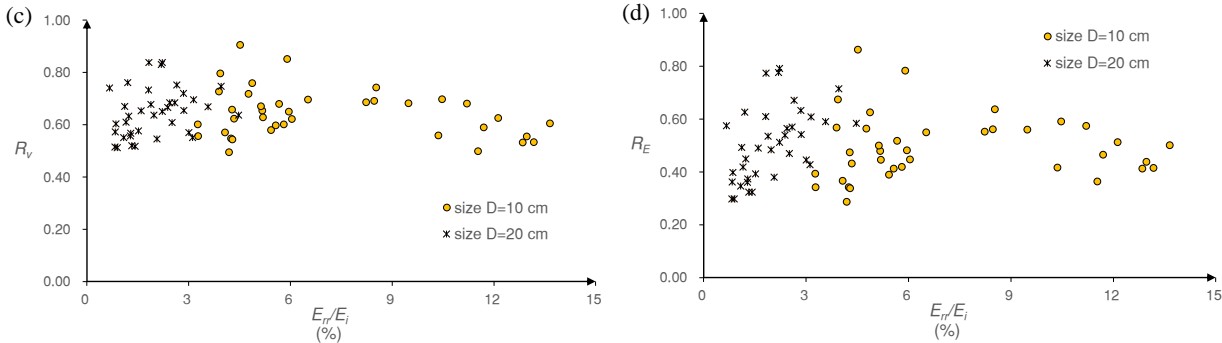

**Figure 11.** The variations of four coefficients of restitution, **(a)** the normal coefficient $R_n$, **(b)** the tangential coefficient $R_t$, **(c)** the kinematic coefficient $R_v$ and **(d)** the kinetic energy coefficient $R_E$, with the ratio $E_{rr}/E_i$

As illustrated in Fig. 10a, more kinetic energy is converted to rotational energy during the collision with a smaller impact angle. Considering the effect of $E_{rr}/E_i$ on $R_n$ and $R_t$, a smaller impact angle is more likely to have a high $R_n$ and a low $R_t$ than a larger impact angle. Therefore, $R_n$ typically decreases with increasing impact angle, and $R_t$ increases as the impact angle increases. When the impact angle is small, two sample sizes show a clear distinction in $E_{rr}/E_i$, as shown in Fig. 10a, which results in the difference in the mean values of $R_n$ and $R_t$ between two sizes in the first impact angle interval in Fig. 5a and 5b.

## 5 Discussion

The test results demonstrated the correlation between the rotation and the effect of the impact angle on the coefficients of restitution. Under free fall conditions, a higher percentage of kinetic energy converted to rotational energy always induces a higher $R_n$ and a lower $R_t$. The percentage can be associated with the ratio between the rebound angle and the impact angle $\beta/\alpha$. As the impact angle decreases, the ratio $\beta/\alpha$ increases, and more kinetic energy is converted to rotational energy. In this section, the reason why a small impact angle achieves a high $\beta/\alpha$ more easily and its consequences are discussed.

### 5.1 The main reason for the high $\beta/\alpha$ in the case of small impact angles

When the impact angle is small, the rebound angle easily exceeds the impact angle and causes a high $\beta/\alpha$, which can be associated with the impact orientation and the damage caused by the impact. The sample has irregular cutting facets and rear edges in this study, while the landing plate was made of concrete slabs of a smaller hardness compared to the falling samples. Suppose that the spherical polyhedron impacts the landing plate with a corner or an edge, damages will occur. Fig. 12a shows the indentations on the surface caused by the impacts and the rough areas resulting from the repeated damages. For an individual indentation, the diameter $d$ and the depth $h$ are measured and noted in Fig. 12a.

The configuration of indentation is simplified as Fig. 12b to evaluate the effect of the indentation on the rebound direction. Once the impact compression ends, the indentation is formed completely, and the rebound motion starts. The rebound angle is constrained by the border of the resulting indentation. The restriction is less susceptible to a small rebound angle because a translational motion along the dashed arrow indicates additional penetration. Theoretically, the rebound angle will be less

than the impact angle accounting for energy loss. When the impact angle is sufficiently large to generate a rebound angle as the solid arrow, the border imposes no constraints on the rebound motion, and the sample can leave with the default rebound angle. However, when the impact angle is small and generates a default rebound angle as shown by the dashed arrow, rotation motion must be involved to overcome the constraint, and an unexpectedly larger rebound angle occurs. Thus, the penetration caused by the impact may contribute to the high $\beta/\alpha$ in the case of small impact angles.

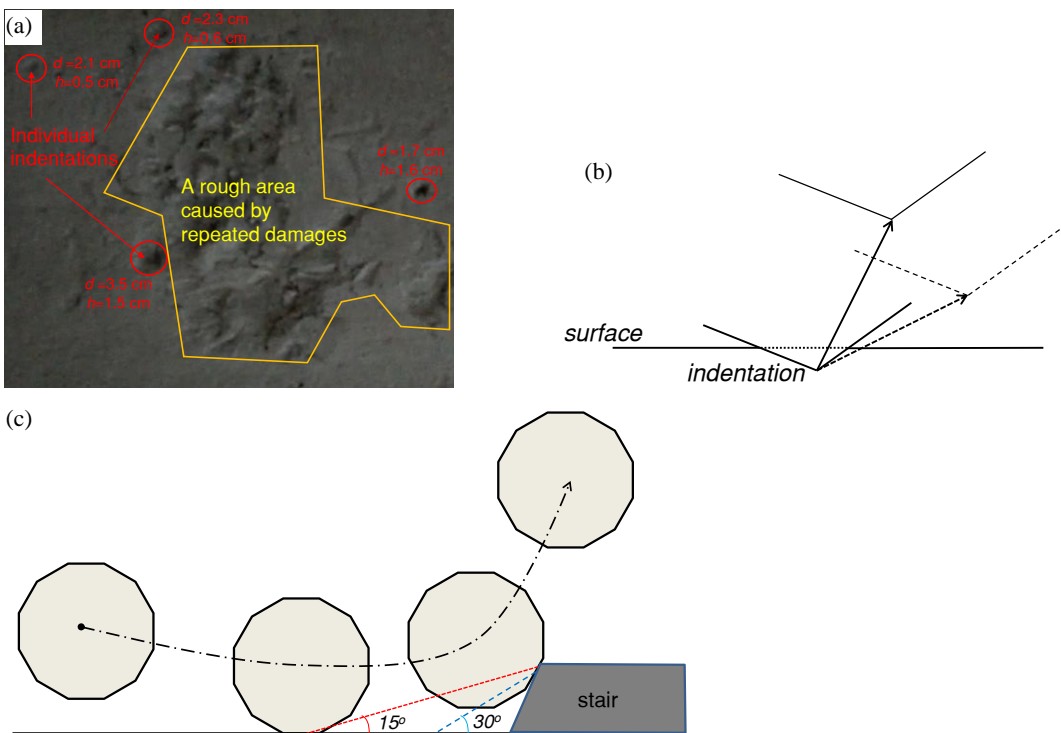

**Figure 12. (a)** Individual indentations and a rough area caused by the impacts and their influence on the rebound motion: **(b)** a simplified configuration used to evaluate the effect of the indentation; **(c)** a small stair model used to evaluate the effect of the macro roughness

Another important factor in generating a high $\beta/\alpha$ is the macro roughness of the landing plate, which comes from repeated damage to the slab surface. Assuming that the macro roughness of the landing plate is represented as a small stair in Fig. 12c, the interaction between the falling sample and surface may have two stages in certain situations. The sample impacts the surface before the stair and starts leaving in the first stage. Then, the sample contacts the stair, and the velocity changes again in the second stage. The time interval between the two stages is so short that the two stages appear to finish simultaneously. The probability that two-stage interactions occur is related to the magnitude of the impact angle. As Fig. 12c illustrates, if the default rebound angle is 15 °, the stair can affect the rebound motion if the sample contacts the surface within 3.73 times the stair height before the stair. As the default rebound angle increases, the surface region where the stair can affect the rebound motion decreases. Considering that a smaller impact angle will, in theory, induce a smaller rebound angle, the reduction of the impact angle must improve the risk of the sample contacting the stair. When the impact angle is small, the sample has

more possibility of having a two-stage interaction and leaving the plate with a rebound angle larger than the impact angle, and a high $\beta/\alpha$ will occur.

In conclusion, the restriction from the configuration of the indentation, as well as the macro roughness caused by repeated damage, is more likely to affect the rebound motion when the impact angle is small. As a consequence, the rebound angle easily exceeds the impact angle in the case of a small impact angle, which ultimately results in a high $\beta/\alpha$.

### 5.2 Interpretation of the normal coefficient of restitution $R_n$ larger than 1.0

Of the various consequences of the rebound angle being greater than the impact angle, high values of the normal coefficient of restitution $R_n$ may be remarkable. Engineers usually take 1.0 as the upper bound of $R_n$ in computer codes, whereas several scholars have reported $R_n$ values larger than 1.0 (Azzoni et al., 1992; Paronuzzi 2009; Spadari et al., 2012). In this section, the relationship between $R_n$ and the direction transition of translational velocity is investigated.

Considering that the rotation before impacting is small in this study, the normal coefficient of restitution $R_n$ can be expressed as Eq. (5) based on the basic definition in Sect. 1.1.

$$R_n = v_{nr} / v_{ni} = \sqrt{E_{rt} / E_{it}} \times \left( \sin\beta / \sin\alpha \right) = \sqrt{E_{rt} / E_i} \times \left( \sin\beta / \sin\alpha \right) \tag{5}$$

By introducing an angle coefficient

$$\lambda = \sin\beta / \sin\alpha \tag{6}$$

Eq. (6) can be simplified as

$$R_n = \lambda \sqrt{E_{rt} / E_i} \tag{7}$$

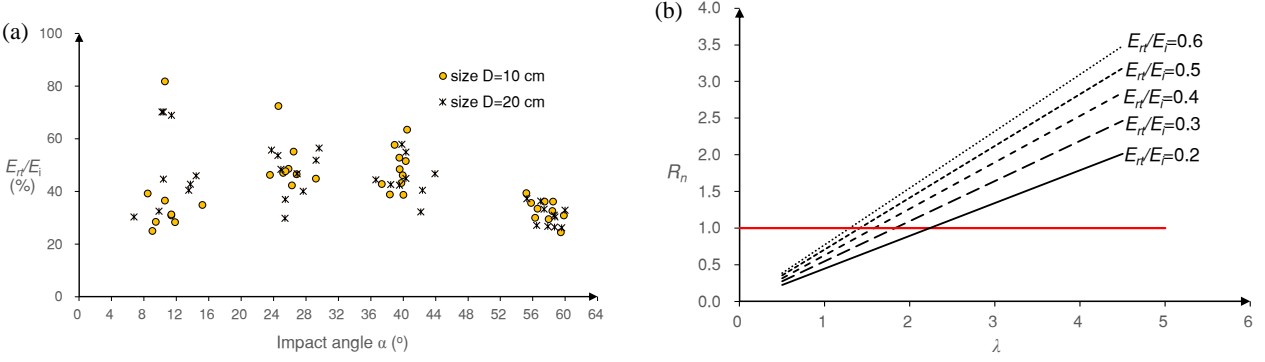

**Figure 13. (a)** The variation of the ratio $E_{rt}/E_i$ with the impact angle and **(b)** the correlation between the normal coefficient of restitution $R_n$ and the angle coefficient $\lambda$ under different $E_{rt}/E_i$ levels.

$E_{rt}/E_i$, the ratio between the translational energy after impact and the total kinetic energy before impact, is plotted in Fig. 13a with respect to the impact angle. As the impact angle increases, the mean value of $E_{rt}/E_i$ increases when the impact angles are smaller than 36º and then decreases. The peak values of $E_{rt}/E_i$ of the four impact angle intervals decrease gradually with increasing impact angles. In this study, the values of $E_{rt}/E_i$ are located in the range (0.20, 0.60) in most cases, which provides us with a reference to explore the conditions of $R_n$ larger than 1.0.

Fig. 13b plots the relationship between $R_n$ and the angle coefficient $\lambda$ under different $E_{rt}/E_i$. The value of $R_n$ increases when increasing the angle coefficient $\lambda$. Even if $E_{rt}/E_i$ is only 0.2, $R_n$ is greater than 1.0 when $\lambda>2.24$. An extremely large rebound angle is not needed to generate such a $\lambda$ when the impact angle is small. For example, when the impact angle is 12 ° and 15 °, a rebound angle of 27.8 ° and 35.5 ° is sufficient to obtain $\lambda>2.24$. Assuming that $E_{rt}/E_i$ is unchanged, a case in which the

rebound angle is larger than the impact angle must lead to a higher $R_n$. Although the value of $\lambda$ corresponding to $R_n=1.0$ varies with $E_{rt}/E_i$, the condition $\lambda>1.0$ is required to obtain $R_n$ greater than 1.0. As shown in Fig. 13b, $R_n$ cannot exceed 1.0 if the rebound angle is lower than the impact angle. As discussed in the previous sections, small impact angles easily result in unexpectedly large rebound angles. If the angle coefficient $\lambda$ formed by the rebound and the impact angle is sufficiently large, $R_n$ will exceed 1.0 even though $E_{rt}/E_i$ is small. Furthermore, assuming a constant $E_{rt}/E_i$, the reduction in the impact angle

decreases the threshold value of the rebound angle that should be satisfied to achieve an $R_n$ in excess of unity, which means that smaller impact angles are more likely to yield $R_n$ larger than 1.0.

## 5.3 Relation between the normal coefficient of restitution and the kinetic energy loss

A smaller impact angle more easily obtains a high $\beta/\alpha$ and a high percentage of kinetic energy converted to rotational energy, which then induces a higher $R_n$. However, the kinetic energy coefficient of restitution $R_E$ appears independent of the

percentage of kinetic energy converted to rotational energy. Therefore, simply treating a higher $R_n$ as a symbol of lower kinetic energy loss may be unreasonable. Stronge (1991) indicated that in the valuation of kinetic energy dissipation, the normal coefficient of restitution is only reliable for nonfrictional collisions. Under frictional collisions conditions, the total kinetic energy may have a paradoxical increase if the normal coefficient of restitution is adopted as the unique reference. As shown in Fig. 14, the correlation between $R_n$ and $R_E$ is more complicated in this study, which verifies Stronge's argument.

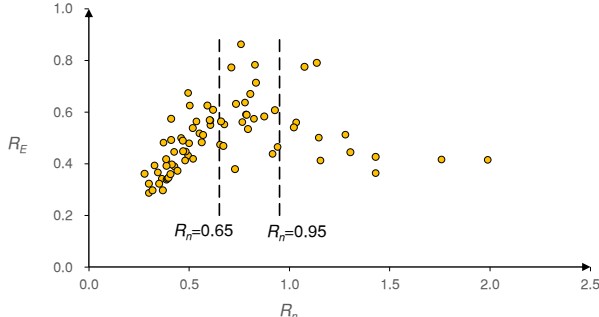

**Figure 14.** The variation of the kinetic energy coefficient of restitution $R_E$ with the normal coefficient $R_n$

Increasing $R_n$ will increase $R_E$ initially but decrease it overall. For simplicity, two boundaries ($R_n=0.65$ and $R_n=0.95$) are added in Fig. 14. The kinetic energy coefficient of restitution $R_E$ increases with increasing $R_n$ when $R_n<0.65$, in agreement with the relationship between $R_n$ and the energy loss level based on the elastic-plastic response analysis. If $R_n$ is greater than

0.95, a larger $R_n$ indicates a smaller $R_E$. High values of $R_n$ are associated with unexpectedly large rebound angles in this study, which means that the unexpectedly large rebound angles can be related to a higher level of kinetic energy loss. $R_E$ is

disordered if $R_n$ lies in (0.65, 0.95), which is caused by the two different trends meeting. Therefore, the normal coefficient of restitution $R_n$ cannot be directly used in the evaluation of the kinetic energy dissipation level.

## 5.4 The difficulty in introducing the effect of impact angle into trajectory simulation

This study, as well as the previous experiments, has demonstrated that the variation of the coefficients of restitution in terms

of the impact angle are significant. For this reason, the impact angle should be involved in determining the coefficients of restitution in rockfall trajectory simulation. However, some problems cause a barrier to developing a reasonable way to account for the effect of the impact angle in computer simulation.

First, although the test scales and conditions have little influence on the general laws that the impact angle affects the coefficients of restitution, it is difficult to construct a uniform formula to reflect the effect under various test conditions.

Taking $R_n$ as an example, the effect of the impact angle on $R_n$ has been formulated by several different functions, such as the linear function (Wu, 1985; Richards et al., 2001), power function (Asteriou et al. 2012) and second-order polynomial (Cagnoli and Manga, 2003). In this study, the power function provides the best correlation coefficient in fitting data points of $R_n$. Furthermore, the mathematic expression regarding the effect of the impact angle on the coefficients of restitution is abandoned in more experiments (Chau et al., 2002; James, 2015). Therefore, we cannot conclude which type of function is

the best choice to describe the effect, and whether a uniform expression occurs is questionable.

Another problem arises from the discreteness of data points. Given the impact angle, the discreteness of data points determines the reliability of the rebound velocity estimated by adopting a typical value of the coefficients of restitution. For all coefficients of restitution, the discreteness of data points experiences a reduction if the impact angle increases in this study. When the impact angle is large, it may be acceptable to predict the rebound using a typical value of the coefficients of

restitution, e.g., the mean value. However, the data points are extremely scattering under small impact angle conditions, which means that using a typical value in the simulation may be unreliable.

Therefore, further research should be carried out to establish a reasonable and comprehensive method to reflect the effect of the impact angle on the coefficients of restitution in rockfall trajectory simulation. The stochastic model has more potential in achieving this target because it accounts for the variation of the coefficients of restitution in terms of various factors based

on data collection (Jaboyedoff et al., 2005; Frattini et al., 2008; Bourrier et al., 2009; Andrew and Oldrich, 2017).

## 6 Conclusions

The coefficients of restitution are critical parameters in the predication of rockfall trajectory by computer codes. Both the terrain characteristics and kinematic parameters can significantly affect the coefficients of restitution. The effect of the impact angles on the coefficients of restitution has been observed, and some laws have been concluded in a series of tests.

Until now, the existing laboratory tests have largely been limited to small scale tests, and whether the previous conclusion is

valid for different scale tests is uncertain. The role of rotation is still unresolved in the effect of the impact angle on the coefficient of restitution.

In the present study, laboratory tests were performed using a 3D motion capture system. Spherical limestone polyhedra with diameters of 10 cm and 20 cm were taken as samples, and C25 concrete slabs were adopted to form the landing plate. By altering the release height and the inclined angle of the landing plate, the effects of the impact angle on the coefficients of restitution were estimated under freefall test conditions. The result comparison between our test and the existing small-scale tests indicated that several general laws occur when accounting for the effect of the impact angle, regardless of the test scales and conditions. The normal coefficient of restitution $R_n$, the kinematic coefficient of restitution $R_v$ and the kinetic energy coefficient of restitution $R_E$ all decrease when increasing the impact angle, while the tangential coefficient of restitution $R_t$ increases as the impact angle increases in most cases. However, the reason for the magnitude difference in the coefficients of restitution between the tests compared is unidentified because the tests differ from each other in multiple test conditions.

In the free fall test, the rotation after impact dissipates part of the kinetic energy of the sample and plays an important role in the effect of the impact angle on the coefficient of restitution. The test results show that the percentage of kinetic energy converted to rotational energy can be associated with the ratio between the rebound angle $\beta$ and the impact angle $\alpha$. When the impact angle is small, the rebound angle is more likely to exceed the impact angle and yields a high $\beta/\alpha$ for the indentations and macro roughness caused by the impacts. As the impact angle decreases, the ratio $\beta/\alpha$ increases, and the percentage of kinetic energy converted to rotational energy increases. Given a $\beta/\alpha$, large samples are more likely to have a stead and small percentage than small samples. A higher percentage of kinetic energy converted to rotational energy always induces a higher $R_n$ and a lower $R_t$. However, no correlations are observed in this study between the rotation energy and the other two coefficients of restitution, $R_v$ and $R_E$. In addition, $R_n$ being larger than 1.0 can be related to the rebound angle being greater than the impact angle under small impact angle conditions.

Although it is verified in this study that several general laws regarding the effect of the impact angle on the coefficients of restitution are independent of the test scales and conditions, we still lack a reliable method to introduce the effect of the impact angle into rockfall trajectory simulation, which is caused by the discreteness of the measured data under small impact angle conditions and the absence of a uniform and reasonable function describing the effect of the impact angle.

Last but not least, only the spherical limestone polyhedrons are taken as the samples, and the detailed impact orientations during impact are not involved in this study. Whether the conclusions are valid for the boulder with other shapes should be further investigated through more elaborate experiments. In view of this, the test results are valid for trajectory simulation codes based on a lumped-mass model and can be referenced in the trajectory predication of spherical rocks impacting hard surfaces using a rigid body model.

## Acknowledgements

Sponsored by Research Fund for Excellent Dissertation of China Three Gorges University, the Open Research Programme of the Hubei Key Laboratory of Disaster Prevention and Mitigation (No. 2017KJZ03), the National Natural Science Funds (No. 51409150), and the CSC Scholarship (No. 201707620009).

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
