# Peer review of "Effects of the impact angle on the coefficient of restitution based on a medium-scale laboratory test"

_Natural Hazards and Earth System Sciences, 2018_

## Referee Comment (RC1) · Anonymous Referee #1 · 30 Apr 2018

The article presents a laboratory study on the dependence of the coefficient of restitution regarding the impact angle, falling height etc. Based on the results a regression has been formulated to obtain normal and tangential coefficients of restitution. The $R^2$ are not very high. This – in my opinion – has one main reason: the blocks are not spherical but have edges an corners. Their impact on the ground mains defines the rebound angle and velocity. The model itself cannot reflect this effect because it neglects the rotational movement of the block that has a significant influence. Therefore, the model presented should be reported as being valid only for trajectory simulation codes based on point masses used to simulate the blocks. The model would not work for simulation codes that use folly shaped three-dimensional blocks. This should be stated

in the introduction, handled in the discussion and be summarized in the conclusions.

**Specific comments:**

P1L13: Please, add short term on the kind of rock movements with or without rotation, "jumping" or vertically falling.

P2L4: Outdated references! Apart from the cited codes there are numerous additional trajectory simulation codes around today (E.g. RAMMS::rockfall, RockyFor3D, Pierre 2, ...). Their impact scenarios partly differ from the classical use of normal and tangential COR. Please update your collection of reference. A (also already 7 years old) list could be found in https://www.nat-hazards-earth-syst-sci.net/11/2617/2011 or also have a look at http://www.nrcresearchpress.com/doi/abs/10.1139/cgj-2016-0039# .WucKWhK5-Lo

P2L19: The COR is a model only. In reality it is almost zero. Example: take a spherical rock and let it fall –> it barely jumps.

P4L23, P9L14, P16L8: replace "increases in the impact" by "increasing"

P5L2: Glover also evaluated coefficients of restitution in http://etheses.dur.ac.uk/10968

P5L12: use kg instead of g because it is doubtful that exact this weight is kept.

P7L5: 60fps might not be enough precise to capture the accelerations (during impact = time of the highest acceleration) there are only very small displacements that are not covered by the resolution of the cameras?

P12L8: Of course, if only translational movements are looked at. The hardness of the impact partners involved is not very relevant. The rebound is influenced mainly from the rock's edges and therefore related to its rotational movement.

P13L7: This is a very precise weight....

P16L30: "Assume" –> "Assuming"

P20L9: The presented concept of COR analysis an experimental/laboratory trajectory regarding the block's centre of gravity. The shape of the block does not play any role as well as its rotational movement. The presented model to determine $R_n$ therefore only works if the trajectory model simulates small mass points without rotational movements. As soon as the trajectory code aims to model spatially shaped blocks with edges and corners above data cannot be used. This consequence should be added to discussion and conclusions.

References:

- please unify formatting (e.g. not all journal titles are italic/cursive)
- P21L5: This references misses journal etc.
- P21L18: what does "p.20-1" mean?
- P21L29-30: are 175 and 83 page numbers?
- P21L30: are 149–56 page numbers?

—————————————————

---

## Short Comment (SC1) · 2 May 2018

This is a good set of experiments. I encourage the authors to take some time to improve their manuscript. Here some comments that can be useful.

LINE 4 PAGE 4. Please note that this RE value omits the rotational kinetic energy and as such, it simplifies the description of the collisions. I do expect that your spherical polyhedrons rotated both before and after their impacts. This should be mentioned in the discussion since it affects the plot in Fig 8. For example, in our experiments (Cagnoli and Manga, 2003), our cylindrical particles did have a rotational kinetic energy but only after the collision with the target as the high-speed video camera confirmed.

LINE 18 PAGE 5. I think that a drawing of the apparatus with vertical and horizontal length scales would improve the readability of the paper.

FIG 8 PAG 11. Here, it seems to me that you felt the obligation to have to find one single best-fit curve even if your data points illustrate a much more complex situation. Rather than concave-down best-fit curves (which are truly not convincing), this plot shows two features: 1) the maximum values decrease as the impact angle increases and 2) the spread of the data points decreases as the impact angle increases. This is true for both your grain sizes. We obtained these same features as shown by Fig 4A in Cagnoli and Manga (2003). I strongly suggest to remove these concave-down curves because they are truly misleading.

FIG 9 PAG 11. It would be useful to identify in this figure each experiment with its own characteristics.

TABLE 2 PAG 13. Please note that our cylinders are 0.89 cm long and with a basal diameter equal to 0.55 cm (Cagnoli and Manga, 2003). However, rebound angles of larger cylinders are also shown in Fig. 2A.

LINE 9 PAG 13. The rebound angles can be larger than the impact angles for two reasons. First, the surface of your concrete slabs cannot be perfectly flat in particular after the target has been damaged by previous impacts. Second, the surface of your particles has a curvature that varies from place to place (i.e., they have edges and corners). In other words, the true impact angle is not known. In our Fig 2A, some rebound angles are also larger than the impact angles. Even if this seems to be a flaw of the experiments, it has to be accepted as the inevitable complexity of rock fragment collisions and it is still useful to understand this complexity. For this reason, it is not correct to exclude what you call "non-ideal data points" when computing best-fit curves. The truth is that a single best-fit curve of the entire set of data points in Fig 8 does not exist. You can plot only a trend line for the maximum values if you really want to.

FIG 11 PAG 14. Please, remove curves 5 from Figs 11a, 11b and 11c, because, in

nature, beta can be larger than alpha. Do Figs 11a and 11b display mean values? If yes, state this clearly. In Figure 11c, draw only curves showing the maximum RE values.

LINE 3 PAG 15. What you say here is true. However, I would rephrase the sentences. The small Rn values in Cagnoli and Manga (2003) are due to the weak strength of pumice whose damage upon impact dissipates energy.

LINE 25 PAG 15. What do you mean with "nadir"? Please find a more appropriate word.

LINE 30 PAG 15. As explained above, curve 1 in Fig 11c is not useful and should be removed from the plot.

LINE 8 PAGE 16. This is not correct. Both your Fig 7 and our Fig 3B confirm that Rt increases as the impact angle increases. The problem is that the data spread is large. But this is due also to irregularity on the surfaces of target and particles, for example.

LINE 21 PAGE 16. This is the same explanation we have provided in our paper (see our Fig 1), but no credit is given.

LINE 18 PAGE 18. The use of the coefficient of restitution does not provide a good description of rock fragment collisions. But credit should be given to who has already said it (e.g., Stronge, 1991). Both your and our data sets show that: 1) there is no such as thing as a single value of the coefficient of restitution, and 2) also the more informative ratio of the kinetic energy is not a constant.

Bruno Cagnoli - Istituto Nazionale di Geofisica e Vulcanologia

References

Cagnoli and Manga, 2003. Pumice-pumice collisions and the effect of the impact angle. Geophys. Res. Lett., 30(12) 1636. doi:10.1029/2003GL017421

Stronge 1991. Friction in collisions: Resolution of a paradox. J. App. Phys. 69, 610-

612.

---

## Referee Comment (RC2) · M. Farin (Referee) · 19 May 2018

The paper discusses the effect of the angle of impact with respect to the slope on the coefficient of restitution of rockfalls in the normal and tangential directions to the slope and the loss of kinetic energy during the impact. The discussion is based on a set of close to natural size rockfall experiments with limestones blocks dropped in free fall on a concrete slab inclined at various angles. The authors observe that the normal coefficient of restitution decreases and the kinetic energy dissipation increases as the impact angle increases, which seems in agreement with previous impact tests experiments with various material, impactor size (for smaller size than in this study)

and various impact speeds. The authors however state that one should not deduce that a high normal coefficient of restitution necessarily means a high kinetic energy coefficient of restitution because no correlation is observed between the to coefficients. Indeed, the kinetic energy coefficient of restitution increases has the normal coefficient of restitution Rn for small values of Rn (i.e., for large angles of impact) but decreases as Rn increases for small angles of impact, when Rn > 1. They interpret that the large Rn coefficients > 1 are caused by basal roughness that impeach the impactor to rebound with a small angle of rebound when there are basal obstacles.

The paper is globally well written and the figures are clear. The experimental setup and results are clearly presented and seem of good quality. However, I think the paper suffer from several problems and should be rewritten and improved. I detailed here my main remarks and more specific comments that refer to line numbers below.

General Comments

- Most of the results given in the paper, in particular the variation of the coefficients of restitution as a function of the impact angle, were already reported in previous studies. It is not clear what this paper brings new to the research on energy losses during impacts. Please state clearly in the introduction what are the main questions that are posed at the end of the previous studies and needed additional experiments and answer to these specific questions in the conclusions. It is not clear what people doing computer simulations of rockfalls should retain from this work and how they could use the presented results.

- I think that one important parameter that could allow us to better understand why kinetic energy losses are larger at high impact angles is the energy lost in rotational modes of the impactor. The more energy is dissipated in rotation after the impact, the less energy is restituted to the block as kinetic energy for rebound (cf Farin et al. (2015) Characterization of rockfalls from seismic signal: Insights from laboratory experiments, JGR:Earth Surface, Figure C1b). The authors could take advantage of the fact that their

experimental setup has 8 cameras around the impact to measure precisely the rotation of the impactors before and after the impact and evaluate the rotational energy. This energy could be defined as $1/2 * I * omega\_r^2$, where I is the moment of inertia of the block (that could be approximated to a full sphere) and omega_r is its rotation speed. A figure showing the kinetic coefficient of restitution, Re, as a function of the rotational energy after impact could be interesting to show to bring additional contribution with respect to the previous work on the subject. Also, it is important to precise in the paper that the 'energy coefficient of restitution' is the 'kinetic energy coefficient of restitution', which does not represent the whole energy lost by the block but only the kinetic energy Ek lost. If a lot of energy is transmitted in rotation energy Er maybe the total energy of the block Ek + Er does not decreases at large impact angles (?).

- The authors suspect at several times in the paper that the impact speed has an influence on the coefficients of restitution. Thus, they should produce a Figure showing the coefficients of restitutions (and the rebound angle) as a function of the impact speed (event if only 3 different impacts speeds are investigated here, they could also use the data from previous work). Such a figure could support their discussion.

- I find that the discussion section is a bit difficult to follow. Maybe it could be reworked with subsections, discussing for example 'Interpretation of normal coefficient of restitution larger than 1', 'Relation between kinetic energy losses and normal coefficient of restitution'. . .

Specific comments: Abstract : - l14: the impact angle 'with respect to the slope' page 2, L2: define the coefficient of restitution

Introduction: - l 26: 'the similitude requirements. . . cannot be easily matched': I do not understand this sentence. Please rewrite.

- page 2 L32: define the energy coefficient of restitution. 'The kinetic coefficient of restitution' is more appropriate.

- page 2 l.34: Please do not give the same results as that given in the abstract. Please raise the general questions that require you to conduct additional experiments and that you answer in this paper, and answer these specific questions in the conclusion section. Sections 1.2 and section 2 should be merged with 1.Introduction and this whole section should lead to the problematic of the paper: what new contribution are you bringing to this research subject? To what questions are you answering?

- Page 3, L.15: n_cor and t_cor are never used in the following of the paper thus they should not be introduced.

- Page 3, L.20: it could be also interesting to present the results for Rv as a function of the impact angle and the kinetic energy lost because lots of people are using this definition. Is it varying differently than Rn with the impact angle ?

- Page 4, l.2: ratio of kinetic energies

- Page 4, l.26: 'the impact angle can influence the rebound angle': be more precise. Does the rebound angle increase or decrease when impact angle increases ?

- Page 4, l.29: 'the kinematic coefficient of restitution Rv was more appropriate than the normal COR for use in correlations with the impact angles'. The relation between Rv and the impact angle should be also represented in this paper to check whether this statement is also true with the present experiments.

- Page 5, l.3-4: These are poor sentences to sum up the previous results and motivate your work. Please clearly state at the end of the introduction what is missing from the previous work and requires you to do additional experiments.

- Page 5, l.14: what is the 'rebound hardness value' ? Does not it have units ? I think it could be more useful to give the Poisson's ratios and Young's modulii of the materials composing the impactors and the slabs. For example, people may want to use your data to compute impact forces (for example using Hertz's impact model) and compare the impact forces to the coefficients of restitution and impact angles and such

computations require the Poisson's ratios and Young's modulii.

- Page 7, L.12: define the acronym COR beforehand

- Page 8, L.9: the sentence 'The values of Rn ...' is unnecessary, one can read the values on the figure.

- Page 9, equations (4-1), (4-2) and other equations in the paper: only two significant digits are sufficient because the standard deviation on the data is at least 0.01. For example, equation (4-1) should rather be: $Rn = 4.41\ a\hat{}(-0.6)$, $R\hat{}2 = 0.81$

- Fig 6: The coefficient of restitution does not seem to depend on diameter, except 2 data points of higher value for D=10cm at low impact angle. In fact, the theory says that the coefficient of restitution should not depend on impactor size for impacts on a thick block (when the thickness of the impacted slab is large compared to the size of the impactor) and that the COR decreases as the impactor size increases when the impact is on a substrate whose thickness is small compared to the impactor size (cf Farin et al. (2015) Characterization of rockfalls from seismic signal: Insights from laboratory experiments, JGR:Earth Surface). The slab you are using could be considered as thin compared to the impactor size but because the slabs seem to be a bit buried in ground, they may be considered as thick substrates, thus the coefficient of restitution does not depend on the impactor size. A comment on this could be interesting to explain the fact that the measured COR is independent of the impactor size in your experiments.

- All Figures in general: Please use a larger and sans-serif font to improve figures readability.

- Figures 6, 7 and 8: I would use the same kind of scaling law (power law) for the 3 coefficients of restitution to compare them. A 2nd order polynomial law for figure 8 makes no sense because (1) you could fit everything why it and (2) you change your mind after that and use a linear law in figure 5c because it compares better with the previous results.

- Please merge some of the figures together (e.g Fig. 2,3,4; Fig. 6,7,8; Fig. 12,13. . .)

- Page 9, L.14: the sentence 'The values of Rt . . .' is unnecessary, one can read the values on the figure.

- Page 10, L.5: the sentence 'The values of Re . . .' is unnecessary, one can read the values on the figure.

- Page 10, L.16: Have you measured the depth of erosion created by the impacts? Maybe the largest impactor have caused more erosion of the slabs and thus lose more energy in deformation of the slab than the smallest impactors. A figure showing the energy lost as a function of the depth of erosion due to the impact could be interesting if you can do it.

- Page 12, L.3: 'The data points are stably located above the 45° line until the impact angle reaches 36°' may be a clearer sentence.

- The '45° line' is misleading because the compared variables are angles. The 'equality line' or 'y = x line' are other possibilities.

- Page 12, l.5: 'the kinetic energy loss constituted 50-75% of the total kinetic energy' This is false: total energy also includes the rotation energy.

- Page 12, l.6: 'the energy loss level can not be assessed by comparing the rebound and impact angle': not clear

- Page 12, l.18: Maybe you should directly compare your results with that of previous studies before drawing conclusions because your conclusions seem to change a bit after the comparison with the other studies (for example you say later than Rt does not depend on the impact angle and you change the scaling law for Re), thus sections 4.1, 4.2 and 5 are redundant and confusing.

- Page 12, L.23 to Page 13 L.2: This should be in the introduction.

- Page 13, L.9: what is the 'ideal state'? If you observe rebound angles larger than 1.2

times the impact angle, there is a chance that we can also observe this in nature. You should not exclude data points just because they do not compare well to the previous work. On contrary, you should keep these points and interpret why you observe such a situation in your experiments and why it is not observed in the previous work.

- Table 2 and Fig. 11: please replace 'Wang 2018' by 'this study' to avoid confusion.

- Page 15, L.3: 'The minimum Rn occurred... erosion and particle breakage'. This explanation that stronger kinetic energy dissipation due to erosion may explain the lower Rn value for Cagnoli's experiment does not work because (1) you also observe erosion by the impacts and the Rn in your experiments are larger and (2) you state later that the normal coefficient of restitution does not correlate with kinetic energy loss...

- Page 15, L.9-12: the exact scaling law that describe best the data is not very important given the large scattering in the data. What matters more is if you can explain the general trend. Also, if you give a scaling law for you data, you should also try to fit the data of the previous work with the same kind of scaling law. If the scaling law works for your data and not with the other work, its usefulness is very limited...

- Page 15, L.15: The variation of the kinetic energy COR with impact angle may be better understood if you also show the rotation energy (more energy dissipated in rotation means less energy restituted in kinetic energy for the rebound). You should not remove data points just because they do not compare well with previous work. Explain the difference otherwise the same conclusions could have been drawn by just comparing the previous work together and this present work contribution is limited.

- Page 10, l.16: 'The impact velocity is an important... resulting coefficients of restitution': please show a figure of the CORs as a function of the impact speed (even including the previous work data) to support your conclusion.

- Discussion section. Different things are discussed here, please add subsections to make the discussion clearer.

- Page 16, L. 24: I do not understand what you mean by 'with a parallel motion'

- Page 16, L. 27: 'Therefore,Ăă. . .' I do not understand the logical link with the previous sentence. If rotation speed has an important effect on rebound angle and coefficient of restitution, you should show it on Figures.

- Page 17: I understand that basal roughness can lead to higher angles of rebound, but in this case, the impactors on intact slabs should have in average lower angles of rebound than impactors on eroded slabs. Can you draw a figure or give the average rebound angles on intact vs eroded slabs to support your discussion ? If you measured the depth of erosion on the slabs, maybe the rebound angle could be correlated to with erosion depth (?).

- Page 20, l. 5: This conclusion does not bring anything new to the research. I believe you could draw much more results from you experimental data.

Please also note the supplement to this comment:
https://www.nat-hazards-earth-syst-sci-discuss.net/nhess-2018-108/nhess-2018-108-RC2-supplement.pdf
* * *
Review of Effects of the impact angle on the coefficient of restitution based on a medium-scale laboratory test.
Wang et al. 2018

The paper discusses the effect of the angle of impact with respect to the slope on the coefficient of restitution of rockfalls in the normal and tangential directions to the slope and the loss of kinetic energy during the impact. The discussion is based on a set of close to natural size rockfall experiments with limestones blocks dropped in free fall on a concrete slab inclined at various angles.

The authors observe that the normal coefficient of restitution decreases and the kinetic energy dissipation increases as the impact angle increases, which seems in agreement with previous impact tests experiments with various material, impactor size (for smaller size than in this study) and various impact speeds.

The authors however state that one should not deduce that a high normal coefficient of restitution necessarily means a high kinetic energy coefficient of restitution because no correlation is observed between the to coefficients. Indeed, the kinetic energy coefficient of restitution increases has the normal coefficient of restitution Rn for small values of Rn (i.e., for large angles of impact) but decreases as Rn increases for small angles of impact, when Rn > 1.

They interpret the large Rn coefficients > 1 to be caused by basal roughness that avoid the impactor to rebound with a small angle of rebound when there are basal obstacles.

The paper is globally well written and the figures are clear. The experimental setup and results are clearly presented and seem of good quality. However, I think the paper suffer from several problems and should be rewritten and improved. I detailed here my main remarks and more specific comments that refer to line numbers below.

General Comments

- Given that most of the results given in the paper, in particular the variation of the coefficients of restitution as a function of the impact angle were already reported in previous studies, it is not clear what this paper brings new to the research on energy losses during impacts. Please state clearly in the introduction what are the main questions that are posed at the end of the previous studies and needed additional experiments and answer to these specific questions in the conclusion. It is not clear what people doing computer simulations of rockfalls should retain from this work and how they could use the presented results.
- I think that one important parameter that could allow us to better understand why kinetic energy losses are larger at high impact angles is the energy lost in rotational modes of the impactor. The more energy is dissipated in rotation after the impact, the less energy is restituted to the block as kinetic energy for rebound (cf Farin et al. (2015) Characterization of rockfalls from seismic signal: Insights from laboratory experiments, JGR:Earth Surface, Figure C1 b). The authors could take advantage of the fact that their experimental setup has 8 cameras around the impact to measure precisely the rotation of the impactor before and after the impact and evaluate the rotational energy. This energy could be defined as $1/2 * I * \omega_r^2$, where I is the moment of inertia of the block and omega_r is its rotation speed. A figure showing the kinetic coefficient of restitution as a function of the rotational energy after impact could be interesting to show to bring additional contribution with respect to the previous work on the subject. Also, its is important to precise in the paper that the « energy coefficient of restitution » is the « kinetic energy coefficient of restitution », which does not represent the whole energy lost by the block but only the kinetic energy Ek lost. If a lot of energy is transmitted in rotational mode energy Er maybe the total energy of the block Ek + Er does not decreases at large impact angles (?).
- The authors suspect at several times in the paper that the impact speed has an influence on the coefficients of restitution. Thus, they should produce a Figure showing the coefficients of restitutions as a function of the impact speed (event if only 3 different impacts speeds are investigated here). Such a figure could support their discussion.

**Fig. 1.**

**Supplement:**

[Figure]

**Journal of Geophysical Research: Solid Earth**

[Figure]

**RESEARCH ARTICLE**

10.1002/2015JB012331

**Key Points:**
- Analytical scaling laws relating rockfalls to signal characteristics are derived from impact models
- The laws are tested for impacts experiments of beads and gravels on various material and geometries
- An energy budget of the impacts is established among radiated elastic energy and inelastic losses

**Correspondence to:**
M. Farin,
farin@ipgp.fr

**Citation:**
Farin, M., A. Mangeney, R. Toussaint, J. de Rosny, N. Shapiro, T. Dewez, C. Hibert, C. Mathon, O. Sedan, and F. Berger (2015), Characterization of rockfalls from seismic signal: Insights from laboratory experiments, *J. Geophys. Res. Solid Earth*, *120*, doi:10.1002/2015JB012331.

Received 4 JUL 2015
Accepted 24 SEP 2015
Accepted article online 29 SEP 2015

**Characterization of rockfalls from seismic signal: Insights from laboratory experiments**

**Maxime Farin[1], Anne Mangeney[1,2], Renaud Toussaint[3], Julien de Rosny[4], Nikolai Shapiro[1], Thomas Dewez[5], Clément Hibert[6], Christian Mathon[5], Olivier Sedan[5], and Frédéric Berger[7]**

[1]Institut de Physique du Globe de Paris, Sorbonne Paris Cité, CNRS (UMR 7154), Paris, France, [2]ANGE team, CEREMA, Inria, Lab. J.-L. Lions, CNRS, Paris, France, [3]IPGS-EOST, Géophysique Expérimentale, CNRS, Strasbourg, France, [4]Institut Langevin, Laboratoire Ondes et Acoustique, CNRS, Paris, France, [5]Service des Risques Naturels, BRGM, France, [6]Lamont-Doherty Earth Observatory, Palisades, New York, USA, [7]Unité de Recherche Ecosystèmes Montagnards, Cemagref, Grenoble, France

**Abstract** The seismic signals generated by rockfalls can provide information on their dynamics and location. However, the lack of field observations makes it difficult to establish clear relationships between the characteristics of the signal and the source. In this study, scaling laws are derived from analytical impact models to relate the mass and the speed of an individual impactor to the radiated elastic energy and the frequency content of the emitted seismic signal. It appears that the radiated elastic energy and frequencies decrease when the impact is viscoelastic or elastoplastic compared to the case of an elastic impact. The scaling laws are validated with laboratory experiments of impacts of beads and gravels on smooth thin plates and rough thick blocks. Regardless of the involved materials, the masses and speeds of the impactors are retrieved from seismic measurements within a factor of 3. A quantitative energy budget of the impacts is established. On smooth thin plates, the lost energy is either radiated in elastic waves or dissipated in viscoelasticity when the impactor is large or small with respect to the plate thickness, respectively. In contrast, on rough thick blocks, the elastic energy radiation represents less than 5% of the lost energy. Most of the energy is lost in plastic deformation or rotation modes of the bead owing to surface roughness. Finally, we estimate the elastic energy radiated during field scale rockfalls experiments. This energy is shown to be proportional to the boulder mass, in agreement with the theoretical scaling laws.

**1. Introduction**

Rockfalls represent a major natural hazard in steep landscapes. Because of their unpredictable and spontaneous nature, the seismic monitoring of these gravitational instabilities has raised a growing interest for risks assessment in the last decades. Recent studies showed that rockfalls can be automatically detected and localized with high precision from the seismic signal they generate [*Suriñach et al.*, 2005; *Deparis et al.*, 2008; *Dammeier et al.*, 2011; *Hibert et al.*, 2011, 2014a]. A burning challenge is to obtain quantitative information on the gravitational event (volume, propagation velocity, extension, etc.) from the characteristics of the associated seismic signal [*Norris*, 1994; *Deparis et al.*, 2008; *Vilajosana et al.*, 2008; *Favreau et al.*, 2010; *Dammeier et al.*, 2011; *Hibert et al.*, 2011, 2014a; *Moretti et al.*, 2012, 2015; *Yamada et al.*, 2012].

Some authors found empirical relationships between the rockfall volume and the maximum amplitude of the signal or the radiated seismic energy [*Norris*, 1994; *Hibert et al.*, 2011; *Yamada et al.*, 2012]. The precursory work of *Norris* [1994] on rockfalls of large volume > $10^4$ m$^3$ at Mount St. Helens showed that the maximum amplitude of the emitted signal depends linearly on the rockfall volume. This is in agreement with the observations of *Yamada et al.* [2012] on landslides triggered in Japan by Typhoon Talas in 2011. The authors observed that the integral of the squared signal amplitude measured at 1 km from the source varied as the square the landslide volume. In contrast, *Hibert et al.* [2011] showed that the seismic energy emitted by rockfalls is proportional to their volume in the Dolomieu crater of the Piton de la Fournaise volcano, Réunion Island. Moreover, *Dammeier et al.* [2011] used a statistical approach and estimated the volume $V$ of several rockfalls in the central Alps from the measurement of the duration $t_s$, envelope area EA, and peak amplitude PA of the generated seismic signal. For 20 well-constrained events, they found the empirical scaling law: $V \propto t_s^{1.0368}\text{EA}^{-0.1248}\text{PA}^{1.1446}$. The volumes estimated with this relation were close to the measured ones, but the results were sensitive to the distance of the seismic stations from the events.

Other surveys investigated the ratio of the radiated seismic energy $W_{el}$ over the potential energy $\Delta E_p$ lost by the rockfalls from initiation to deposition [*Deparis et al.*, 2008; *Hibert et al.*, 2011, 2014a; *Lévy et al.*, 2015]. *Deparis et al.* [2008] studied 10 rockfalls that occurred between 1992 and 2001 in the French Alps and estimated that the ratio $W_{el}/\Delta E_p$ was between $10^{-5}$ and $10^{-3}$. *Hibert et al.* [2011, 2014a] observed that the ratios of the seismic energy $W_{el}$ radiated by the rockfalls in the Dolomieu crater over their potential energy lost $\Delta E_p$ varied from $5.10^{-5}$ to $2.10^{-3}$. Finally, *Lévy et al.* [2015] found $W_{el}/\Delta E_p \approx 1.1.10^{-5} - 2.8.10^{-5}$ for pyroclastic and debris flows that occurred on the Souffrière Hills volcano in Montserrat Island, Lesser Antilles. Most of the aforementioned studies focused on a specific rockfall site [*Norris*, 1994; *Deparis et al.*, 2008; *Dammeier et al.*, 2011; *Hibert et al.*, 2011, 2014a; *Yamada et al.*, 2012; *Lévy et al.*, 2015]. It is, however, difficult to test the developed techniques on other sites because only a few of rockfall areas are nowadays simultaneously seismically and optically monitored.

Because gravitational events are very complex, it is still not clear what parameters controls their seismic emission. The seismic signals generated by rockfalls on the field are partially composed of waves emitted by individual impacts of boulders, triggering high-frequency noise, typically higher than 1 Hz [e.g., *Deparis et al.*, 2008; *Vilajosana et al.*, 2008; *Helmstetter and Garambois*, 2010; *Hibert et al.*, 2014b; *Lévy et al.*, 2015] and by long-period stresses variations owing to the mass acceleration and deceleration over the topography, responsible for lower frequencies in the signal (<1 Hz) [e.g., *Kanamori and Given*, 1982; *Favreau et al.*, 2010; *Allstadt*, 2013]. To start the work on understanding the seismic emission of rockfalls, we focus here on the seismic signal generated by impacts.

The dynamics of impact can be described at first order by the classical model proposed by *Hertz* [1882] that gives the analytical expression of the force of impact of an elastic sphere on a solid elastic surface [see *Johnson*, 1985]. From the comparison of the impact forces and durations measured from the emitted seismic signal with that predicted by *Hertz* [1882], *Buttle and Scruby* [1990] and *Buttle et al.* [1991] managed to retrieve the diameter of submillimetrical particles impacting a thick block. However, their computation was based on the direct compressive wave, measured at the opposite of the impact on the target block. Their configuration can therefore not be exported to field context. Also based on *Hertz*'s [1882] theory, *Tsai et al.* [2012] expressed the long-period power spectral density generated by the impacts of sediments on the bed of rivers as a function of the river parameters such the particle size distribution, the impact rate, and the bed load flux. From seismic measurements of *Burtin et al.* [2008] on trans-Himalayan Trisuli River, *Tsai et al.* [2012] were then able to quantitatively deduce the bed load flux.

In this paper, we adopt a similar approach. The basic idea is to derive from *Hertz*'s [1882] model analytical scaling laws relating the radiated elastic energy and the frequencies of the seismic signal generated by an impact to the mass and the speed of the impactor. These laws can then be inverted to deduce the impact parameters from a measurement of the emitted seismic signal. Note that *Tsai et al.* [2012] assumed for their analytical model that the impact duration was instantaneous because they focused on signals of long periods compared with this duration. On the contrary, we do not assume an instantaneous impact here because we try to use the whole spectrum content. Indeed, in order to robustly estimate the impact parameters from the emitted signal using our scaling laws, we need to determine the absolute energy radiated in elastic waves and, therefore, the entire amplitude spectrum of the seismic signal generated by the impact. This implies (1) to record signals with sampling periods much smaller than the impact duration and (2) to know well the elastic properties of the impactor and of the substrate, i.e., their elastic modulii, their density, the type of mode excited in the substrate after an impact, its dispersion, and how its energy attenuates with increasing distance from the source.

These two conditions are not easy to address in the field because usual sampling times are of the order of the typical impact durations (∼0.01 s) and because of the strong heterogeneity of the ground. Therefore, in order to test our analytical scaling laws, we perform controlled laboratory experiments of impacts of spherical beads on thin plates with an ideal smooth surface, then on rough thick blocks, i.e., in a context similar to that of the field. A series of impact experiments is also conducted with gravels to quantify how the relations between impact properties and signal characteristics change when the impactor has a rough surface, which is a more realistic case, i.e., closer to what is observed for natural rockfalls.

During an impact, a significant part of the impactor's energy can be lost in inelastic processes such as plastic, i.e., irreversible, deformation of the impactor or the ground [*Davies*, 1949] or viscoelastic dissipation in the vicinity of the impact [*Falcon et al.*, 1998]. These losses are not considered in *Hertz*'s [1882] elastic impact

model. In this paper, we use analytical models of viscoelastic and elastoplastic impact to estimate how the frequencies of the emitted vibration and the radiated elastic energy deviate from that predicted using *Hertz*'s [1882] theory when inelastic dissipation occurs. Using these models, we interpret the discrepancy observed between the measured values in our experiments and those predicted by the elastic model of *Hertz* [1882]. Another advantage of the laboratory experiments is that the total energy lost during the impact can be easily measured from the velocity change of the impactor before and after the impact. We can then establish a quantitative energy budget among the energy radiated in elastic waves and that dissipated in inelastic processes. This allow us to better understand the process of wave generation by an impact and to roughly extrapolate what should be the relative importance of the different loss processes for natural rockfalls.

This paper is structured as follows. In section 2, we recall the theory for elastic, viscoelastic, and elastoplastic impacts of a sphere on a plane surface and we derive the analytical scaling laws from this theory. The experimental setup is presented in section 3. In section 4, we test experimentally the scaling laws established in section 2 and retrieve the masses and speeds of the impactors from the measured seismic signals. In addition, we establish the energy budget of the impacts among elastic and inelastic losses and observe how this budget varies on smooth thin plates and rough thick blocks when the bead mass and the elastic parameters change. In section 5, the discrepancy of the experimental results with the theory is discussed. Finally, the analytical scaling laws demonstrated in this paper are compared with empirical relations observed in drop experiments of large boulders in a natural context. We identify the issues that should be overcome in order to apply our scaling laws to natural impact situations.

**2. Theory: Relations Between Impact Parameters and Seismic Characteristics**

The vibration displacement $\mathbf{u}(\mathbf{r}, t)$ at the distance $\mathbf{r}$ from an impact is given by the time convolution of the force $\mathbf{F}(\mathbf{r}_s, t)$ applied to the ground at position $r_s$ with Green's function $\bar{\bar{\mathbf{G}}}(\mathbf{r}, \mathbf{r}_s, t)$ of the structure where the wave propagates [*Aki and Richards*, 1980]:

$$\mathbf{u}(\mathbf{r}, t) = \bar{\bar{\mathbf{G}}}(\mathbf{r}, \mathbf{r}_s, t) * \mathbf{F}(\mathbf{r}_s, t), \tag{1}$$

where $*$ stands for the time convolution product. In our experiments, we only have access to the vibration acceleration in the direction normal to the surface $a_z(r, t)$. In the time Fourier domain, this acceleration is given by

$$\tilde{A}_z(r, f) = -(2\pi f)^2 \tilde{G}_{zz}(r, f) \tilde{F}_z(f), \tag{2}$$

where $f$ is the frequency and $\tilde{F}_z(f)$ is the time Fourier transform of the vertical impact force $F_z(t)$. The expression of Green's function $\tilde{G}_{zz}(r, f)$ is different when the impact duration is greater or smaller than the two-way travel time of the emitted wave in the structure thickness, i.e., for impacts on thin plates and on thick blocks, respectively. A plate of thickness $h$ vibrates normally to its surface because the fundamental $A_0$ mode of Lamb carries most of the energy [*Royer and Dieulesaint*, 2000; *Farin et al.*, 2015]. The module of Green's function of this mode of vibration can be approximated by [e.g., *Goyder and White*, 1980]:

$$|\tilde{G}_{zz}(r, f)| = \frac{1}{8Bk^2} \sqrt{\frac{2}{\pi kr}}, \tag{3}$$

where $k$ is the wave number, $B = h^3 E_p / 12(1 - v_p^2)$ is the bending stiffness, and $E_p$ and $v_p$ are the Young's modulus and the Poisson ratio of the impacted structure, respectively. At low frequencies, i.e., for $kh << 1$, the wave number $k$ is related to the angular frequency $\omega$ by $k^4 = \omega^2 \rho_p h / B$, where $\rho_p$ is the plate density.

In contrast, an impact on a thick block generates compressive, shear, and Rayleigh waves [*Miller and Pursey*, 1955; *Aki and Richards*, 1980]. For $kr >> 1$, i.e., in far field, the displacement mainly results from Rayleigh waves and Green's function can be approximated by [*Miller and Pursey*, 1955; *Farin et al.*, 2015]

$$|\tilde{G}_{zz}(r, f)| \approx \frac{\xi^2 \omega}{2\mu c_P} \frac{\sqrt{x_0(x_0^2 - 1)}}{f_0'(x_0)} \sqrt{\frac{2c_P}{\pi \omega r}}, \tag{4}$$

[Figure]

**Figure 1.** (a) Schematic showing the penetration depth $\delta_z$ of a sphere of radius $R$ on a plane surface during an impact. Geometrically, the surface of contact is a circle of radius $a$. Normalized force of impact $F_z(t/t_0)/F_0$ for (b) different values of the viscoelastic parameter $\alpha$ (or $\lambda_Z$ for *Zener*'s [1941] theory; see section 2.1.1.2) and for (c) different values of the stresses ratio $P_Y/P_0$. $F_0$ and $t_0 = T_c/2$ are, respectively, the force and the time at maximal compression during an elastic impact, i.e., for $\alpha = 0$ and $P_Y/P_0 = 1$.

where $\mu$ is the shear Lamé coefficient, $c_P$ is the compressional wave speed, $\xi = \sqrt{2(1 - \nu_p)/(1 - 2\nu_p)}$, $f_0(x) = (2x^2 - \xi^2)^2 - 4x^2\sqrt{(x^2 - 1)(x^2 - \xi^2)}$, and $x_0$ is the real root of $f_0$.

In this section, we derive analytical scaling laws that relate the energy radiated in elastic waves and the characteristic frequencies of the vibration $\tilde{A}_z(r, f)$ emitted by an impact to the impact parameters (mass $m$ and speed $V_z$). Because the vibration $\tilde{A}_z(r, f)$ is controlled by the impact force $\tilde{F}_z(f)$ (equation (2)), the scaling laws are different when the impact is elastic or when viscoelastic dissipation or plastic deformation occur. Let us first recall the expression of the impact force for an elastic impact and how it changes for an inelastic impact. Note that we do not use any elastoviscoplastic model of impact here because elastic energy radiation, viscoelastic dissipation, and plastic deformation are never simultaneously significant in our experiments, even though it could be the case on the field. For example, in certain cases, viscoelastic and plastic losses are negligible and an elastic impact model is sufficient to describe the energy transfer.

**2.1. Impact Models**

**2.1.1. Elastic Impact Model**

**2.1.1.1. Hertz's Model**

*Hertz* [1882] gives the force of elastic contact of a sphere of mass $m$ on a plane as a function of their interpenetration depth $\delta_z(t)$ (Figure 1a):

$$F_z(t) = -K\delta_z^{3/2}(t), \tag{5}$$

where

$$K = \frac{4}{3}R^{1/2}E^*, \tag{6}$$

with $R$, the sphere radius and $1/E^* = (1 - \nu_s^2)/E_s + (1 - \nu_p^2)/E_p$, where $\nu_s$, $\nu_p$, $E_s$, and $E_p$ are, respectively, the Poisson's ratios and the Young's moduli of the constitutive materials of the sphere and the impacted plane.

During an impact, the displacement of the center of mass of the sphere is equal to the interpenetration $\delta_z(t)$. Neglecting the gravity force, the equation of motion of the sphere is then

$$m\frac{\mathrm{d}^2\delta_z(t)}{\mathrm{d}t^2} = -K\delta_z^{3/2}(t). \tag{7}$$

The solution of equation (7) is of the form $\delta_z(t) = \delta_{z0}f(t/T_c)$. The maximum interpenetration depth $\delta_{z0}$ and the impact duration $T_c$ are, respectively, given by [*Johnson*, 1985]

$$\delta_{z0} = \left(\frac{5mV_z^2}{4K}\right)^{2/5}, \tag{8}$$

and

$$T_c \approx 2.94\frac{\delta_{z0}}{V_z} \approx 2.87\left(\frac{16m^2}{9K^2V_z}\right)^{1/5}, \tag{9}$$

where $V_z$ is the impact speed.

The maximum value of the impact force is therefore, according to equation (5)

$$F_0 = K\delta_{z0}^{3/2} = K\left(\frac{5mV_z^2}{4K}\right)^{3/5}. \tag{10}$$

In the following, the interpenetration depth $\delta_z(t)$, the time $t$ and the force $F_z(t)$ are, respectively, scaled by $\delta_{z0}$, $\delta_{z0}/V_z$, and $F_0$, which contain all the information on the impact characteristics.

**2.1.1.2. Hertz-Zener's Model for Impacts on Thin Plates**

*Hertz*'s [1882] model (equation (8)) is valid provided that the energy radiated in elastic waves during the impact represents a small proportion of the impact energy $\frac{1}{2}mV_z^{1/2}$ [*Hunter*, 1957; *Johnson*, 1985]. This is not the case when the thickness of the impacted structure is around or lower than the diameter of the impactor, i.e., for impacts on thin plates and membranes [e.g., *Zener*, 1941; *Farin et al.*, 2015]. When the energy lost in plate vibration during the impact is not negligible, *Zener* [1941] proposed a more exact description than *Hertz*'s [1882] model of the interaction between the sphere and the plate's surface. One has to distinguish the sphere displacement $z$, given by

$$m\frac{d^2z(t)}{dt^2} = -F_z(t), \tag{11}$$

from the plate's surface displacement $u_z$ at the position of the impact, whose time derivative is

$$\frac{du_z(t)}{dt} = Y_{el}F_z(t), \tag{12}$$

where $Y_{el}$ is the real part of the time derivative of Green's function at the impact position $\Re\left(dG_{zz}(r_0, t)/dt\right)$, i.e., the radiation admittance. This function is given by [*Goyder and White*, 1980] for plates

$$Y_{el} = \frac{1}{8\sqrt{B\rho_p h}}, \tag{13}$$

with $B$, the bending stiffness, and $h$, the plate thickness. In these equations, the impact force $F_z(t)$ follows *Hertz*'s [1882] theory (equation (5)).

The difference of equation (11) and the derivative of equation (12) gives the following equation for the relative movement of the sphere and of the substrate, i.e., the interpenetration $\delta_z(t) = z(t) - u_z(t)$, in dimensionless form with $\delta^* = \delta_z/\delta_{z0}$ and $t^* = V_z t/\delta_{z0}$:

$$\frac{d^2\delta^*}{dt^{*2}} = -\frac{5}{4}\left(\delta^{*3/2} + \lambda_Z\frac{d\delta^*}{dt^*}\delta^{*1/2}\right), \tag{14}$$

with

$$\lambda_Z \approx 0.175\frac{E^{*2/5}}{\rho_s^{1/15}\sqrt{B\rho_p h}}m^{2/3}V_z^{1/5}. \tag{15}$$

In equation (14), we retrieve the impact model of *Hertz* [1882] (equation (7)) with a corrective term that depends on the parameter $\lambda_Z$. This corrective term becomes negligible when the thickness $h$ of the structure

is much larger than the diameter $d$ of the impactor because the parameter $\lambda_Z$ tends toward 0 [*Zener*, 1941]. Therefore, for impacts on elastic half-spaces, i.e., on thick blocks, the corrective term disappears and the model of *Zener* [1941] (equation (14)) matches with that of *Hertz* [1882] (equation (7)). As a consequence, this model is only relevant for impacts on thin plates.

Equation (14) is solved numerically for different values of $\lambda_Z$ with the initial conditions $\delta^*(0) = 0$ and $\frac{d\delta^*}{dt^*}(0) = 1$. The impact force $F_z(t)/F_0 = \delta^{*3/2}$ is shown in Figure 1b. When $\lambda_Z$ increases, i.e., when $m$ and $V_z$ increase, the force profile loses its symmetry with respect to its maximum, its amplitude decreases, and its duration increases. For an inelastic coefficient $\lambda_Z = 0.25$, the force is only slightly affected. Practically, $\lambda_Z$ is always smaller than 0.5 in our experiments.

**2.1.2. Viscoelastic Impact Model**

Viscous dissipation is related to the viscosities of the materials involved in the impact and can be described as a heat loss. Viscoelastic solids are often represented by a spring and a dashpot in parallel (Kelvin-Voigt model). *Hertz*'s [1882] theory has been extended to viscoelastic impacts, adding a force $F_{\text{diss}}(t)$ in equation (7) to model viscous dissipation [*Kuwabara and Kono*, 1987; *Falcon et al.*, 1998; *Ramírez et al.*, 1999]:

$$F_{\text{diss}}(t) = -\frac{3}{2}DK\frac{d\delta_z(t)}{dt}\delta_z^{1/2}(t), \tag{16}$$

with $D$, a characteristic time depending on the materials viscosities and elastic constants [*Hertzsch et al.*, 1995; *Brilliantov et al.*, 1996; *Ramírez et al.*, 1999]. The expression of $D$ is only given in the literature in case when the sphere and the plane have the same elastic parameters $E$ and $v$:

$$D = \frac{2}{3}\frac{\chi^2}{(\chi + 2\eta)}\frac{(1 - v^2)(1 - 2v)}{Ev^2}, \tag{17}$$

where $\chi$ and $\eta$ are the bulk and shear viscosities, respectively. We cannot measure these two last parameters in our experiments, and they are not tabulated in our frequencies range of interest; therefore, $D$ will be an adjustable parameter.

The dimensionless equation of motion for a viscoelastic impact is then

$$\frac{d^2\delta^*}{dt^{*2}} = -\frac{5}{4}\left(\delta^{*3/2} + \alpha\frac{d\delta^*}{dt^*}\delta^{*1/2}\right), \tag{18}$$

which is the same expression as for *Zener*'s [1941] model (equation (14)) but with a different parameter

$$\alpha = \frac{3}{2}D\frac{V_z}{\delta_{z0}} \simeq 1.4D\frac{E^{*2/5}}{\rho_s^{1/15}}\frac{V_z^{1/5}}{m^{1/3}}, \tag{19}$$

the viscoelastic parameter [*Ramírez et al.*, 1999]. For $\alpha = 0$ (i.e., $D = 0$), equation (18) matches with equation (7) for elastic impacts.

Because equations (14) and (18) are identical, when $\alpha$ increases, the force profile varies exactly the same way as when $\lambda_Z$ increases in *Zener*'s [1941] model (Figure 1b). However, note that the corrective terms to *Hertz*'s [1882] model in the viscoelastic and *Zener*'s [1941] models have a different physical origin. The viscoelastic corrective term is due to the fact that the impactor and the ground have an intrinsic viscosity [*Falcon et al.*, 1998]. This term is stronger when the mass $m$, or diameter $d$, of the sphere decreases (equation (19)). On the contrary, the corrective term of *Zener*'s [1941] model comes from the fact that a larger amount of the impactor's kinetic energy is transferred into plate vibration during the impact when the sphere's diameter $d$ is large compared to the plate thickness $h$ [*Zener*, 1941] (equation (15)). We can therefore assume that the viscoelastic and *Zener*'s [1941] impact models are never simultaneously effective.

**2.1.3. Elastoplastic Impact Model**

Plastic (i.e., not reversible) deformations result from irreversible structural modifications which occur when the pressure on the contact area $P(t) = F_z(t)/2\pi R\delta_z(t)$ reaches the dynamic yield strength $P_Y = 3Y_d$ of the material, where $Y_d$ is the dynamic yield stress of the softest material [*Crook*, 1952; *Johnson*, 1985]. Plastic deformation can be evidenced by the apparition of a crater at the impact position. The energy lost to create this crater modifies the shape of the impact force with respect to the case of an elastic or viscoelastic impact. A model was proposed by *Troccaz et al.* [2000] to describe the evolution of the impact force when the limit of elastic

behavior is exceeded. This model is based on the hypothesis that only the sphere or the structure deforms plastically. Such an impact is composed of three successive phases:

1. The impact is elastic while $P(t) < P_Y$ and the impact force $F(t)$ follows equation (5).
2. When $P(t) \geq P_Y$ the deformation is fully plastic and the force expression becomes $F_z(t) = -2\pi R P_Y \delta_z(t)$ until the force reaches a maximum $F_{max}$, which is smaller than the maximum value $F_0$ for an elastic impact.
3. The rebound is elastic with $F_z(t) = F_{max}\left((\delta_z(t) - \delta_r)/(\delta_{max} - \delta_r)\right)^{3/2}$, where $\delta_{max}$ is the maximum interpenetration reached and $\delta_r$ is the residual deformation after plastic deformation, which is neglected (i.e., considered to be 0) in the following.

The dimensionless equation of motion during plastic deformation (phase 2) is then, if $\delta_z(t)$ and time $t$ are, respectively, scaled by $\delta_{z0}$ and $\delta_{z0}/V_z$,

$$\frac{d^2\delta^*}{dt^{*2}} = -\frac{5}{4}\frac{P_Y}{P_0}\delta^*,$$ (20)

where $P_0$ is the maximum stress during Hertz's elastic impact:

$$P_0 = \frac{K\delta_{z0}^{3/2}}{2\pi R\delta_{z0}} = \frac{2}{3\pi}\left(\frac{5}{4}\right)^{1/5}\rho_s^{1/5}E^{*4/5}V_z^{2/5}.$$ (21)

Equation (20) depends only on the stresses ratio $P_Y/P_0$ that is independent of the impactor mass $m$. When this ratio is greater or equal to 1, the impact is purely elastic. The amplitude of the impact force decreases as the stresses ratio $P_Y/P_0$ decreases (Figure 1c). Both the duration of the impact and the time to reach the maximum amplitude increase for an elastoplastic impact with respect to the elastic case.

**2.2. Analytical Scaling Laws**
The seismic signal generated by an impact can be characterized by the radiated elastic energy $W_{el}$ and by a frequency. Here we relate analytically these seismic characteristics with the mass $m$ and the speed $V_z$ of the impactor using the impact models presented above.

**2.2.1. Radiated Elastic Energy**
The energy $W_{el}$ radiated in elastic waves is the work done by the impact force $F_z(t)$ during the impact, i.e.,

$$W_{el} \hat{=} \int_{-\infty}^{+\infty} F_z(t)\frac{du_z(t)}{dt}dt = \int_{-\infty}^{+\infty} |\tilde{F}_z(f)|^2 \tilde{Y}_{el}(f)df,$$ (22)

according to Parceval's theorem, where $\frac{du_z(t)}{dt}$ is the vibration speed at the impact position (equation (12)) and $\tilde{Y}_{el}(f)$ is the time Fourier transform of the radiation admittance.

The radiated elastic energy $W_{el}$ is different for impacts on thin plates and on thick blocks because the radiation admittance $\tilde{Y}_{el}(f)$ has a different expression. Developing equation (22), we obtain in Table 1 analytical expressions for the elastic energy $W_{el}$ radiated during an impact on thin plates and thick blocks, as a function of the impact parameters (see Appendix A for details on the calculations). On thin plates,

$$W_{el} = a_1 C_{plate} m^{5/3} V_z^{11/5}$$ (23)

and on thick blocks,

$$W_{el} = a_2 C_{block} m V_z^{13/5},$$ (24)

where coefficients $a_1$ and $a_2$ depend only on the elastic parameters (see Table 1). In these expressions, $C_{plate} = \int_{-\infty}^{+\infty} |g(t^*)|^2 dt^*$ and $C_{block} = \int_0^{+\infty} f^{*2}|\tilde{g}(f^*)|^2 df^*$, where $|g(t^*)| = |F_z(t)|/F_0$ with $t^* = V_z t/\delta_{z0}$ and where $\tilde{g}(f^*)$ is the time Fourier transform of $g(t^*)$. For an elastic impact, i.e., with $F_z(t)$ given by equation (5), we obtain $C_{plate} \simeq 1.21$ and $C_{block} \simeq 0.02$. The function $g(t^*)$ has a lower amplitude when the impact is inelastic compared to the case of an elastic impact (Figures 1b and 1c). Therefore, both coefficients $C_{plate}$ and $C_{block}$ decrease when the viscoelastic parameter $\alpha$ increases and when the stresses ratio $P_Y/P_0$ decreases (Figures 2a and 2b). Moreover, on thin plates $C_{plate}$ also decreases when the parameter $\lambda_z$ increases (Figure 2a). As a consequence, less energy is radiated in the form of elastic waves when the impact is inelastic with respect to the case of an elastic impact.

**Table 1.** Scaling Laws for the Radiated Elastic Energy and the Energy Dissipated in Viscoelasticity[a]

|  | Plates | Blocks |
|---|---|---|
| $W_{el}$ | $a_1 C_{plate} m^{5/3} V_z^{11/5}$ | $a_2 C_{block} m V_z^{13/5}$ |
|  | $a_3 C_{plate} R^5 H^{11/10}$ | $a_4 C_{block} R^3 H^{13/10}$ |
| $W_{visc}$ | $C_{visc} m V_z^2$ | |
|  | $a_5 C_{visc} R^3 H$ | |
|  | $a_1 \approx 0.18 \frac{E^{*2/5}}{\rho_s^{1/15} \sqrt{B\rho_p h}}$ | $a_2 \approx 15.93 \frac{\xi^4 \beta E^{*6/5}}{\rho_p \rho_s^{1/15} c_p^3}$ |
|  | $a_3 = (2g)^{11/10} (\frac{4}{3}\pi \rho_s)^{5/3} a_1$ | $a_4 = (2g)^{13/10} \frac{4}{3}\pi \rho_s a_2$ |
|  | $a_5 = 2g \frac{4}{3}\pi \rho_s$ | |

[a]Radiated elastic energy $W_{el}$ and energy $W_{visc}$ dissipated in viscoelasticity for plates of thickness $h$ and blocks as a function of the impact parameters. The coefficients $a_i$ depend only on the elastic parameters of the impactor and of the structure. The parameter $\beta$ is a function of the Poisson's ratio $v_p$ only (see Figure A1 of Appendix A). The coefficients $C_{plate}$ and $C_{block}$ are represented on Figure 2.

On thick blocks, the radiated elastic energy $W_{el}$ is proportional to the impactor's mass $m$ for a given impact speed $V_z$ (equation (24)). Moreover, the ratio of $W_{el}$ over the impact energy $E_c = \frac{1}{2} m V_z^2$ varies as $V_z^{3/5}$ and is independent of the sphere mass $m$, which is in agreement with *Hunter*'s [1957] findings.

It is important to note that the analytical expressions for the radiated elastic energy $W_{el}$ in Table 1 are only controlled by the impact force $F_z$ and by the rheological parameters of the impactor and the substrate in the vicinity of the impact but do not depend on wave dispersion and viscous dissipation during wave propagation within the substrate.

**2.2.2. Characteristic Frequencies**

The frequency content of the seismic signal emitted by an impact can give information on the impact duration. To describe the amplitude spectrum $|\tilde{A}_z(r, f)|$ of the acceleration vibration, we can measure either of the following:

1. A mean frequency $f_{mean}$ that is less sensitive to the signal to noise ratio than the frequency for which the amplitude spectrum is maximum [*Vinningland et al.*, 2007a, 2007b]:

$$f_{mean} = \frac{\int_0^{+\infty} |\tilde{A}_z(r, f)| f df}{\int_0^{+\infty} |\tilde{A}_z(r, f)| df}, \tag{25}$$

2. The bandwidth $\Delta f$:

$$\Delta f = 2 \sqrt{\frac{\int_0^{+\infty} |\tilde{A}_z(r, f)| f^2 df}{\int_0^{+\infty} |\tilde{A}_z(r, f)| df} - f_{mean}^2}. \tag{26}$$

Regardless of the complexity (fracturation, layers, etc.) of the substrate where the waves emitted by the impact propagate, it is important to notice that the mean frequency $f_{mean}$ and the bandwidth $\Delta f$ are always inversely

[Figure]

**Figure 2.** Values of the constants (a) $C_{plate}$, (b) $C_{block}$, and (c) $C_{visc}$ as a function of the inelastic parameters $\alpha$ for a viscoelastic impact (or $\lambda_Z$ for *Zener*'s [1941] theory) (green) and $P_Y/P_0$ for an elastoplastic impact (red).

**Table 2.** Characteristics Frequencies[a]

|  | $f_{mean}$ | $\Delta f$ |
| --- | --- | --- |
| Plates | $0.75/T_c$ | $0.72/T_c$ |
| Blocks | $1/T_c$ | $0.6/T_c$ |

[a]Theoretical mean frequency $f_{mean}$ and bandwidth $\Delta f$, as, respectively, defined by equations (25) and (26), of the acoustic signal generated by an elastic impact on a thin plate and on a thick block.

proportional to the duration of the impact, which is given by the force history at the position of the impact. Here we normalize these frequencies by *Hertz*'s [1882] impact duration $T_c$. The coefficients of proportionality between $f_{mean}$, $\Delta f$, and $1/T_c$ are estimated for elastic, viscoelastic, and elastoplastic impacts by computing a synthetic spectrum $|\tilde{A}_z(r,f)|$ using equation (2) with the forces represented in Figures 1b and 1c for different values of $\alpha$ and $P_Y/P_0$. The frequencies for an elastic impact, i.e., for $\alpha = 0$ and $P_Y/P_0 = 1$, are given in Table 2. Both frequencies $f_{mean}$ and $\Delta f$ are smaller when the impact is inelastic compared to the case of an elastic impact (Figure 3). They decrease by ~5% when $\alpha$ increases from 0 to 0.5 and by ~25% when the stresses ratio $P_Y/P_0$ decreases from 1 to 0.5.

When normalized by $T_c$, the characteristic frequencies are also affected by wave dispersion and viscous attenuation of energy during propagation, i.e., by Green's function of the structure. These propagation effects are independent of the profile of the impact force, i.e., of the fact that the impact is elastic or inelastic. For the computation of the characteristic frequencies on thick blocks, we used for simplicity the far field approximation of Green's function of Rayleigh waves (equation (4)). This approximation is correct for impacts on homogeneous media such that investigated in the laboratory experiments of section 4. In the field, however, the propagation medium is much more complex and other modes with a different dispersion could develop. In this case, the frequencies normalized by $T_c$ shown in Table 2 could change. Active or passive seismic surveys can allow to evaluate locally Green's function of a specific site. This Green's function can then be used in equations (25) and (26) to estimate how much the normalized frequencies differ from that computed using Green's function of Rayleigh waves. This is, however, beyond the scope of the paper. In addition to dispersion,

[Figure]

**Figure 3.** Theoretical values of the (a and c) mean frequency $f_{mean}$ and (b and d) bandwidth $\Delta f$ for (Figures 3a and 3b) thin plates and (Figures 3c and 3d) thick blocks, as a function of the inelastic parameters $\alpha$ (green) and $P_Y/P_0$ (red). All frequencies are multiplied by *Hertz*'s [1882] impact duration $T_c$ to be dimensionless.

viscous attenuation of energy during propagation can have a significant influence on the measured frequency on the field, especially for high frequencies. *Gimbert et al.* [2014] investigated the amplitude spectrum generated by the turbulent flow in rivers and showed that its central frequency can decrease by a factor of 10 when the distance $r$ from the source increases from 5 m to 600 m, for a quality factor $Q = 20$. To quantify the effect of viscous attenuation on frequencies in our impact experiments, we multiply the synthetic spectrum in equations (25) and (26) by the factor $\exp(-\gamma(\omega)r)$, where $1/\gamma(\omega)$ represents the characteristic distance of energy attenuation. In our experiments, the propagation media are homogeneous and we record the seismic signals close to the impacts, from $r = 2$ cm to about $r = 30$ cm. In this range of distances $r$ and for the substrates investigated in section 4, we estimate that the characteristic frequencies $f_{mean}$ decreases and $\Delta f$ increases by less than 5% when $r$ increases, which is negligible. However, for every practical applications, it is crucial to evaluate wave dispersion and viscous attenuation during propagation and correct the measured seismic signal from these effects before computing its energy $W_{el}$ and its frequencies $f_{mean}$ and $\Delta f$. This correction is systematically performed in our experiments.

**2.2.3. Inverse Scaling Laws**

We can invert the scaling laws derived in this section for the radiated elastic energy $W_{el}$ and for the frequencies $f_{mean}$ and $\Delta f$ (Tables 1 and 2) to express the mass $m$ and the impact speed $V_z$ as functions of the radiated elastic energy $W_{el}$ and a characteristic frequency $f_c$ of the seismic signal that is either $f_{mean}$ or $\Delta f$.

On thin plates, $W_{el} = a_1 C_{plate} m^{5/3} V_z^{11/5}$, $f_{mean} = 0.75/T_c$, and $\Delta f = 0.72/T_c$, then developing the expression of $T_c$ (equation (9)), we obtain

$$m = c_1 \left( \frac{E^{*2}}{(a_1 C_{plate})^{3/11} \rho_s^{1/3}} \right)^{11/16} \frac{W_{el}^{3/16}}{f_c^{33/16}} \tag{27}$$

and

$$V_z = c_2 \left( \frac{\rho_s^{1/3}}{a_1 C_{plate} E^{*2}} \right)^{5/16} W_{el}^{5/16} f_c^{25/16}, \tag{28}$$

where $c_1 \approx 0.046$ or $0.05$ and $c_2 \approx 10.8$ or $10.1$ if $f_c$ is $f_{mean}$ or $\Delta f$, respectively. The coefficient $a_1$ is given in Table 1.

On thick blocks, the inversion of the relations $W_{el} = a_2 C_{block} m V_z^{13/5}$, $f_{mean} = 1/T_c$, and $\Delta f = 0.6/T_c$ gives

$$m = c_3 \left( \frac{E^{*6/5}}{(a_2 C_{block})^{3/13} \rho_s^{1/5}} \right)^{13/16} \frac{W_{el}^{3/16}}{f_c^{39/16}} \tag{29}$$

and

$$V_z = c_4 \left( \frac{\rho_s^{1/5}}{a_2 C_{block} E^{*6/5}} \right)^{5/16} W_{el}^{5/16} f_c^{15/16}, \tag{30}$$

where $c_3 \approx 4.88$ or $4.7$ and $c_4 \approx 0.018$ or $0.02$ if $f_c$ is $f_{mean}$ or $\Delta f$, respectively. The value of $a_2$ is given in Table 1.

The physical characteristics of an impact can then be theoretically deduced from the generated seismic signal. With a continuous recording the seismic signals emitted by rockfalls, such that performed in Dolomieu crater, Réunion Island [e.g., *Hibert et al.*, 2014a], the relations (27) to (30) could be very useful for risks assessment related to these events. Note that the estimation of the impact parameters $m$ and $V_z$ requires a prior evaluation of the elastic properties $\rho_i$, $E_i$, and $\nu_i$ of the impactor and the ground. It should also be noticed that $m$ and $V_z$ strongly depend on the frequency $f_c$. For example, on blocks, if the characteristic frequency is underestimated by a factor of 2, the mass $m$ will be overestimated by a factor of $2^{39/16} \simeq 5.4$. It is therefore necessary to record the entire frequency spectrum to obtain a good estimation of the impact parameters. Because of temporal aliasing during signal sampling, an ideal sampling frequency should be higher than 2 times the highest frequency of the spectrum, which should be at least $f_{mean} + \Delta f/2$. According to Table 2, the sampling frequency should then be at minimum $3/T_c$. For example, in our laboratory experiments in section 4.3, the smallest impact duration $T_c$ is estimated to be $3.10^{-6}$ s. Consequently, we record signals with a sampling

frequency higher than 1 MHz. In contrast, the required sampling frequencies for natural rockfalls should be from a few tens to a few hundreds of hertz, increasing with decreasing impactor's mass. This is discussed in section 5.3.

In section 4.3, the scaling laws presented in Tables 1 and 2 are tested with impacts experiments. Moreover, the masses $m$ and the speeds $V_z$ of the impactors in the experiments are retrieved from the measured seismic signals using equations (27) to (30), and they are compared with their real values.

**2.3. Energy Budget and Coefficient of Restitution**

Another objective of this paper is to establish an energy budget of the impacts. In that way, we compare the radiated elastic energy $W_{el}$ to the total energy lost during the impact $\Delta E_c$. From a practical point of view, the total energy lost by a spherical bead rebounding normally and without rotation can be easily measured from the difference of the bead kinetic energy before and after the impact:

$$\Delta E_c = \frac{1}{2}mV_z^2(1 - e^2),\tag{31}$$

where $e$ is the normal coefficient of restitution, which is the ratio of the bead vertical speeds after and before the impact, respectively, $V'$ and $V_z$ [e.g., *Tillett*, 1954; *Hunter*, 1957; *Reed*, 1985; *Falcon et al.*, 1998; *McLaskey and Glaser*, 2010].

$\Delta E_c$ is the sum of the energy radiated in elastic waves ($W_{el}$), lost in viscoelastic dissipation in the vicinity of the contact ($W_{visc}$) and dissipated by all other processes ($W_{other}$). These other losses can be due to plastic deformation [*Davies*, 1949], surface forces between the sphere and the surface, as, e.g., electrostatic forces [*Israelachvili*, 2002], or in general grain scale interactions [*Duran*, 2010; *Andreotti et al.*, 2013]:

$$\Delta E_c = W_{el} + W_{visc} + W_{other}.\tag{32}$$

In our impacts experiments, the radiated elastic energy $W_{el}$ is deduced from a measurement of the generated seismic signal. Here we present an analytical expression for the energy $W_{visc}$ that will be used later to estimate the losses related to viscoelastic dissipation.

**2.3.1. Energy Lost by Viscoelastic Dissipation**

The energy $W_{visc}$ lost by viscoelastic dissipation in the vicinity of the impact results from the work done by the viscoelastic force $F_{diss} = -\frac{3}{2}DK\frac{d\delta_z(t)}{dt}\delta_z^{1/2}(t)$ during the impact

$$W_{visc} = \int_0^{+\infty} F_{diss}(t) \cdot \frac{d\delta_z(t)}{dt}dt.\tag{33}$$

Using the dimensionless variables $\delta^* = \delta_z/\delta_{z0}$ and $t^* = V_z t/\delta_{z0}$ and the viscoelastic parameter $\alpha = \frac{3}{2}DV_z/\delta_{z0}$, we can show that

$$W_{visc} = C_{visc}mV_z^2,\tag{34}$$

where $C_{visc} = \int_0^{+\infty} \left(\frac{d\delta^*}{dt^*}\right)^2 \delta^{*1/2}dt^*$ is a function of $\alpha$ only (Figure 2c). For an elastic impact, no work is done by the viscoelastic force because $C_{visc} = 0$. The expression of $W_{visc}$ is independent of the fact that the impact is on a plate or on a block because it concerns the energy dissipated in the impact region.

The proportion of total energy $E_c$ dissipated by viscoelasticity can be developed in powers of the mass $m$ and the impact speed $V_z$ using the third order Taylor series $C_{visc} \approx 1.24\alpha - 1.51\alpha^2 + 0.86\alpha^3$ and the expression of $\alpha$ in equation (19):

$$\frac{W_{visc}}{E_c} = 2C_{visc} \approx 3.47x - 5.92x^2 + 4.72x^3 + O(x^3),\tag{35}$$

where $x = DE^{*2/5}\rho_s^{-1/15}m^{-1/3}V_z^{1/5}$, which is in agreement with the viscoelastic impact models of *Kuwabara and Kono* [1987] and *Ramírez et al.* [1999].

**2.3.2. Total Energy Lost**

Finally, if we assume that the sole energy dissipation processes are elastic waves radiation and viscoelastic dissipation and that other energy dissipation processes (e.g., plastic deformation) are negligible, the proportion of the lost energy $\Delta E_c$ radiated in elastic waves is, on plates,

$$\frac{W_{el}}{\Delta E_c} = \frac{a_1 C_{plate} m^{2/3} V_z^{1/5}}{a_1 C_{plate} m^{2/3} V_z^{1/5} + C_{visc}}, \tag{36}$$

and the proportion of the lost energy $\Delta E_c$ dissipated in viscoelasticity is

$$\frac{W_{visc}}{\Delta E_c} = \frac{C_{visc}}{a_1 C_{plate} m^{2/3} V_z^{1/5} + C_{visc}}. \tag{37}$$

In these expressions, at first-order $C_{visc} \propto m^{-1/3}$ (equation (35)). Therefore, when the mass $m$ of the impactor increases, the proportion of the lost energy $\Delta E_c$ radiated in elastic waves should tend toward 100% and that loss by viscoelastic dissipation should tend toward 0%. The transition from a viscoelastic impact (for small masses) toward an elastic impact (for large masses) occurs when $a_1 C_{plate} m^{2/3} V_z^{1/5} = C_{visc}$, i.e., for a critical mass $m_c \approx 8D\sqrt{B\rho_p h}$.

On blocks, we get

$$\frac{W_{el}}{\Delta E_c} = \frac{a_2 C_{block} V_z^{3/5}}{a_2 C_{block} V_z^{3/5} + C_{visc}}, \tag{38}$$

and

$$\frac{W_{visc}}{\Delta E_c} = \frac{C_{visc}}{a_2 C_{block} V_z^{3/5} + C_{visc}}. \tag{39}$$

For large masses $m$, the ratio $W_{el}/\Delta E_c$ becomes independent of $m$ and tends toward 100% because $C_{visc}$ is negligible. When $m$ decreases, the ratio $W_{el}/\Delta E_c$ decreases and the ratio $W_{visc}/\Delta E_c$ increases.

This model is somewhat ideal because the energy dissipated by other processes such as plastic deformation is not negligible when the impactor's mass $m$ is large, in particular when the contact surface is rough. As a consequence, the ratio $W_{el}/\Delta E_c$ practically never reaches 100% when $m$ increases (see section 4.4.2).

The validity of theoretical scaling laws established in this section for the radiated elastic energy, the frequencies, and the lost energy is tested in section 4 with simple impact experiments. Prior to this, the experimental setup is presented in the next section.

**3. Experimental Setup**

We conduct laboratory experiments of beads and gravels impacts on horizontal hard substrates. The generated seismic vibration is recorded on the surface by monocomponent piezoelectric charge shock accelerometers (type 8309, *Brüel and Kjaer*). The response of the sensors is flat between 1 Hz and 54 kHz. The impactor is initially held by a screw and dropped without initial velocity and rotation to ensure reproducibility (Figure 4a). The height of fall $H$ varies between 2 cm and 40 cm. The impact speed $V_z$ is calculated assuming a fall without air friction: $V_z = \sqrt{2gH}$, with $g$ the gravitational acceleration.

We drop spherical beads of steel, glass, and polyamide (Figure 4b) of diameter $d$ ranging from 1 mm to 20 mm to observe the influence of the mass and of the elastic parameters on the results. We conduct the same experiments with granite gravels of irregular shapes and of similar size and mass than the beads to test if the analytical scaling laws established for spheres impacts are still valid if the impactor is not spherical. The properties of the impactors used in the experiments are shown in Table 3.

Four target substrates are used: (i) a smooth PMMA (i.e. polymethyl methacrylate) plate of dimensions $120 \times 100 \times 1$ cm$^3$, (ii) a circular 1 cm thick smooth glass plate of radius 40 cm, (iii) a rough marble block of dimensions $20 \times 20 \times 15$ cm$^3$, and (iv) a rough concrete pillar of dimensions $3 \times 1.5 \times 0.6$ m$^3$. The seismic vibration is recorded at different distances from the impacts to measure waves group speed $v_g = \partial\omega/\partial k$ and

[Figure]

**Figure 4.** (a) Scheme of the experimental setup. An impactor of diameter $d$ is initially held by a screw and dropped without initial speed or rotation on a hard structure of thickness $h$. The height of fall $H$ varies from 2 cm to 30 cm. The impact generates elastic waves, recorded by an array of accelerometers. (b) Spherical beads of glass, polyamide, and steel and granite gravels used as impactors in the experiments.

phase speed $v_\phi = \omega/k$ of the direct wave front in these substrates. These characteristics and the elastic parameters of the investigated structures are summarized in Table 4. We assume that the rheological properties $E_p$, $v_p$, and $\rho_p$ of the substrates in the vicinity of the impact are the same than that within the substrates, where the waves propagate. This hypothesis is valid for the homogeneous solids investigated here, but it may not be correct in the fractured and layered media encountered in the field, whose elastic properties vary with depth. In any cases, it is necessary to determine these properties in order to quantify the radiated elastic energy $W_{el}$ and to deduce thereafter the impact parameters $m$ and $V_z$ from the seismic signal.

**4. Experimental Results**

**4.1. Methods to Estimate the Radiated Elastic Energy**

Let us first describe the signals recorded in our experiments of bead impacts on the different substrates and how we compute the radiated elastic energy $W_{el}$ in each case. A bouncing bead generates a series of short and impulsive acoustic signals (Figures 5a, 5b, 6a, and 6b). The bead can rebound more than 50 times on the smooth glass plate, while it rebounds only 2 or 3 times on the concrete block owing to surface roughness (Figures 5b and 6a). We estimate the coefficient of normal restitution $e = \sqrt{H'/H}$ from the time of flight $\Delta t$ between the successive rebounds because the rebound height is given by $H' = g\Delta t^2/8$ [Falcon et al., 1998; Farin, 2015]. The total energy lost during an impact is then given by $1 - e^2$ (see equation (31)).

The PMMA and glass plates and the concrete block are sufficiently large to measure most of the first wave arrival before the return of the first reflections off the lateral sides (Figures 5c, 5f, and 6e). In these cases, we estimate the radiated elastic energy $W_{el}$ from the energy flux crossing a surface surrounding the impact, as detailed in Farin et al. [2015], i.e., for plates,

$$W_{el} = 2rh\rho_p \int_0^{+\infty} v_g(\omega)|\tilde{V}_z(r, \omega)|^2 \exp(\gamma(\omega)r)\, d\omega \qquad (40)$$

and for blocks

$$W_{el} = 2\rho_p r v_g c_p \pi_R^{surf}(r) \frac{\beta(f_0'(x_0))^2}{2\pi\xi^4(x_0^2 - 1)} \int_0^{+\infty} |\tilde{V}_z(r, \omega)|^2 \omega^{-1} \exp(\gamma(\omega)r)\, d\omega. \qquad (41)$$

**Table 3.** Characteristics of the Impactors Used in Experiments: Density $\rho_s$, Young's Modulus $E_s$, Poisson'S Ratio $v_s$, Diameter $d$, and Mass $m$

|  | Material | $\rho_s$ (kg m$^{-3}$) | $E_s$ (GPa) | $v_s$ - | $d$ (mm) | $m$ (g) |
|---|---|---|---|---|---|---|
| Spherical beads | Glass | 2500 | 74 | 0.2 | 1–20 | $1.3 \cdot 10^{-3}$ to 10 |
|  | Polyamide | 1140 | 4 | 0.4 | 2–20 | $6 \cdot 10^{-4}$ to 4.8 |
|  | Steel | 7800 | 203 | 0.3 | 1–20 | $4.1 \cdot 10^{-3}$ to 33 |
| Gravels | Granite | 3600 | 60 | 0.27 | $\approx 4-28$ | 0.08 to 18 |

**Table 4.** Characteristics of the Materials Used in Experiments[a]

| Material | | $\rho_p$ (kg m$^{-3}$) | $E_p$ (GPa) | $\nu_p$ - | $\gamma$ (1/m) | $\tau$ (s) | $v_g$ (m s$^{-1}$) | $v_\phi$ (m s$^{-1}$) |
|---|---|---|---|---|---|---|---|---|
| Glass | $kh < 1$ | 2500 | 74 | 0.2 | $0.014f^{1/6}$ | $3.8f^{-2/3}$ | $18.6f^{1/2}$ | $9.3f^{1/2}$ |
| | $kh > 1$ | | | | $8.5 \times 10^{-5}f^{2/3}$ | | 3100 | 3100 |
| PMMA | $kh < 1$ | 1180 | 2.4 | 0.37 | 1 | $0.09f^{-1/2}$ | $11.7f^{1/2}$ | $5.8f^{1/2}$ |
| | $kh > 1$ | | | | $4.8 \times 10^{-3}f^{2/3}$ | $0.15f^{-2/3}$ | 1400 | 1400 |
| Concrete | - | 2300 | 16.3 | 0.4 | $2.3.10^{-5}f$ | $28f^{-1}$ | 1530 | 1530 |
| Marble | - | 2800 | 26 | 0.3 | $2.5.10^{-5}f$ | $23.1f^{-1}$ | 1750 | 1750 |

[a]Density $\rho_p$, Young's modulus $E_p$, Poisson's ratio $\nu_p$, characteristic distance $1/\gamma$ and time $\tau$ of energy attenuation, group velocity $v_g$, and phase velocity $v_\phi$ (that depend on the frequency $f$ (in Hz)) (see the supplementary materials of *Farin et al.* [2015], for details on the measurement of $\gamma$ and $\tau$).

In these expressions, $v_g$ is the group speed, $|\tilde{V}_z(r, \omega)|$ is the time Fourier transform of the vertical vibration speed at the surface, and $\pi_R^{surf}(r)$ is the percentage of Rayleigh waves in the signal at the surface at distance $r$ from the impact [*Farin et al.*, 2015]. The factor $\exp(\gamma(\omega)r)$ compensates the viscous dissipation of energy with distance. The characteristic distance of energy attenuation $1/\gamma(\omega)$ is estimated experimentally for every

[Figure]

**Figure 5.** Acceleration signal $a_z(r, t)$ generated by the successive impacts of a steel bead of diameter $d = 5$ mm, dropped from height $H = 10$ cm on (a) the PMMA plate and (b) the glass plate. The time of flight $\Delta t$ between two impacts is equal to the duration between two peaks. (c and d) Zoom on the signal of the first rebound, filtered below 100 kHz. The coda envelope decreases exponentially with time in the glass plate (red line). (c, e, and f) The first arrival is delimited by a red frame, and the first reflections off the plate lateral sides arrive at the right of the blue dashed line. The arrival time of the reflections is computed knowing the wave speed and the distance between the sensor and the substrate sides. (g and h) The time Fourier transform of the first arrival gives the amplitude spectrum $|\tilde{A}_z(r, f)|$ as a function of the frequency $f$. The thick blue line in Figures 5e–5h represents the synthetic signal and amplitude spectrum obtained by convolution of *Hertz*'s [1882] force of impact with Green's function.

**AGU** **Journal of Geophysical Research: Solid Earth** 10.1002/2015JB012331

[Figure]

**Figure 6.** Acceleration signal $a_z(r, t)$ generated by the successive impacts of a steel bead of diameter $d = 5$ mm, dropped from height $H = 10$ cm on (a) the concrete block and (b) the marble block. (c and d) Zoom on the signal of the first rebound, filtered below 100 kHz. The coda envelope decreases exponentially with time (red line). (e and f) The first arrival is delimited by a red frame, and the first reflections off the plate lateral sides arrive at the right of the blue dashed line. The arrival time of the reflections is computed knowing the wave speed and the distance between the sensor and the substrate sides. (g and h) The time Fourier transform of the first arrival gives the amplitude spectrum $|\tilde{A}_z(r, f)|$ as a function of the frequency $f$. The thick blue line in Figures 5e–5h represents the synthetic signal and amplitude spectrum obtained by convolution of *Hertz*'s [1882] force of impact with Green's function.

substrates (Table 4) [see *Farin et al.*, 2015, for details]. The coefficient $\beta$ depends only on the Poisson's ratio $\nu_p$ (see Figure A1 in Appendix A).

Because the substrates size is limited, wave reflections off the boundaries are recorded by the sensors. Side reflections are strongly attenuated in PMMA which is a more damping material than glass, concrete, and marble (Figure 5c). On the contrary, the wave is reflected many times in the glass plate and in the two blocks and its averaged amplitude decreases exponentially with time owing to viscous dissipation during wave propagation (Figures 5d, 6c, and 6d). An adjustment of an exponential curve on the squared signal, filtered below 2000 Hz, allows us to quantify the characteristic decay time of energy $\tau$ in the substrate (Table 4) [see Appendix B of *Farin et al.*, 2015, for details on the experimental procedure]. This situation is referred to as a diffuse field in the literature [e.g., *Weaver*, 1985; *Mayeda and Malagnini*, 2010; *Sánchez-Sesma et al.*, 2011]. In this case, we can estimate the radiated elastic energy $W_{el}$ from the reflected coda. Indeed, in diffuse field approximation, the squared normal vibration speed averaged over several periods decreases exponentially:

$$\overline{v_z(t)^2} = \overline{v_z(t = 0)^2} \exp\left(-\frac{t}{\tau}\right), \tag{42}$$

where $t = 0$ is the instant of the impact. Knowing the characteristic time $\tau$, we extrapolate the vibration speed at the instant $t = 0$ and deduce the radiated elastic energy $W_{el}$ from [*Farin et al.*, 2015]:

$$W_{el} \approx \left(1 + \left(\frac{\mathcal{H}}{\mathcal{V}}\right)^2_{\text{diffuse}}\right) \rho_p V \overline{v_z(t = 0)^2}, \tag{43}$$

where $V$ is the block volume and $\left(\frac{\mathcal{H}}{\mathcal{V}}\right)_{\text{diffuse}}$ is the ratio of horizontal to vertical amplitude at the surface of the structure in diffuse field approximation. On thin plates, $\left(\frac{\mathcal{H}}{\mathcal{V}}\right)_{\text{diffuse}} \simeq 0$. On a thick block of Poisson's ratio $\nu_p$, *Sánchez-Sesma et al.* [2011] give $\left(\frac{\mathcal{H}}{\mathcal{V}}\right)_{\text{diffuse}} \approx 1.245 + 0.348\nu_p$. Due to statistical assumptions, the diffuse method leads to larger uncertainties on the results compared to that based on the energy flux [*Farin et al.*, 2015]. However, it is the only method that can be applied when the first arrival cannot be distinguished from its side reflections, as, for example, in the marble block (Figure 6f).

**4.2. Comparison With Synthetic Signals**

We compare the measured vibration acceleration $a_z(r, t)$ with a synthetic signal which is the time convolution of *Hertz*'s [1882] force of elastic impact (Figure 1b with $\alpha = 0$) with Green's function (equations (3) and (4)) (Figures 5e–5h and 6h).

A good agreement is observed in terms of amplitude and frequencies on the PMMA plate, but the agreement is less satisfactory on the other substrates. On glass, only the beginning of the signal is well reproduced by the theory (Figure 5f). A resonance of the accelerometer coupled to the glass plate for 38 kHz could explain why the recorded vibration lasts longer than the synthetic one (Figure 5f). This effect clearly appears on the Fourier transform of the signal with a peak of energy around 38 kHz (Figure 5h). Using a laser Doppler vibrometer that measures the exact surface vibration speed but with a much lower sensitivity than the accelerometers, we determined that the resonance overestimates the vibration energy by a factor of 4. To compensate this effect, we divide the measured radiated elastic energy $W_{\text{el}}$ by this factor. On concrete, the synthetic is significantly different than the recorded signal in terms of higher amplitude and frequencies (Figures 6f and 6h). The impact may be not completely normal to the surface owing to the surface roughness, and this could reduce the energy on the normal component, as discussed later in section 5. On marble, the frequencies of the measured signal are close to that of the synthetic one but the amplitude is higher than in theory, probably because side reflections arrive before the end of the first arrival (Figures 6e and 6g). This has no consequence on the estimation of the radiated elastic energy $W_{\text{el}}$ for this block because we use the diffuse method (equation (43)). Note that the peaks of energy for $f > 50$ kHz in the synthetic spectrum on the concrete and marble block are not visible in the recordings, because the accelerometers are not sensitive in this frequency range (see Appendix B).

**4.3. Experimental Test of the Analytical Scaling Laws**
**4.3.1. Radiated Elastic Energy**

Regardless of the bead material, the measured radiated elastic energy $W_{\text{el}}$ on the PMMA and glass plates matches well with the theoretical energy $W_{\text{el}}^{\text{th}}$ predicted in equation (23) for an elastic impact, with $C_{\text{plate}} = 1.21$ (Figure 7). For the smallest and the largest beads investigated, however, the data points separate from the theoretical line and the discrepancy can reach a factor of 5. This is clearer for steel beads (Figures 7c and 7g) and for glass beads on the glass plate (Figure 7e).

On blocks, the theory predicts that $W_{\text{el}}^{\text{th}} \propto mV_z^{13/5}$ (equation (24) and Table 1). The experimental data of beads impacts on the concrete and marble blocks follow qualitatively this law (Figure 8). In most of the experiments, however, the measured energy $W_{\text{el}}$ is lower than in theory. Moreover, on concrete, the measured radiated elastic energy $W_{\text{el}}$ separates from the theoretical trend for the smallest and the largest beads investigated (Figures 8a–8c). The discrepancy with the theory on Figures 7 and 8 is interpreted in the discussion.

Surprisingly, the elastic energy $W_{\text{el}}$ radiated by the impacts of granite gravels follows well the scaling law in $m^{5/3}V_z^{11/5}$ on plates (Figures 7d and 7h) and in $mV_z^{13/5}$ on blocks (Figures 8d and 8h). The measured energy $W_{\text{el}}$ is, however, smaller than in theory, by a factor of 2 on plates and up to 10 times smaller on blocks. The experiments with gravels show that Hertz's analytical model of elastic impact, established for spheres, can also describe at first order the impact dynamics of impactors with a complex shape. As a consequence, we expect that it may also be applied for natural rockfalls.

**4.3.2. Characteristics Frequencies**

We compute the mean frequency $f_{\text{mean}}$ and the bandwidth $\Delta f$ using equations (25) and (26), respectively (Figure 9). Note that the seismic signals generated by bead impacts in our experiments contain much higher frequencies (1 Hz–100 kHz) than those recorded for natural rockfalls (1 Hz–50 Hz) [e.g., *Deparis et al.*, 2008; *Hibert et al.*, 2011]. This is because the beads diameters are in average smaller than the diameter of natural

![AGU logo] **Journal of Geophysical Research: Solid Earth**          10.1002/2015JB012331

[Figure]

**Figure 7.** Radiated elastic energy $W_{el}$ as a function of $m^{5/3}V_z^{11/5}$ for impacts of (a and e) glass, (b and f) polyamide, and (c and g) steel beads and (d and h) gravels on (Figures 5a–5d) the PMMA plate and on (Figures 5e–5h) the glass plate. The red line corresponds to the theoretical energy $W_{el}^{th}$ given in Table 1 for an elastic impact, i.e., with $C_{plate} = 1.21$. The black dashed line is a fit to the data of the law $W_{el} = cm^{5/3}V_z^{11/5}$, with coefficient $c$ indicated in International System Units (SI). In most cases, this line collapses with the theoretical line in red. Error bars on $W_{el}$ (±35%) are computed from ±1 standard deviation on a series of 20 experiments and are symbols sized.

boulders, which could be from a few millimeters to a few meters large. In addition, the sampling frequency is much higher and high frequencies are much less attenuated in our experiments than on the field.

On the glass plate, as the accelerometers are not sensitive to frequencies larger than 50 kHz, the frequencies computed with these sensors saturate to about 40 kHz for the smallest beads, i.e., the smallest impact durations $T_c$ (black crosses on Figures 9c and 9d). Therefore, the accelerometers type 8309 are used only for the impacts that generate energy below 50 kHz. For the signals of higher frequencies, we use in parallel piezoelectric ceramics (MICRO-80, *Physical Acoustics Corporation*) sensitive between 100 kHz to 1 MHz. These last sensors can, however, not be used to quantify the radiated elastic energy $W_{el}$ since they are not very sensitive to frequencies lower than 100 kHz.

[Figure]

**Figure 8.** Radiated elastic energy $W_{el}$ as a function of $mV_z^{13/5}$ for impacts of (a and e) glass, (b and f) polyamide, and (c and g) steel beads and (d and h) gravels on (Figures 8a–8d) the concrete block and on (Figures 8e–8h) the marble block. The red line corresponds to the theoretical energy $W_{el}^{th}$ given in Table 1 for an elastic impact, i.e., with $C_{block} = 0.02$. The black dashed line is a fit to the data of the law $W_{el} = cmv_z^{13/5}$, with coefficient $c$ indicated in International System Units (SI).

Regardless of the bead material, the frequencies of the signals generated by impacts on PMMA, glass, and marble collapse well within ±20% with the theoretical scaling laws of Table 2 as a function of the duration of impact $T_c$ (Figures 9a to 9d, 9g, and 9h). The agreement is better for the frequency bandwidth $\Delta f$ than for the mean frequency $f_{mean}$. The agreement is also very satisfactory for the granite gravels of complex shape, even though the theoretical values of the frequencies were computed using Hertz's impact model for a sphere (see section 2.2.2).

In concrete, the wavelength $c_R/f \approx 1$ cm for frequencies around 40 kHz, which is of the order of the size of the heterogeneities. High frequencies $f > 40$ kHz are therefore strongly attenuated during wave propagation in this block. This could explain the discrepancy with the theory for these frequencies on Figure 9e.

[Figure]

**Figure 9.** (a), (c), (e), and (g) Mean frequency $f_{mean}$ and (b), (d), (f), and (h) bandwidth $\Delta f$ as a function of *Hertz*'s [1882] impact duration $T_c$ (equation (9)) for impacts of glass, polyamide, and steel beads and granite gravels on (Figures 9a and 9b) the PMMA plate, (Figures 9c and 9d) the glass plate, (Figures 9e and 9f) the concrete block, and (Figures 9g and 9h) the marble block. The red line corresponds to the theoretical prediction (Table 2), and the red dashed line in Figure 9e is a fit to the data. The black crosses in Figures 9c and 9d correspond to the frequencies of the signals generated by steel beads measured with the accelerometers type 8309, which resonate around 38 kHz on the glass plate (see text). Error bars are of the size of the symbols and are ±20%.

**4.3.3. Estimating Impact Properties From the Seismic Signal**

We use equations (27) to (30) with the coefficients for an elastic impact $C_{plate} = 1.21$ and $C_{block} = 0.02$ to retrieve the mass $m$ and the impact speed $V_z$ of the impactors in our experiments. The agreement with the real values is correct, within a factor of 2 for the mass $m$ (Figure 10a) and within a factor of 3 for the impact speed $V_z$ (Figure 10b), both on smooth thin plates and rough thick blocks. For impacts of rough gravels on the two plates, the predicted values are still close to the real ones, with a factor of 1.5, even when inelastic dissipation occurs. The underestimation of $m$ and $V_z$ in certain cases is consistent with the aforementioned discrepancy of the radiated energy $W_{el}$ with theory (Figures 7 and 8).

It is therefore possible to have an estimation of the mass $m$ and the impact speed $V_z$ of an impactor on a plate and on a block from the characteristics of the generated seismic signal, with less than an order of magnitude from the real values, using only *Hertz*'s [1882] analytical model of elastic impact. This method only requires to know the elastic parameters of the involved materials.

[Figure]

**Figure 10.** (a) Mass $m_{inv}$ inverted from signal bandwidth $\Delta f$ and radiated elastic energy $W_{el}$ using equation (27) for plates and equation (29) for blocks as a function of the real mass $m$. (b) Impact speed $V_{zinv}$ inverted using equation (28) for plates and equation (30) for blocks as a function of the real impact speed $V_z$. The black full line is a perfect fit.

**4.4. Energy Budget of the Impacts**

Inelastic losses during an impact can reduce the energy radiated in the form of elastic waves $W_{el}$ compared to that predicted by *Hertz*'s [1882] model (see section 2.2.1). This may explain part of the discrepancy observed between the measured radiated elastic energy $W_{el}$ and its theoretical value $W_{el}^{th}$ on Figures 7 and 8 and consequently between the values of the masses $m$ and speeds $V_z$ inverted from seismic signals and their real values on Figure 10. In order to interpret these discrepancies, we establish in this section an energy budget of the impacts.

For that purpose, we compare on Figures 11 and 13 the measured radiated elastic energy $W_{el}$ (empty symbols) with the total energy lost during the impact $\Delta E_c$, estimated with the coefficient of restitution $e$ (full symbols). The difference $\Delta E_c - W_{el}$ is likely lost in inelastic processes, such as viscoelastic dissipation or plastic deformation. This allows us to establish an energy budget of the impacts (Figures 12 and 14).

Furthermore, we also compare the measured radiated energy $W_{el}$ with the theoretical one—noted $W_{el}^{th}$, red line on Figures 11 and 13—predicted by the scaling law in Table 1 for an elastic impact, with $C_{plate} = 1.21$ and $C_{block} = 0.02$, respectively. Note that on plates, we take into account the dependence of $C_{plate}$ coefficient to $\lambda_Z$ parameter for large beads (see section 2.1.1.2 and Figure 2a). The corrected theoretical elastic energy on plates is noted $W_{el}^{th'}$ on Figure 11. The discrepancy with theory is discussed in section 5.1.

**4.4.1. Energy Budget on Smooth Thin Plates**

On smooth thin plates, the energy $\Delta E_c$ lost by the bead during an impact is mostly radiated in elastic waves ($W_{el}$) or dissipated by viscoelasticity during the impact ($W_{visc}$) (Figures 11 and 12).

More energy is radiated in elastic waves as the bead mass $m$ and the ratio of the bead diameter $d$ on the plate thickness $h$ increase, regardless of the elastic parameters (empty symbols on Figures 11 and 12). For the smallest beads investigated, only 0.1% to 0.3% of the impact energy $E_c$ is radiated in elastic waves. In contrast, the impact energy $E_c$ can be almost entirely converted into elastic waves when the bead diameter $d$ is greater than the plate thickness $h$ (Figure 11c). For large beads, the measured ratio of $W_{el}/E_c$ is close to the theoretical ratio $W_{el}^{th'}/E_c$ (full red line on Figure 11) but diverges as the bead diameter $d$ decreases.

We adjust the viscoelastic parameter $D$ in equation (35) to match the theoretical expression of the lost energy ratio $\Delta E_c/E_c = W_{el}^{th'}/E_c + W_{visc}/E_c$ (thick green line on Figure 11) with the variation of $1 - e^2$ (full symbols). The agreement is found to be the best for values of $D$ ranging from 35 ns to 580 ns (Table 5).

The adjustment of $D$ with experimental data allows us to quantify the viscoelastic energy $W_{visc}$ (blue line on Figure 11). More energy is lost by viscoelastic dissipation as the bead mass $m$ and the ratio $d/h$ decrease, and this is almost the sole process of energy loss when the bead diameter $d$ is smaller than $0.2h$ (Figure 12). The transition from a viscoelastic impact toward an elastic impact is observed for the critical mass $m_c \approx 8D\sqrt{B\rho_p h}$,

[Figure]

**Figure 11.** Ratio of the measured radiated elastic energy $W_{el}$ over the impact energy $E_c = \frac{1}{2}mV_z^2$ (empty symbols) and measured lost energy ratio $\Delta E_c/E_c = 1 - e^2$ (full symbols) as a function of bead mass $m$ and of the ratio of the bead diameter $d$ on the plate thickness $h$ for impacts of (a and d) glass, (b and e) polyamide, and (c and f) steel beads on (Figures 11a–11c) the PMMA plate and on (Figures 11d–11f) the glass plate. The red dashed line corresponds to the theoretical ratio $W_{el}^{th}/E_c$ with $W_{el}^{th}$ in equation (23) for an elastic impact, i.e., with $C_{plate} = 1.21$. The red full line is the energy ratio $W_{el}^{th'}/E_c$ corrected with $C_{plate}$ dependence on parameter $\lambda_Z$, the blue line is the viscoelastic energy ratio $W_{visc}/E_c$ (equation (35)), and the thick green line is the theoretical lost energy ratio, which is the sum of $W_{el}^{th'}/E_c$ and $W_{visc}/E_c$.

as predicted in section 2.3.2 (at the crossing between the red and blue lines on Figure 11). Interestingly, a bouncing bead loses less of its initial energy $E_c$ for masses $m$ close to the critical mass $m_c$.

For the largest beads of glass and steel, some energy is likely lost in plastic deformation of the softer material involved (Figure 12). As a matter of fact, we observed small indentations on the surface of the plates after the impacts of these beads but not for polyamide beads.

Note that the energy budget is very different for impacts of rough gravels on the same plates. Indeed, the ratio $W_{el}/E_c$ is 3.3% ± 1.8% regardless of the gravel mass $m$. Moreover, about 33% ± 17% of the initial energy is lost in translational energy of rebound and 13% ± 11% is converted into rotational energy of the gravel. As a matter of fact, half of the gravel's initial energy is in average lost in plastic deformation. (see Appendix C for more details).

[Figure]

**Figure 12.** Percentage of the total energy lost in elastic waves $W_{el}/\Delta E_c$ (red full line), by viscoelastic dissipation $W_{visc}/\Delta E_c$ (blue dashed line) and by other processes $W_{other}/\Delta E_c$ (orange dotted line) as a function of (a and c) the bead mass $m$ and (b and d) the ratio of the bead diameter $d$ over the plate thickness $h$ for impacts of glass (circles), polyamide (triangles), and steel (diamonds) beads dropped from height $H = 10$ cm on (Figures 12a and 12b) the PMMA plate and on (Figures 12c and 12d) the glass plate.

**4.4.2. Energy Budget on Rough Thick Blocks**

On the rough thick blocks, the energy budget is very different than on the smooth plates (Figures 13 and 14). Indeed, a much smaller proportion of energy seems to be lost in elastic waves and in viscoelastic dissipation. The rest is likely dissipated by other processes such as plastic deformation, adhesion, or rotational modes of the bead owing to surface roughness.

The measured radiated elastic energy $W_{el}$ represents only from 0.01% to 2% of the impact energy $E_c$, regardless of the bead mass $m$ (empty symbols on Figure 13). Theory predicts that the ratio $W_{el}^{th}/E_c$ is independent of the mass $m$ (red line). However, the measured ratio $W_{el}/E_c$ slightly increases with bead mass $m$ on concrete and decreases on marble for different reasons explained in the discussion.

Contrary to plates, it is difficult here to determine what proportion of the lost energy $\Delta E_c$ is dissipated by viscoelasticity and what proportion is lost in other processes. However, one remarks that $1 - e^2$ increases when the mass $m$ decreases (full symbols on Figure 13). This variation may be due to viscoelastic dissipation which is stronger when the bead mass $m$ decreases (equation (35)). We make the strong assumption that the percentage of energy lost in other processes $W_{other}/E_c$ is constant and independent of the bead mass $m$.

**Table 5.** Viscoelastic Constant $D$ (in ns)[a]

| substrate | | PMMA | glass | concrete | marble |
|---|---|---|---|---|---|
| bead | glass | 230 | 80 | 100 | 180 |
| | polyamide | 580 | 550 | 300 | 300 |
| | steel | 190 | 35 | 200 | 200 |

[a]Value of the viscoelastic constant $D$ appearing in equation (19) and adjusted on experimental data for impacts of spherical beads of different material (rows) on the different substrates (columns).

[Figure]

**Figure 13.** Ratio of the measured radiated elastic energy $W_{el}$ over the impact energy $E_c = \frac{1}{2}mV_z^2$ (empty symbols) and measured lost energy ratio $\Delta E_c/E_c = 1 - e^2$ (full symbols) as a function of bead mass $m$ for impacts of (a and d) glass, (b and e) polyamide, and (c and f) steel beads on (Figures 13a–13c) the concrete block and on (Figures 13d–13f) the marble block. The red line represents the theoretical ratio $W_{el}^{th}/E_c$ with $W_{el}^{th}$ in equation (24) with $C_{block} = 0.02$. The blue line is the viscoelastic energy ratio $W_{visc}/E_c$ (equation (35)). The dashed green line is the theoretical lost energy ratio $W_{el}^{th}/E_c + W_{visc}/E_c$. The thick green line is the same ratio plus the percentage $W_{other}/E_c$ of energy lost in other processes, which is assumed independent of the bead mass $m$ (see text).

We then adjust the viscoelastic coefficient $D$ (Table 5) to fit $\Delta E_c/E_c = W_{el}^{th}/E_c + W_{visc}/E_c + W_{other}/E_c$ (thick green line on Figure 13) with the variation of $1 - e^2$ (full symbols). This allows to quantify the energy $W_{visc}$ lost in viscoelastic dissipation (blue line).

In the case where no other energy losses than elastic waves radiation or viscoelastic dissipation occur, we predicted that the ratios $W_{el}/\Delta E_c$ and $W_{visc}/\Delta E_c$ should increase and tend toward 100% when the mass $m$ increases and decreases, respectively (equations (38) and (39)). Here elastic waves radiation and viscoelastic dissipation follow the same dependence on the mass than that predicted but represent, respectively, from 0.03% to 5% and from 2% to 40 % of the lost energy $\Delta E_c$ only (Figure 14). For impacts on rough substrates as the two blocks investigated here, but also on the field, it is therefore important to take into account the energy $W_{other}$ lost in other processes. In our experiments, this energy seems to be an increasing percentage of the lost energy $\Delta E_c$, from 50% to more than 99%, as the bead mass $m$ increases (Figure 14).

[Figure]

**Figure 14.** Percentage of the total energy lost in elastic waves $W_{el}/\Delta E_c$ (red full line), by viscoelastic dissipation $W_{visc}/\Delta E_c$ (blue dashed line) and by other processes $W_{other}/\Delta E_c$ (orange dotted line) as a function of the bead mass $m$ for impacts of glass (circles), polyamide (triangles), and steel (diamonds) beads dropped from height $H = 10$ cm on (a) the concrete block and on (b) the marble block.

**4.4.3. Evaluation of the Energy Budget for Natural Rockfalls**

The energy budget of impacts on rough blocks in our laboratory experiments can be used to extrapolate that of natural rockfalls. On the field, the impactor masses vary from a few grams to a few tons and drop heights vary from a few centimeters to several tens of meters. Owing to strong energy dissipation in such complex media, only impacts of large masses can be detected by seismic methods. Viscoelastic dissipation should therefore be negligible in most situations encountered on the field. For example, we can estimate the energy lost in viscoelastic dissipation for a granite gravel of $m = 100$ g impacting the ground with impact speed $V_z = 10$ m s$^{-1}$ using equation (35) with the coefficient $D = 80$ ns of glass, which has similar properties than granite, and a typical Young's modulus $E_p = 10$ MPa for the ground [*Geotechdata.info*, 2013]. It results that the viscoelastic energy $W_{visc}$ lost during the impact represents only 0.04% of the impact energy $E_c$, which is negligible. Moreover, it should be even smaller for larger masses $m$. The energy $W_{plast}$ dissipated in plastic deformation of the ground or of the impactor is expected to be much more significant on the field than in our laboratory experiments and even more so when the mass $m$ increases because large stresses are applied on damaged materials with a low yield stress. For such impacts with a rough contact, the energy $W_{plast}$, in addition to other energy lost in rotation and translational modes of the impactor, should then represent almost all of the lost energy $\Delta E_c$ (see Appendix C). Consequently, the ratio of the radiated elastic energy over the lost energy $W_{el}/\Delta E_c$ may not exceed a few percents. For example, for impacts of beads on the rough concrete block, for which plastic deformation is significant, the ratio $W_{el}/\Delta E_c$ seems to saturate to 2% $\pm$ 1% for $m \simeq 1$ g and then decreases (Figure 14a).

**5. Discussion**

**5.1. Discrepancy From Hertz's Model**

The characteristic frequencies of the signal generated by an impact do not significantly deviate from *Hertz*'s [1882] prediction when the impact is inelastic (Figure 9). On the contrary, in some experiments, the measured radiated elastic energy $W_{el}$ diverges from that (noted $W_{el}^{th}$) given by the scaling laws in Table 1 (Figures 7 and 8). As a consequence, the masses $m$ and speeds $V_z$ retrieved from the measured signal in our experiments using the elastic model deviate from their real values (Figure 10). Let us discuss here the observed discrepancy.

**5.1.1. Small Bead Diameters**

On smooth thin plates, for small bead diameters, viscoelastic dissipation is the major energy loss process (Figure 12). For a steel bead of diameter 1 mm impacting the glass plate, using equation (19) with $D = 35$ ns (see Table 5), the coefficient $C_{plate}$ is found to be equal to 1.15 instead of 1.21 for an elastic impact (see Figure 2a). Thus, the viscoelastic impact theory predicts that the radiated elastic energy $W_{el}^{th}$ should be only of 5% smaller than for an elastic impact, which is negligible compared with the observed difference of 73% (Figure 7g).

The major source of discrepancy is probably due to the fact that our sensors are band limited up to 50 kHz. Indeed, for the 1 mm bead, 50% of the radiated energy is in theory higher than 50 kHz (see Appendix B). The

remaining 23% may be lost in adhesion of the bead on the plate during the impact. In addition, some energy may be lost in electrostaticity or capillarity, which are greater for the smallest beads [*Andreotti et al.*, 2013]. The discrepancy is totally explained by the limited bandwidth of the accelerometers for a steel bead of diameter $d = 2$ mm on the glass plate: about 30% of the energy is over 50 kHz and the measured energy $W_{el}$ is 35% smaller than $W_{el}^{th}$ (Figure 7g). Similarly, on concrete, for a steel bead of diameter $d = 2$ mm, the theory predicts that only 17% of the radiated elastic energy is below 50 kHz. As a consequence, the measured energy $W_{el}$ represents only 17% of the theoretical energy $W_{el}^{th}$ (Figure 8c). For greater bead diameters, both measured and theoretical energies are contained below 50 kHz and the agreement with elastic theory is better (Figures 7 and 8). In contrast, on marble the radiated elastic energy is closer to the theory for the smallest beads (Figures 13d to 13f). For small bead diameters, less wave reflections occur within the block and the measured energy may therefore be overestimated because the diffuse field is not completely set [*Farin et al.*, 2015].

This emphasize the importance for future applications to use seismic sensors sensitive in the widest frequency range as possible. In cases where we cannot measure the highest frequencies of the seismic vibration generated by an impact, note that it is possible to retrieve the momentum $mV_z$ of the impactor from the low-frequency content of measured amplitude spectrum (see Appendix D).

**5.1.2. Large Bead Diameters**

On smooth thin plates, the divergence of the measured radiated elastic energy $W_{el}$ from the theoretical one $W_{el}^{th}$ for large bead diameters is partly compensated when we take into account the decrease of the coefficient $C_{plate}$ when the parameter $\lambda_z$ increases (Figures 2a and 11). However, in some experiments, $W_{el}$ is still smaller than the theory when the bead diameter $d$ is larger than the plate thickness $h$ (Figures 11c, 11d, and 11f). This difference may be due to plastic deformation which is more likely to occur for the largest beads investigated.

**5.1.3. Impacts With a Rough Contact**

Two complementary effects can explain the discrepancy of the measured radiated elastic energy with theory for impacts of spherical beads on the two rough blocks and for impacts of gravels (Figures 7d, 7h, and 8).

First, plastic deformation is a likely cause for measuring a smaller radiated elastic energy than in theory on the blocks. If $P_Y/P_0 = 0.6$ in the elastoplastic model, the radiated elastic energy predicted in Table 1 is 2 times smaller than for an elastic impact because the coefficient $C_{block} \approx 0.01$ instead of 0.02 (Figure 2b). This factor of 2 corresponds to that observed between the measured energy $W_{el}$ and the theoretical one $W_{el}^{th}$ for impacts of glass and steel beads on the concrete block (Figures 8a and 8c). Measuring the discrepancy of the radiated elastic energy from elastic theory could then be a mean to estimate the dynamic yield strength $P_Y$ of a material. For example, for a steel bead of diameter $d = 5$ mm dropped from height $H = 10$ cm on concrete, the maximum stress is $P_0 \approx 300$ MPa and, if $P_Y/P_0 = 0.6$, the dynamic yield strength would be $P_Y \approx 180$ MPa, which is greater than the typical values of $P_Y$ for concrete (20–40 MPa) [*The Engineering Toolbox*, 2014] but of the same order of magnitude.

An additional process can accommodate the discrepancy. If a spherical bead impacts a rough surface or as a gravel impacts a flat surface, the equivalent radius of contact may be smaller than the radius of the impactor (Figure 15). Table 1 shows that the radiated elastic energy $W_{el}$ increases with the impactor radius $R$ as $R^5$ on plates and as $R^3$ on blocks. Then, if the radius of contact $R$ is only 1.15 smaller on plates, the theoretical radiated elastic energy $W_{el}$ is 2 times smaller, and this explain the discrepancy observed for gravels on the plates (Figures 7d and 7h). On blocks, if the effective radius of contact $R$ is 2.1 times smaller, the radiated elastic energy $W_{el}$ is 10 times smaller, which could explain the small energy values measured on the marble block (Figures 8e to 8h). The radius of contact $R$ should be even smaller when gravels impact the rough blocks and the radiated elastic energy $W_{el}$ is then smaller (Figures 8d and 8h). By comparison, the characteristic frequencies $f_{mean}$ and $\Delta f$ are inversely proportional to the radius $R$ (because $T_c \propto R$) and are therefore less affected by a change in this radius than the radiated elastic energy $W_{el}$. This is visible on Figure 9 because the frequencies of the signal emitted by gravels are close to that of spherical beads.

As the effective radius of contact decreases for a given mass $m$, the stresses are concentrated on a smaller area during the impact and plastic deformation is more likely to occur (see Appendix C). Interestingly, even though the energy lost in plastic deformation is very important for impacts of gravels and on the rough blocks, the measured radiated elastic energy $W_{el}$ and frequencies $f_{mean}$ and $\Delta f$ still follow well the scaling laws in mass $m$ and impact speed $V_z$ predicted using Hertz's model of impact of a sphere on a plane (Figures 7, 8, and 9). Therefore, we expect that Hertz's model should still be valid at first order on the field and, consequently, that

[Figure]

**Figure 15.** Schematic of the contacts between a sphere and a rough surface and between a rough gravel and a flat surface.

the radiated elastic energy $W_{el}$ should be proportional to $mV_z^{13/5}$ and that the characteristic frequencies $f_{mean}$ and $\Delta f$ should be proportional to $1/T_c \propto m^{-1/3}V_z^{1/5}$. The problem is, however, to determine the coefficients of proportionality in these relations because they depend on the rheological parameters of the impactor and the ground (Table 1), on the fact that the impact is elastic or inelastic (Figures 2 and 3) and on the roughness of contact, which are each extremely difficult to estimate practically. A solution may be to calibrate the coefficients of proportionality of these relations on a given site by dropping some boulders of known mass $m$ and estimating their impact speed $V_z$ with a camera. Once calibrated, these laws can be inverted as in section 2.2.3 and used to retrieve the masses $m$ and impact speeds $V_z$ of other rockfalls on the same site from the generated seismic signals. The advantage of this method is that it is not necessary to know the elastic parameters of the ground. Even so, energy attenuation as a function of frequency during wave propagation within the substrate needs to be evaluated in order to correct the measured signals.

**5.2. Errors on the Estimation of the Masses and Impact Speeds**

Here we comment the errors on our estimation of the impactor's masses from measured seismic signals in Figure 10. These errors are greater than that of *Buttle et al.* [1991] who managed to size submillimetric particles in a stream with a standard deviation less than 10%. However, their estimations were based on the impact force and duration on the direct compressive wave, measured at the opposite of the impact on the target block. Practically, this method is difficult to apply on the field because seismic stations are at the surface. Furthermore, the force and duration of the impact are more complicated to estimate from the seismic signal than the radiated elastic energy and the frequencies because they require a deconvolution process that induce additional errors [e.g., *McLaskey and Glaser*, 2010]. Our method has the advantage to not be intrusive and, in principle, exportable to field problems.

**5.3. Application to Natural Rockfalls**

*Dewez et al.* [2010] conducted field scale drop experiments of individual basalt boulders on a rock slope in Tahiti, French Polynesia. The main objective of this study was to estimate hazards associated with rockfalls in a volcanic context. Boulder trajectories were optically monitored using two cameras with 50 frames per second. A photogrammetry technique then allowed the authors to compute the position of each boulder in time with an error smaller than the boulder radius [*Dewez et al.*, 2010]. In parallel, the seismic signal generated by boulder impacts on the ground was recorded with a sampling frequency of 100 Hz by a board band seismometer type *STS* located a few tens of meters away. Here we want to observe how the elastic energy radiated by boulder impacts scales with the boulder mass and speed in this natural context.

**5.3.1. Comparison of Field Measurements With Hertz's Prediction**

The waves generated by the impacts propagate in a very damaged and complex medium that may involve several layers of different density. In this medium, viscous attenuation of energy can be very strong, especially for high frequencies. For example, waves of frequency 100 Hz only propagate in the first centimeters or meters deep below the surface. The attenuation of energy as a function of frequency can be evaluated, for example, by measuring the signal emitted by a given impact at different distances, as we did in our laboratory experiments [*Farin et al.*, 2015]. Unfortunately, no estimation of the attenuation has been conducted in this field study. We therefore assume a classical model of viscous attenuation of energy with distance $r$ and multiply the measured signals $\tilde{A}(r, f)$ by the factor $\exp\left(\gamma(f)r/2\right)$, with $\gamma(f) = 2\pi f/(Qc_R)$, where $Q$ is the quality factor of the ground and $c_R$ is the phase speed of Rayleigh waves [*Aki and Richards*, 1980]. We use $c_R = 800$ m s$^{-1}$ as in the Piton de la Fournaise volcano, Reunion Island, where the ground has a similar structure as in Tahiti [*Hibert et al.*, 2011]. Using this model with $Q = 20$ and typical boulder properties in these experiments, $m = 1000$ kg

[Figure]

**Figure 16.** (a) Vertical vibration speed $v_z(r, t)$ generated by two successive impacts of a boulder of mass $m = 326$ kg on the rock slope. (b) Spectrogram of the signal in Figure 16a. Darker shape represents higher energy (normalized). The black lines highlight the triangular shape of the spectrograms. (c) Amplitude spectrum $|\tilde{V}_z(r, f)|$ for the first impact, with the peak $f_{peak}$ and mean $f_{mean}$ frequencies and the bandwidth $\Delta f$. Dashed line: synthetic spectrum computed with the convolution of *Hertz*'s [1882] force of elastic impact with Green's function of Rayleigh waves. (d–f) Radiated elastic energy $W_{el}$ for different boulders documented in Tahiti as a function of (Figure 16d) the mass $m$ and of parameters (Figure 16e) $mV_z^{13/5}$ and (Figure 16f) $mV_z^{0.5}$, with associated coefficients of determination $R^2$. (g) Ratio of the radiated seismic energy $W_{el}$ over the kinetic energy $\Delta E_c$ lost during the impact as a function of the boulder mass $m$ for $V_z = 5 \pm 2$ m s$^{-1}$. Red full and dashed lines in Figures 16d–16g are adjustments to the data for "hard" and "soft" impacts, respectively.

and $V_z = 10$ m s$^{-1}$, we estimate that if we do not correct the measured signals from viscous attenuation, we measure 90% of the radiated elastic energy at $r = 13$ m from the source and 50% at $r = 90$ m. In contrast, for $Q = 5$, i.e., for a more damaged medium, we would measure 90% of the radiated elastic energy at $r = 3$ m and 50% at only $r = 22$ m. The corrected amplitude spectrum should in theory be equivalent to the emitted spectrum. However, the spectrum cannot be corrected for all frequencies where the measured amplitude is below the noise level. It is therefore important to record the seismic signals as close as possible to the impacts. For the following computations, we use the quality factor $Q = 10$, which is of the order of the values obtained by *Ferrazzini and Aki* [1992] in the similar context of Kilauea volcano in Hawai'i.

We first focus on the seismic signals emitted by the impacts of a boulder of mass $m = 326$ kg at $r \simeq 30$ m from the seismometer (Figure 16a). The signals have a short duration $\sim 0.8$ s and are impulsive, as the ones generated by bead impacts (e.g., Figure 6c). The impacts excite a frequency range from $\sim 10$ Hz to 40 Hz (Figure 16b). Most of the recorded seismic spectra lies between 10 Hz and 20 Hz with a peak frequency $f_{peak} \approx 15.5$ Hz, a mean frequency $f_{mean} \approx 18.4$ Hz, and a bandwidth $\Delta f \approx 18.3$ Hz (Figures 16b and 16c).

We compare the measured spectrum with a synthetic amplitude spectrum predicted by *Hertz*'s [1882] theory of impact using equation (2). Green's function used in the computation depends on the excited mode. *Deparis et al.* [2008], *Dammeier et al.* [2011], and *Lévy et al.* [2015] showed that rockfall events generate principally Rayleigh surface waves. Rayleigh waves develop in far field, i.e., for $kr >> 1$, where $k = 2\pi f / c_R$ is the wave number [*Miller and Pursey*, 1954; *Gimbert et al.*, 2014; *Farin et al.*, 2015]. Using the phase speed $c_R = 800$ m s$^{-1}$, we estimate that $kr >> 1$ when the frequency $f$ is greater than about 4 Hz. Since the recorded seismic energy is mostly between 10 Hz and 40 Hz, we can therefore reasonably use the far field Green's function of Rayleigh waves of equation (4) convolved with *Hertz*'s [1882] impact force to compute the synthetic spectrum (Figure 16c).

The characteristics of the impactor are $R = 0.35$ m, $m = 326$ kg, and $V_z = 11$ m s$^{-1}$. We assume a typical Young's modulus $E_p = 10$ MPa for a loose soil such that observed on the slope [*Geotechdata.info*, 2013]. *Hertz*'s [1882] elastic theory then predicts that the duration of impact should be $T_c \simeq 0.035$ s (equation (9)). For Rayleigh surface waves, the mean frequency should therefore be $f_{mean} = 1/T_c \simeq 28$ Hz and the bandwidth $\Delta f = 0.6/T_c \simeq 17$ Hz, which are close to the measured values (Table 2 and Figure 16c).

The amplitude of the synthetic spectrum is similar to that of the measured spectrum except around 15 Hz where a peak of energy is observed in the measured spectrum (Figure 16c). The peak of energy may be due to a resonance around 15 Hz of the seismometer or of the first sediment layers because it is observed on every measured spectra [*Schmandt et al.*, 2013; *Farin*, 2015]. The shape of the measured and synthetic spectrum is very different. This may be due to plastic deformation, which is very important for impacts on loose and fractured soil.

**5.3.2. Elastic Energy Radiated by Boulders' Impacts**

Despite the discrepancy between the theory and the measurement, we observe how the elastic energy $W_{el}$ radiated by the impacts of all boulders depends on the boulder mass $m$ and impact speed $V_z$. The calculation of $W_{el}$ is based on the integration of the energy flux over a cylinder surrounding the impacts [*Hibert et al.*, 2011; *Farin et al.*, 2015]:

$$W_{el} = \int_0^{+\infty} 4\pi r h \rho_p c_R |\tilde{V}(r,f)|^2 \exp\left(\gamma(f)r\right) \, \mathrm{d}f, \tag{44}$$

where $h = c_R/f$ is the Rayleigh wavelength and $|\tilde{V}(r,f)|^2 = |\tilde{V}_X(r,f)|^2 + |\tilde{V}_Y(r,f)|^2 + |\tilde{V}_Z(r,f)|^2$ is the sum of the squared time Fourier transforms of the vibration speeds in the three directions of space $v_X(r,t)$, $v_Y(r,t)$, and $v_Z(r,t)$, respectively. The coefficient $\gamma(f) = 2\pi f/(Qc_R)$ is the same than that used to compute the synthetic spectrum in the previous section, with $c_R = 800$ m s$^{-1}$ and $Q = 10$.

The nature of the contact between the boulder and the ground during the impact plays a crucial role on the transfer of the seismic energy. Therefore, we separated the "hard" impacts, occurring on outcropping rock, from the "soft" impacts, occurring on loose soil or on grass (Figures 16d–16g). The measured radiated elastic energy $W_{el}$ seems to be proportional to the mass $m$ as predicted analytically for impacts on thick blocks (Table 1 and Figure 16d). This dependance is clearer for soft impacts. However, the measured radiated elastic energy $W_{el}$ does not scale well with the parameter $mV_z^{13/5}$ derived from Hertz's theory (Figure 16e). We adjust the power $a$ of parameter $mV_z^a$ to obtain a better fit with $W_{el}$. The best fit is observed for power $a \simeq 0.5$, i.e., with a much weaker dependence on the impact speed $V_z$ than in theory, with $W_{el} \propto V_z^{0.5}$ rather than

$W_{el} \propto V_z^{13/5}$ (Figure 16f). The scaling law in $V_z^{0.5}$ may be biased because boulders systematically impacted loose soil when they reached high speeds $V_z$, while they often impacted outcropping rocks for lower speeds $V_z$. The energy transfer is lower for "loose" impacts than for hard impacts, and this may then lead to the observed weaker dependence in $V_z$ (Figure 16g). As a matter of fact, the mean ratio of the radiated elastic energy $W_{el}$ over the kinetic energy $\Delta E_c$ lost during the impacts is 1 order of magnitude higher for hard impacts than for soft impacts (Figure 16g). Interestingly, the ratio $W_{el}/\Delta E_c$ is between $10^{-4}$ and $10^{-1}$, which is in agreement with the values observed by *Hibert et al.* [2011].

No clear dependence on $m$ and $V_z$ was observed for the characteristic frequencies of the signal $f_{mean}$ and $\Delta f$. These frequencies are between 10 Hz and 30 Hz, regardless of the contact quality, i.e., of the fact that the impact is hard or soft (see Figure 92 in Chapter 4 of *Farin* [2015]).

An explanation for the discrepancy between observed and theoretical elastic energy $W_{el}$ and for the fact that we did not observe any trend for the frequencies may be that we cannot record frequencies higher than 50 Hz because the sampling frequency is 100 Hz. Impacts of boulders are expected to generate waves of higher frequencies. For example, *Helmstetter and Garambois* [2010] dropped a boulder of similar dimensions on the Séchilienne rockslide site in the French Alps. Seismic signals generated by the impacts were sampled at 250 Hz by several seismic stations located a few tens of meters away. In the spectrogram of these signals, energy is visible up to 100 Hz. As we previously observed in laboratory experiments, when we do not measure the highest frequencies of the generated signal, the discrepancy between the theory and the measurement increases (e.g., for small masses $m$ in Figures 8a–8c). Another possibility is that the factor $\exp\left(\gamma(f)r\right)$, with $\gamma(f) = 2\pi f/(Qc_R)$, may be too simple to describe the wave propagation in such a damaged medium. Indeed, multiple modes with different dispersion relations can be excited in different frequency range in such layered media. However, the data are not sufficient to determine how wave disperse and attenuate within the ground on this specific site.

Owing to the large scattering of the seismic data, it is difficult to neither validate nor invalidate the applicability on the field of the analytical scaling laws developed in this paper. However, this study highlights several challenges that need to be addressed in order to be able to retrieve the impact parameters in future seismic studies of boulder impacts. If the radiated elastic energy or the characteristic frequencies of the emitted signals is underestimated, this will lead to either overestimate or underestimate the masses and impact speeds, as evidenced in our laboratory experiments (Figure 10). Therefore, one should measure as much as possible the entire energy spectrum emitted by the impacts and, to do so, use a high sampling frequency, ideally greater than $3/T_c$ (see section 2.2.2). For example, when the impactor's mass increases from $m = 10$ kg to $m = 2000$ kg in the rockfalls experiments investigated here, the impact duration $T_c$ increases from 0.01 s to 0.06 s, respectively, and the sampling frequency should be at least from 300 Hz to 50 Hz, respectively. Moreover, because energy at high frequencies attenuate very rapidly in fractured media, one should record the signal a close as possible from the impacts, up to typically a few tens of meters away. Finally, one should have a good knowledge of the elastic properties of the impactor and the ground in the vicinity of the impact, as well as within the ground, i.e., how it disperses and attenuates the frequencies. This could be achieved using several seismic stations recording at different distances from the source.

**6. Conclusions**

We developed analytical scaling laws relating the characteristics of the acoustic signal generated by an impact on a thin plate and on a thick block (radiated elastic energy and frequencies) to the parameters of the impact: the impactor's mass $m$ and speed before impact $V_z$ and the elastic parameters. These laws were validated with laboratory experiments of impacts of spherical beads of different materials and gravels on thin plates with a smooth surface, which is an ideal case, and on rough thick blocks, which are closer to the case of the field. Viscoelastic and elastoplastic dissipation occurred in the range of masses and impact speeds investigated. In these experiments, the radiated elastic energy is estimated from vibration measurements, independently of the other processes of energy dissipation. A number of conclusions can be drawn from our results:

1. The impactor mass $m$ and speed $V_z$ can be estimated from two independent parameters measurable on the field of the seismic signal: the radiated elastic energy and a characteristic frequency, using equations (27)–(30). The estimations of $m$ and $V_z$ are close to the real values within a factor of 2 and 3, respectively, even when the impactor has a complex shape. If the radiated elastic energy is underestimated (respectively, overestimated) by a factor of 10, the mass $m$ and the impact speed $V_z$ are underestimated

(respectively, overestimated) by a factor of 1.5 and 2, respectively. We noted that the radiated elastic energy is smaller when the surface roughness increases because the radius of contact is smaller. However, the signal characteristics measured during impacts of rough impactors on rough surfaces follows well the scaling laws established for impacts of spherical beads on a plane surface.

2. We also established a quantitative energy budget of the impacts on the plates and blocks investigated, and we estimated what should be this budget for natural rockfalls:

a. On the smooth plates, elastic waves, and viscoelastic dissipation are the main processes of energy losses. Viscoelastic dissipation is major for impactors of diameter less than 10% of the plate thickness, while elastic waves radiation represents only from 0.1% to 0.3% of the impact energy. When the bead diameter increases, the energy lost in viscoelastic dissipation decreases while the energy radiated in elastic waves increases. For beads of diameter larger than the plate thickness, almost all of the energy is radiated in elastic waves.

b. On the rough blocks, elastic dissipation represents only between 0.03% and 5% of the lost energy. In contrast, energy lost in other processes such as plastic deformation increases with the bead mass from 50% to more than 99% of the lost energy because of surface roughness. The energy dissipated in viscoelasticity decreases from 50% to 2% of the lost energy as the bead mass increases.

c. Most of the energy lost during a natural rockfall should be dissipated in plastic deformation or in translational or rotational modes of the impactors. Plastic or, in general, irreversible dissipation reduces the energy radiated in elastic waves and is difficult to quantify. That being said, regardless of the impactor's mass and speed, the energy radiated in elastic waves may not be more than a few percent of the impact energy. Energy lost in viscoelastic dissipation during the impact should be negligible in the range of masses detected by seismic stations on the field.

The impact experiments with rough impactors on rough substrates demonstrated that Hertz's model can be used to describe at first order the dynamics of an impact when the contact surface is not plane. Thus, we expect that the simple analytical relations derived in this paper between the characteristics of the impact and that of the emitted signal can allow us to better understand the process of elastic waves generation by impacts on the field. The major limitation for estimating the impact properties from the signal on the field would certainly be the fact that a great part of the radiated energy is lost in high frequencies during wave propagation in highly fractured media. Therefore, we encourage future seismic studies of rockfalls to record signals as close as possible to the impacts and to use a high-frequency sampling, at least 3 times the inverse impact duration. For example, for typical masses from 10 kg to 2000 kg detected by seismic stations on the field, we estimated that the impact duration varies from 0.01 s to 0.06 s, respectively, and that the sampling frequency should then be at minimum from 300 Hz to 50 Hz, respectively. In addition, it is important to correct measured seismic signals from wave dispersion and attenuation within the substrate. If these conditions are fulfilled, the scaling laws derived in this study should provide estimates of the order of magnitude of the masses and speeds of the impactors. Finally, in addition to direct field applications, the scaling laws developed for plates can be also useful in the industry as a nonintrusive technique to estimate the size and speed of particles in a granular transport and in shielding problems.

**Appendix A: Demonstration of the Analytical Scaling Laws for the Radiated Elastic Energy**

The objective of this appendix is to demonstrate the analytical scaling laws showed in Table 1 for the radiated elastic energy $W_{el}$ as a function of the impactor's mass $m$ and speed $V_z$ for thin plates and thick blocks.

The radiated elastic energy is defined by

$$W_{el} = \int_{-\infty}^{+\infty} |F_z(t)|^2 Y_{el}(t) dt = 2 \int_0^{+\infty} |\tilde{F}_z(f)|^2 \tilde{Y}_{el}(f) df, \tag{A1}$$

with $\tilde{Y}_{el}(f)$ the radiation admittance, which has a different expression on thin plates and on thick blocks.

**A1. Thin Plates**
On thin plates, $\tilde{Y}_{el}(f)$ is independent of frequency $f$ and is given by

$$Y_{el} = \frac{1}{8\sqrt{B\rho_p h}}. \tag{A2}$$

**Figure A1.** Coefficient $\beta$ defined by equation (A7) as a function of the Poisson ratio $v_p$.

where $B$ is the bending stiffness and $\rho_p$ and $h$ are the plate density and thickness, respectively.

Therefore,

$$W_{el} = \frac{1}{8\sqrt{B\rho_p h}} \frac{F_0^2 \delta_{z0}}{V_z} \int_{-\infty}^{+\infty} |g(t^*)|^2 dt^*, \quad (A3)$$

with $t^* = \delta_{z0}t/V_z$ and where $g(t^*)$ is the shape function represented on Figures 1b and 1c. The integral in this equation is noted $C_{plate}$ and depends on the inelastic parameters $\alpha$ and $P_Y/P_0$, i.e., of the fact that the impact is elastic, viscoelastic, or elastoplastic (Figures 2a and 2b). For an elastic impact, $C_{plate} = 1.21$.

Developing $F_0$ and $\delta_{z0}$ as functions of the impact parameters using their expressions in equations (5) and (8), respectively, we get

$$\frac{F_0^2 \delta_{z0}}{V_z} = \left(\frac{4}{3}\right)^{1/3} \left(\frac{5}{4}\right)^{8/5} \pi^{-1/15} \rho_s^{-1/15} E^{*2/5} m^{5/3} V_z^{11/5}. \quad (A4)$$

Finally, equations (A3) and (A4) give the scaling law relating the radiated elastic energy $W_{el}$ to the impact parameters on thin plates:

$$W_{el} = a_1 C_{plate} m^{5/3} V_z^{11/5}, \quad (A5)$$

with $a_1 \approx 0.18 E^{*2/5}/(\rho_s^{1/15}\sqrt{B\rho_p h})$.

**A2. Thick Blocks**

On thick blocks, the radiation admittance $\tilde{Y}_{el}(f)$ was computed in time Fourier domain by *Miller and Pursey* [1955]:

$$\tilde{Y}_{el}(f) = \frac{2\pi\xi^4\beta f^2}{\rho_p c_P^3}, \quad (A6)$$

where $\xi = \sqrt{2(1 - v_p)/(1 - 2v_p)}$, $c_P$ is the compressive wave speed and $\beta$ is the imaginary part of

$$\int_0^X \frac{x\sqrt{x^2 - 1}}{f_0(x)} dx, \quad (A7)$$

with $f_0(x) = (2x^2 - \xi^2)^2 - 4x^2\sqrt{(x^2 - 1)(x^2 - \xi^2)}$ and $X$, a real number greater than the positive real root of $f_0$. The coefficient $\beta$ depends only on the Poisson's ratio $v_p$ (Figure A1, see the Appendix of *Farin et al.* [2015], for details on the computation of $\beta$).

Therefore,

$$W_{el} = \frac{4\pi\xi^4\beta}{\rho_p c_P^3} \frac{F_0^2 V_z}{\delta_{z0}} \int_0^{+\infty} f^{*2}|\tilde{g}(f^*)|^2 df^*, \quad (A8)$$

with $f^* = V_z f/\delta_{z0}$ and $\tilde{g}(f^*)$ is the time Fourier transform of the function $g(t^*)$ represented on Figures 1b and 1c. We note $C_{block}$ the integral in this equation. $C_{block}$ depends on the inelastic parameters $\alpha$ and $P_Y/P_0$ (Figures 2a and 2b). With an impact force $F_z(t)$ given by *Hertz*'s [1882] elastic theory, i.e., for $\alpha = 0$ and $P_Y/P_0 = 1$, we have $C_{block} = 0.02$.

If we develop $F_0$ and $\delta_{z0}$ as functions of the impact parameters, we get

$$\frac{F_0^2 V_z}{\delta_{z0}} = \frac{4}{3}\left(\frac{5}{4}\right)^{4/5} \pi^{-1/5} \rho_s^{-1/5} E^{*6/5} m V_z^{13/5}. \quad (A9)$$

Finally, inserting equation (A9) into equation (A8) we obtain the analytical expression of the radiated elastic energy $W_{el}$ on thick blocks:

$$W_{el} = a_2 C_{block} m V_z^{13/5}, \quad (A10)$$

with the coefficient $a_2 \approx 15.93\xi^4\beta E^{*6/5}/(\rho_p\rho_s^{1/5}c_P^3)$.

[Figure]

**Figure B1.** Cumulated radiated elastic energy $W_{el}^{cumul}$ for the impact of steel beads of different diameters $d$ (different colors) on (a) the PMMA plate, (b) the glass plate, (c) the concrete block, and (d) the marble block, as a function of frequency $f$. Full line: experiments, dashed line: synthetics obtained with the convolution of Green's function with *Hertz*'s [1882] force of elastic impact.

**Appendix B: Cumulative Distribution of Energy**

In this appendix, we show how the radiated elastic energy radiated by impacts is distributed over the frequencies.

The cumulative distribution of the radiated elastic energy shows that impacts generate signals with higher frequencies as the bead diameter $d$ decreases, regardless of the structure (Figure B1). It is clear that the sensors used in our experiments do not measure energy for frequencies higher than 50 kHz. This is not a problem for impacts on the PMMA plate and for beads of diameter $d$ larger than 5 mm because all of the radiated elastic energy is in theory below 50 kHz (Figure B1a). However, for impacts of beads of 1 mm in diameter on glass, concrete, and marble, more than 50% of the energy is for frequencies higher than 50 kHz (Figures B1b to B1d). Some of the radiated energy may not be measured for the smallest beads investigated. Note that for experiments on the glass plate and on the concrete and marble blocks, the profile of the cumulative energy is steep and saturates to a given frequency $f \approx 38$ kHz, $f \approx 30$ kHz, and $f \approx 40$ kHz, respectively, as the bead diameter $d$ decreases (Figures B1b to B1d).

**Appendix C: Influence of the Impactor Shape on the Energy Budget**

In this appendix, we investigate the energy budget of impacts of gravels on the glass plate.

When a spherical bead is dropped without initial speed and rotation on a smooth surface it rebounds almost vertically and without spin. In contrast, a rough gravel rebounds to a much smaller height and can reach a large horizontal distance $x$ with a high-rotation speed $\omega_r$ up to about 400 rad s$^{-1}$, depending on the face it lands on (Figure C1a). For these complex impactors, the kinetic energy converted in translational and rotational modes is therefore not negligible. The translational kinetic energy of rebound is $E_c' = \frac{1}{2}mV'^2$, where $V' = V_x'\mathbf{u}_x + V_z'\mathbf{u}_z$ is the rebound speed in the cartesian frame $(0, \mathbf{u}_x, \mathbf{u}_z)$. $V_x' \approx 0$ cm s$^{-1}$ for spherical beads but varies from 5 cm s$^{-1}$ to 40 cm s$^{-1}$ for gravels. The rotation energy is $E_\omega = \frac{1}{2}I\omega_r^2$, where $I$ is the moment of inertia of the gravel, given by $I = \frac{2}{5}mR^2$ if we assume that the gravel is spherical with an equivalent radius $R$. From camera recordings, we estimate that 32% $\pm$ 17% of the impact energy $E_c$ is converted into translational energy of rebound $E_c'$ and that 13% $\pm$ 11% is converted into rotational energy $E_\omega$. Regardless of the shape and mass $m$ of the gravel, less energy is converted into translational energy $E_c'$ as its rotates faster after the impact (Figure C1b).

[Figure]

**Figure C1.** (a) Different rebound trajectories followed by the same gravel of mass $m = 0.23$ g dropped from height $H = 10$ cm several times on the glass plate (full lines) and one rebound trajectory followed a spherical bead of diameter $d = 4$ mm dropped from the same height $H$ (dashed line). Gravels of different masses $m$ (different symbols) are dropped without initial spin from height $H = 10$ cm on the glass plate. (b) Translational kinetic energy $E_c'$ of the gravels after rebound as a function of their rotation speed $\omega_r$ after rebound. (c–e) Percentage of impact energy lost in elastic waves $W_{el}/E_c$ as a function of the percentage of the impact energy $E_c$ converted (Figure C1c) in rebound translational energy $E_c'$, (Figure C1d) in rotational energy $E_\omega$, and (Figure C1e) in plastic deformation $W_{plast}$.

The percentage of energy radiated in elastic waves $W_{el}/E_c$ is 3.3% $\pm$ 1.8% and seems independent of the energy converted in translation energy $E_c'/E_c$ or in rotational modes $E_\omega/E_c$ (Figures C1c and C1d).

In section 4.4.1, we adjusted the inelastic parameter $D$ on the variation of the coefficient of restitution $e$ to estimate the energy lost in viscoelastic dissipation (Figure 12). This is not possible for gravels because of the large dispersion in the results. As granite has similar elastic properties than glass, we assume that $D$ is the same than for glass beads impacts on the glass plate, i.e., $D = 80$ ns (see Table 5). Therefore, the viscoelastic dissipation $W_{visc}$ for impacts of gravels on the glass plate may represent 3.7% $\pm$ 1% of $E_c$. The rest of the energy (48% $\pm$ 14%) is lost to deform plastically the gravel and or the glass plate. This is therefore the main process of energy dissipation for gravels impacts.

The proportion of energy radiated in elastic waves $W_{el}/E_c$ seems to decrease when more energy is lost in plastic deformation (Figure C1e), which is in agreement with the elastoplastic model (Figure 2a).

**Appendix D: Determining Impactor Momentum From Low Frequencies**

In some experiments on Figure 10, the estimations of $m$ and $V_z$ are affected because the highest frequencies of the generated vibration are not measured by the sensors or because of a resonance. The purpose of this appendix is to show that we can use the low-frequency content of the signal to estimate the momentum $mV_z$ of the impactor.

[Figure]

[Figure]

— synthetics
- - - synthetics approximation in 0
— measured signal
- - - power law fit

**Figure D1.** Measured amplitude spectrum $|\tilde{A}_z(r,f)|$ (black line) and synthetic spectrum (thick blue line) for the impact of a steel bead of diameter 5 mm on (a) the glass plate and (b) the concrete block. The blue dashed line is the power law approximation for low frequencies of the synthetic spectrum. The red dashed line is an adjustment of the low frequencies content of the measured spectrum with the power law.

For frequencies $f \sim 0$ Hz, we assume as *Tsai et al.* [2012] that the impact duration $T_c$ is instantaneous relative to the frequencies of interest. The time Fourier transform $\tilde{F}(f)$ of the *Hertz* [1882] force $F(t)$ then becomes constant in frequency:

$$\tilde{F}(f) = \int_{-\infty}^{+\infty} F(t)\exp(-ift)\mathrm{d}t \sim \int_{-\infty}^{+\infty} F(t)\mathrm{d}t, \quad (\mathrm{D1})$$

where, if we normalize the force $F(t)$ by its maximum value $F_0$ and time $t$ by the impact duration $T_c$ and develop their respective expressions (equations (9) and (10)),

$$\int_{-\infty}^{+\infty} F(t)\mathrm{d}t \approx 2mV_z. \quad (\mathrm{D2})$$

The amplitude spectrum of the vibration acceleration can then be approximated by [*Aki and Richards*, 1980]

$$|\tilde{A}_z(r, f \to 0)| \sim 2mV_z(2\pi f)^2|\tilde{G}_{zz}(r,f)|. \quad (\mathrm{D3})$$

Using the expression of Green's function $|\tilde{G}_{zz}(r,f)|$ given by equations (3) and (4) on plates and blocks, respectively, we show that

$$|\tilde{A}_z(r, f \to 0)| \sim af^b, \quad (\mathrm{D4})$$

with $a \approx 0.73 mV_z \frac{1}{B\sqrt{r}}(\frac{B}{\rho_p h})^{5/8}$ and $b = 3/4$ on plates and $a \approx 100 mV_z \frac{\xi^2}{\mu c_P} \frac{\sqrt{x_0(x_0^2-1)}}{f_0'(x_0)} \sqrt{\frac{2c_P}{\pi r}}$ and $b = 5/2$ on blocks.

In order to determine the momentum $mV_z$ of a steel bead of diameter 5 mm dropped from height 10 cm on the glass plate and on the concrete block, we adjust the power law (D4) with the measured spectra $|\tilde{A}_z(r,f)|$ for frequencies $f < 10$ kHz (Figure D1). The obtained momentum is $mV_z \approx 6.9.10^{-4}$ kg m s$^{-1}$ on glass plate and $mV_z \approx 6.33.10^{-4}$ kg m s$^{-1}$ on the concrete block, which is in good agreement with the real momentum $mV_z \approx 6.85.10^{-4}$ kg m s$^{-1}$. Finally, if either $m$ or $V_z$ is known, this method can be used to estimate the other parameters.

**Notation**

| | |
|---|---|
| $c_P, c_R$ | Compressional and Rayleigh waves speed (m s$^{-1}$) |
| $D$ | Viscoelastic coefficient (s) |
| $d, R$ | Bead diameter and radius (m) |
| $E_c$ | Initial kinetic energy (J) |
| $E_s, E_p, \nu_s, \nu_p$ | Young's modulii (Pa) and Poisson's coefficients of the sphere and the plane |
| $E^*$ | Equivalent Young's modulus (Pa) |
| $e$ | Coefficient of restitution ($-$) |
| $\mathbf{F}, F_z$ | Force and normal force (N) |
| $F_0, P_0$ | Maximum force and stress of elastic impact (N; Pa) |
| $F_{max}, \delta_{max}$ | Maximum force and penetration depth of inelastic impact (N) |
| $f, \omega$ | Frequency and angular frequency (s$^{-1}$) |
| $f_{peak}, f_{mean}, \Delta f$ | Peak, mean frequencies and bandwidth (Hz) |
| $g$ | Acceleration of gravitation (m s$^{-2}$) |
| $H$ | Height of fall (m) |
| $h$ | Plate thickness (m) |

| | |
|---|---|
| $K$ | Parameter in *Hertz*'s [1882] theory |
| $k$ | Wave number (m$^{-1}$) |
| $V$ | Volume (m$^3$) |
| $m$ | Mass (kg) |
| $r$ | Distance from the impact (m) |
| $T_c$ | Impact duration (s) |
| $t$ | Time (s) |
| $\mathbf{u}_i$ | Normalized vector of the direction $i$ |
| $v_i, a_i$ | Vibration speed and acceleration in the direction $\mathbf{u}_i$ (m s$^{-1}$; m s$^{-2}$) |
| $\tilde{V}_i$ | Time Fourier transform of $v_i$ and $a_i$, respectively (m; m s$^{-1}$) |
| $V_z, V'$ | Impact speed and speed after rebound (m s$^{-1}$) |
| $v_g, v_\phi$ | Group and phase velocities (m s$^{-1}$) |
| $W_{el}, \Delta E_c$ | Radiated elastic energy and total energy lost (J) |
| $W_{el}^{th}, W_{el}^{th'}$ | Theoretical radiated elastic energy predicted by *Hertz*'s [1882] and *Zener*'s [1941] models (J) |
| $W_{visc}, W_{other}, E_c', E_\omega$ | Energy lost in viscoelastic dissipation, in other processes, kinetic energy of rebound and rotation (J) |
| $x, y, z$ | Coordinates in the cylindric reference frame of the block (m) |
| $Y_d, P_d$ | Dynamic yield stress and dynamic yield strength (Pa) |
| $\alpha$ | Viscoelastic parameter (−) |
| $\beta$ | Coefficients involved in energy calculation (−) |
| $\gamma$ | Attenuation coefficient of energy with distance (m$^{-1}$) |
| $\delta_z$ | Penetration depth and maximum of this depth during the impact (m) |
| $\lambda_Z$ | *Zener*'s [1941] parameter (−) |
| $\rho_s, \rho_p$ | Densities of the sphere and the plane (kg m$^3$) |
| $\tau$ | Characteristic time of energy attenuation within the structure (s) |
| $\chi, \eta$ | Bulk and shear viscosities (Pa s) |
| $\omega_r$ | Rotation speed (rad s$^{-1}$) |

**Acknowledgments**

We thank E. Falcon, A. Valance, Y. Forterre, D. Royer, A. Schubnel, T. Reuschlé, and L. Jouniaux for their helpful discussions. We are indebted to A. Steyer for the technical help. We thank Aude Nachbaur, Hiromi Kobayashi, Christophe Rivière, Emmanuel Des Garets, and Emilie Nowak for their assistance in rockfall experiments in Tahiti. We are grateful to Florent Gimbert and an anonymous reviewer for their interesting comments to our initial manuscript. This work was supported by the ERC contract ERC-CG-2013-PE10-617472 SLIDEQUAKES and the Agence Nationale de la Recherche ANR-11-BS01-0016 LANDQUAKES, ANR REALISE, ITN FLOWTRANS and the USPC PAGES project. This is IPGP contribution 3684.

**References**

Aki, K., and P. Richards (1980), *Quantitative Seismology: Theory and Methods*, vol. 1, W.H. Freeman, San Francisco, Calif.

Allstadt, K. (2013), Extracting source characteristics and dynamics of the August 2010 Mount Meager landslide from broadband seismograms, *J. Geophys. Res. Earth Surf.*, *118*, 1472–1490, doi:10.1002/jgrf.20110.

Andreotti, B., Y. Forterre, and O. Pouliquen (2013), *Granular Media: Between Fluid and Solid*, vol. 1, Cambridge Univ. Press, Cambridge, U. K.

Brilliantov, N. V., F. Spahn, J.-M. Hertzsch, and T. Pöschel (1996), Model for collisions in granular gases, *Phys. Rev. E*, *53*, 5382–5392, doi:10.1103/PhysRevE.53.5382.

Burtin, A., L. Bollinger, J. Vergne, R. Cattin, and J. L. Nábělek (2008), Spectral analysis of seismic noise induced by rivers: A new tool to monitor spatiotemporal changes in stream hydrodynamics, *J. Geophys. Res.*, *113*, B05301, doi:10.1029/2007JB005034.

Buttle, D. J., and C. B. Scruby (1990), Characterization of particle impact by quantitative acoustic emission, *Wear*, *137*(1), 63–90, doi:10.1016/0043-1648(90)90018-6.

Buttle, D. J., S. R. Martin, and C. B. Scruby (1991), Particle sizing by quantitative acoustic emission, *Res. Nondestr. Eval.*, *3*(1), 1–26, doi:10.1007/BF01606508.

Crook, A. W. (1952), A study of some impacts between metal bodies by a piezo–lectric method, *Philos. Trans. R. Soc. A*, *212*(1110), 377–390, doi:10.1098/rspa.1952.0088.

Dammeier, F., J. R. Moore, F. Haslinger, and S. Loew (2011), Characterization of alpine rockslides using statistical analysis of seismic signals, *J. Geophys. Res.*, *116*, F04024, doi:10.1029/2011JF002037.

Davies, R. M. (1949), The determination of static and dynamic yield stresses using a steel ball, *Proc. R. Soc. A*, *197*(1050), 416–432, doi:10.1098/rspa.1949.0073.

Deparis, J., D. Jongmans, F. Cotton, L. Baillet, F. Thouvenot, and D. Hantz (2008), Analysis of rock-fall and rock-fall avalanche seismograms in the French Alps, *Bull. Seismol. Soc. Am.*, *98*(4), 1781–1796, doi:10.1785/0120070082.

Dewez, T. J. B., A. Nachbaur, C. Mathon, O. Sedan, H. Kobayashi, C. Riviere, F. Berger, E. Des Garets, and E. Nowak (2010), OFAI: 3D block tracking in a real-size rockfall experiment on a weathered volcanic rocks slope of Tahiti, French Polynesia. paper presented at Rock Slope Stability Conf. 2010, pp. 1–13, Paris, France, 24–25 Nov. 2010.

Duran, J. (2010), *Sands, Powders and Grains: An Introduction to the Physics of Granular Materials*, Academic Press, Boston.

Falcon, E., C. Laroche, S. Fauve, and C. Coste (1998), Behavior of one inelastic ball bouncing repeatedly off the ground, *Eur. Phys. J. B*, *3*(1), 45–57, doi:10.1007/s100510050283.

Farin, M. (2015), Études expérimentales de la dynamique et de l'émission sismique des instabilités gravitaires, PhD thesis, IPGP, Paris.

Farin, M., A. Mangeney, J. de Rosny, R. Toussaint, J. Sainte-Marie, and N. Shapiro (2015), Experimental validation of theoretical methods to estimate the energy radiated by elastic waves during an impact, *J. Sound Vib.*, doi:10.1016/j.jsv.2015.10.003.

Favreau, P., A. Mangeney, A. Lucas, G. Crosta, and F. Bouchut (2010), Numerical modeling of landquakes, *Geophys. Res. Let.*, *37*, L15305, doi:10.1029/2010GL043512.

🌀AGU **Journal of Geophysical Research: Solid Earth**          10.1002/2015JB012331

Ferrazzini, V., and K. Aki (1992), *Volcanic Seismology: Preliminary Results From a Field Experiment on Volcanic Events at Kilauea Using an Array of Digital Seismographs*, pp. 168–189, Springer, Berlin.

Geotechdata.info (2013), Soil Young's modulus. [Available at http://www.geotechdata.info/parameter/soil-young%27s-modulus.html], accessed: 2015-20-04.

Gimbert, F., V. C. Tsai, and M. P. Lamb (2014), A physical model for seismic noise generation by turbulent flow in rivers, *J. Geophys. Res. Earth Surf.*, *119*, 2209–2238, doi:10.1002/2014JF003201.

Goyder, H., and R. G. White (1980), Vibrational power flow from machines into built-up structures. Part I: Introduction and approximate analyses of beam and plate-like foundations, *J. Sound Vib.*, *68*(1), 59–75, doi:10.1016/0022-460X(80)90452-6.

Helmstetter, A., and S. Garambois (2010), Seismic monitoring of Séchilienne rockslide (French Alps): Analysis of seismic signals and their correlation with rainfalls, *J. Geophys. Res.*, *115*, F03016, doi:10.1029/2009JF001532.

Hertz, H. (1882), Über die Berührung fester elastischer Körper (On the vibration of solid elastic bodies), *J. Reine Angew. Math.*, *92*, 156–171, doi:10.1515/crll.1882.92.156.

Hertzsch, J., F. Spahn, and N. V. Brilliantov (1995), On low-velocity collisions of viscoelastic particles, *J. Phys. II*, *5*(11), 1725–1738, doi:10.1051/jp2:1995210.

Hibert, C., A. Mangeney, G. Grandjean, and N. M. Shapiro (2011), Slope instabilities in Dolomieu crater, Réunion Island: From seismic signals to rockfall characteristics, *J. Geophys. Res.*, *116*, F04032, doi:10.1029/2011JF002038.

Hibert, C., et al. (2014a), Automated identification, location, and volume estimation of rockfalls at piton de la fournaise volcano, *J. Geophys. Res. Earth Surf.*, *119*, 1082–1105, doi:10.1002/2013JF002970.

Hibert, C., G. Ekström, and C. Stark (2014b), Dynamics of the Bingham Canyon Mine landslides from seismic signal analysis, *Geophys. Res. Let.*, *41*, 4535–4541, doi:10.1002/2014GL060592.

Hunter, S. C. (1957), Energy absorbed by elastic waves during impact, *J. Mech. Phys. Solids*, *5*(3), 162–171, doi:10.1016/0022-5096(57)90002-9.

Israelachvili, J. (2002), *Intermolecular and Surface Forces*, 3rd ed., Springer, New York.

Johnson, K. (1985), *Contact Mechanics*, Cambridge Univ. Press, Cambridge, U. K.

Kanamori, H., and J. W. Given (1982), Analysis of long-period seismic waves excited by the May 18, 1980, eruption of Mount St. Helens—A terrestrial monopole, *J. Geophys. Res.*, *87*, 5422–5432, doi:10.1029/JB087iB07p05422.

Kuwabara, G., and K. Kono (1987), Restitution coefficient in a collision between two spheres, *Jpn. J. Appl. Phys.*, *26*(8), 1230–1233.

Lévy, C., A. Mangeney, F. Bonilla, C. Hibert, E. S. Calder, and P. J. Smith (2015), Friction weakening in granular flows deduced from seismic records at the Soufriére Hills Volcano, Montserrat, *J. Geophys. Res. Solid Earth*, doi:10.1002/2015JB012151.

Mayeda, K., and L. Malagnini (2010), Source radiation invariant property of local and near-regional shear-wave coda: Application to source scaling for the $M_w$ 5.9 Wells, Nevada sequence, *Geophys. Res. Lett.*, *37*(7), doi:10.1029/2009GL042148.

McLaskey, G. C., and S. D. Glaser (2010), Hertzian impact: Experimental study of the force pulse and resulting stress waves, *J. Acoust. Soc. Am.*, *128*(3), 1087, doi:10.1121/1.3466847.

Miller, G. F., and H. Pursey (1954), The field and radiation impedance of mechanical radiators on the free surface of a semi-infinite isotropic solid, *Proc. R. Soc. A.*, *223*(1155), 521–541, doi:10.1098/rspa.1954.0134.

Miller, G. F., and H. Pursey (1955), On the partition of energy between elastic waves in a semi-infinite solid, *Proc. R. Soc. A.*, *233*(1192), 55–69, doi:10.1098/rspa.1955.0245.

Moretti, L., A. Mangeney, Y. Capdeville, E. Stutzmann, C. Huggel, D. Schneider, and F. Bouchut (2012), Numerical modeling of the Mount Steller landslide flow history and of the generated long period seismic waves, *Geophys. Res. Lett.*, *39*, L16402, doi:10.1029/2012GL052511.

Moretti, L., K. Allstadt, A. Mangeney, Y. Capdeville, E. Stutzmann, and F. Bouchut (2015), Numerical modeling of the Mount Meager landslide constrained by its force history derived from seismic data, *J. Geophys. Res. Solid Earth*, *120*, 2579–2599, doi:10.1002/2014JB011426.

Norris, R. D. (1994), Seismicity of rockfalls and avalanches at three cascade range volcanoes: Implications for seismic detection of hazardous mass movements, *Bull. Seismol. Soc. Am.*, *84*(6), 1925–1939.

Ramírez, R., T. Pöschel, N. V. Brilliantov, and T. Schwager (1999), Coefficient of restitution of colliding viscoelastic spheres, *Phys. Rev. E*, *60*(4), 4465–4472, doi:10.1103/PhysRevE.60.4465.

Reed, J. (1985), Energy losses due to elastic wave propagation during an elastic impact, *J. Phys. D Appl. Phys.*, *18*(12), 2329–2337, doi:10.1088/0022-3727/18/12/004.

Royer, D., and E. Dieulesaint (2000), *Elastic Waves in Solids I: Free and Guided Propagation*, Springer, Berlin.

Sánchez-Sesma, F. J., R. L. Weaver, H. Kawase, S. Matsushima, F. Luzon, and M. Campillo (2011), Energy partitions among elastic waves for dynamic surface loads in a semi-infinite solid, *Bull. Seismol. Soc. Am.*, *101*(4), 1704–1709, doi:10.1785/0120100196.

Schmandt, B., R. C. Aster, D. Scherler, V. C. Tsai, and K. Karlstrom (2013), Multiple fluvial processes detected by riverside seismic and infrasound monitoring of a controlled flood in the Grand Canyon, *Geophys. Res. Lett.*, *40*, 4858–4863, doi:10.1002/grl.50953.

Suriñach, E., I. Vilajosana, G. Khazaradze, B. Biescas, G. Furdada, and J. M. Vilaplana (2005), Seismic detection and characterization of landslides and other mass movements, *Nat. Hazards Earth Syst. Sci.*, *5*(6), 791–798, doi:10.5194/nhess-5-791-2005.

The Engineering Toolbox (2014), Concrete properties. [Available at http://www.engineeringtoolbox.com/concrete-properties-d_1223.html], accessed: 11-20-14.

Tillett, J. (1954), A study of the impact of spheres on plates, *Proc. Phys. Soc. B*, *67*(9), 677–688, doi:10.1088/0370-1301/67/9/304.

Troccaz, P., R. Woodcock, and F. Laville (2000), Acoustic radiation due to the inelastic impact of a sphere on a rectangular plate, *J. Acoust. Soc. Am.*, *108*, 2197–2202, doi:10.1121/1.1312358.

Tsai, V. C., B. Minchew, M. P. Lamb, and J.-P. Ampuero (2012), A physical model for seismic noise generation from sediment transport in rivers, *Geophys. Res. Lett.*, *39*, L02404, doi:10.1029/2011GL050255.

Vilajosana, I., E. Suriñach, A. Abellan, G. Khazaradze, D. Garcia, and J. Llosa (2008), Rockfall induced seismic signals: Case study in Montserrat, Catalonia, *Nat. Hazards Earth Syst. Sci.*, *8*(4), 805–812, doi:10.5194/nhess-8-805-2008.

Vinningland, J. L., O. Johnsen, E. G. Flekkøy, R. Toussaint, and K. J. Måløy (2007a), Experiments and simulations of a gravitational granular flow instability, *Phys. Rev. E*, *76*, 051306, doi:10.1103/PhysRevE.76.051306.

Vinningland, J. L., O. Johnsen, E. G. Flekkøy, R. Toussaint, and K. J. Måløy (2007b), Granular Rayleigh-Taylor instability: Experiments and simulations, *Phys. Rev. Lett.*, *99*, 048001, doi:10.1103/PhysRevLett.99.048001.

Weaver, R. L. (1985), Diffuse elastic waves at a free surface, *J. Acoust. Soc. Am.*, *78*, 131–136, doi:10.1121/1.392576.

Yamada, M., Y. Matsushi, M. Chigira, and J. Mori (2012), Seismic recordings of landslides caused by Typhoon Talas (2011), Japan, *Geophys. Res. Lett.*, *39*, L13301, doi:10.1029/2012GL052174.

Zener, C. (1941), The intrinsic inelasticity of large plates, *Phys. Rev.*, *59*, 669–673, doi:10.1103/PhysRev.59.669.

---

## Author Comment (AC1) · 7 Jul 2018

I, on behalf of other co-authors, would like to express our gratitude for the reviewer's attitude towards the reviewing.

General comments: Most of the results given in the paper, in particular the variation of the coefficients of restitution as a function of the impact angle, were already reported in previous studies. It is not clear what this paper brings new to the research on energy losses during impacts. Please state clearly in the introduction what are the main questions that are posed at the end of the previous studies and needed additional experiments and answer to these specific questions in the conclusions. It is not clear

what people doing computer simulations of rockfalls should retain from this work and how they could use the presented results. I think that one important parameter that could allow us to better understand why kinetic energy losses are larger at high impact angles is the energy lost in rotational modes of the impactor. The more energy is dissipated in rotation after the impact, the less energy is restituted to the block as kinetic energy for rebound (cf Farin et al. (2015) Characterization of rockfalls from seismic signal: Insights from laboratory experiments, JGR:Earth Surface, Figure C1b). The authors could take advantage of the fact that their experimental setup has 8 cameras around the impact to measure precisely the rotation of the impactors before and after the impact and evaluate the rotational energy. This energy could be defined as 1/2* I * omega_rËĘ2, where I is the moment of inertia of the block (that could be approximated to a full sphere) and omega_r is its rotation speed. A figure showing the kinetic coefficient of restitution, Re, as a function of the rotational energy after impact could be interesting to show to bring additional contribution with respect to the previous work on the subject. Also, it is important to precise in the paper that the 'energy coefficient of restitution' is the 'kinetic energy coefficient of restitution', which does not represent the whole energy lost by the block but only the kinetic energy Ek lost. If a lot of energy is transmitted in rotation energy Er maybe the total energy of the block Ek + Er does not decreases at large impact angles (?). The authors suspect at several times in the paper that the impact speed has an influence on the coefficients of restitution. Thus, they should produce a Figure showing the coefficients of restitutions (and the rebound angle) as a function of the impact speed (event if only 3 different impacts speeds are investigated here, they could also use the data from previous work). Such a figure could support their discussion. I find that the discussion section is a bit difficult to follow. Maybe it could be reworked with subsections, discussing for example 'Interpretation of normal coefficient of restitution larger than 1', 'Relation between kinetic energy losses and normal coefficient of restitution': : :

Reply: Thanks you very much for your suggestion! The initial purpose of this study is to investigate whether the existing conclusions is valid when the test scale changes. To

date, restrained by the measure devices, the existing laboratory test are mainly small scale tests. For the model test on coefficients of restitution, the similarity theory is still absent because the influence factors are much more than the material properties and sizes. It is questionable whether the test scale influence the laws regarding the effect of the impact angle on the coefficients of restitution. So, bigger samples and a new measure technique are adopted to perform a medium-scale test, and the above question is expected to be answered by the result comparisons between our test and the existing small scale tests. Considering comments by all reviewers, the rotation is involved in the calculation of the energy coefficient of restitution RE, and the role of rotation in the effect of the impact angle on the coefficients of restitution is investigated. Because the magnitudes of the total kinetic energy before impact varies, the percentage of the total kinetic energy converted to rotational energy is used as a reference. Results show that the percentage increases as the impact angle decreases, and large samples are more likely to have a stead and small percentage than small samples. A higher percentage always induces a higher Rn and a lower Rt. While, no clear correlations occurs between the percentage and the other two coefficients, Rv and RE. In the revised manuscript, this has been listed as another contribution of this study. Thank you again for your suggestion! In this study, the small scale tests performed by Chau (2002), Cagnoli and Manga (2003), Asteriou (2012) are selected in the comparison. It is our pleasure that Cagnoli has also posted his comments. The magnitude difference in the coefficients of restitution between the tests compared attracted our attention, and we considered the impact velocity difference as the main reason. Actually, this deduction is arbitrary, considering that those tests differ from each other in multiple test conditions listed in Table 2. Cagnoli suggested that "The small Rn values in Cagnoli and Manga (2003) are due to the weak strength of pumice whose damage upon impact dissipates energy" in the comment. In Asteriou's latest paper (Asteriou, P. and Tsiambaos, G.: Effect of impact velocity, block mass and hardness on the coefficients of restitution for rockfall analysis, Int J Rock Mech Min Sci, 106, 41-50, 2018), a free fall test is performed using spherical balls vertically impacting the surface, and

results show that Rn reduces when increasing the impact velocity, and increases as the material become harder. Because multiple factors can affect the magnitude, it is unreasonable to appraise the effect of one specific factor on the magnitude of the coefficient of restitution using data from the tests under various conditions together. The mean value of Rn versus the impact velocity in this study are drawn with different slope angles in the new manuscript as Fig. 7, and no determined trend is observable. We cannot make a definitive conclusion which factor is the main reason for the magnitude difference in the coefficients of restitution between the tests compared. I am very sorry for the poor structure in the previous manuscript. Considering all comments, the structure of the paper and all figures are rearranged. I hope the new manuscript has an easy access to be scanned.

The main changes in manuscript: Considering all comments, the structure of this paper is rearranged. All figures are modified and rearranged. The purpose of this study is described as: (1) to verify whether the test scale influence the laws regarding the effect of the impact angle on the coefficients of restitution, (2) to determine the role of rotation in the effect of the impact angle on the coefficients of restitution. Rotation is involved in this study. As a consequence, the kinetic energy coefficient of restitution RE is recalculated, and results of the kinematic coefficient of restitution Rv is added in this study. The fitting curve are replaced by mean value lines of data points, and the fitting formula is removed. In the original manuscript, we considered the impact velocity difference as the main reason for the magnitude difference in the coefficients of restitution between the tests compared. In the revision, we withdraw this deduction. The role of rotation in the effect of the impact angle on the coefficients of restitution is investigated. As the percentage of the total kinetic energy converted to rotational energy increases, Rn increases but Rt decreases. The percentage increases as the impact angle decreases, and large samples are more likely to have a stead and small percentage than small samples.

To special comments Abstract: - l14: the impact angle 'with respect to the slope' page2,

[Figure]

L2: define the coefficient of restitution Reply: Thank you very much! In the revised abstract, the related sentence has been rephrase as your suggestion. Section 1 and 2 in the original manuscript are merged together and restructured as the introduction section in the new manuscript. The definition of the coefficient of restitution are given first, and then the previous study are illustrated.

Introduction: - l 26: 'the similitude requirements: : : cannot be easily matched': I do not understand this sentence. Please rewrite. Reply: When conducting a model test, the similarity ratio is usually important. While, a matured similarity theory is absent for those laboratory tests on the coefficients of restitution. The main reason is that the various factors are involved, such as the material properties, the shape of the rocks, the roughness, and the kinematic parameter. Thus, it is questionable whether the existing conclusions that the impact angle affects the coefficients of restitution based on small scale tests are valid when the test scale changes. In the new manuscript, the related sentence is rewritten. "Therefore, the existing results are restrained by the small scale of the laboratory tests. Influence factors are much more than the material properties and sizes, which induces the absence of the matured similarity theory for the model test on the coefficient of restitution (Heidenreich, 2004)."

page 2 L32: define the energy coefficient of restitution. 'The kinetic coefficient of restitution' is more appropriate. Reply: Thank you a lot! We have inspected the related literatures. Sometimes RE is called as the kinetic energy coefficient of restitution, and in some papers it is directly called as the energy coefficient of restitution. Of course, the first is more appropriate and it has been revised in the new manuscript.

page 2 l.34: Please do not give the same results as that given in the abstract. Please raise the general questions that require you to conduct additional experiments and that you answer in this paper, and answer these specific questions in the conclusion section. Sections 1.2 and section 2 should be merged with 1.Introduction and this whole section should lead to the problematic of the paper: what new contribution are you bringing to this research subject? To what questions are you answering? Reply:

Thank you for your suggestion! Section 1 and 2 in the original manuscript are merged together and restructured as the introduction section in the new manuscript. And the purpose of this study include: (1) to verify whether the test scale influence the laws that the impact angle affects the coefficients of restitution, (2) to determine the role of rotation in the effect of the impact angle on the coefficients of restitution. In the revised manuscript, the purpose has been stated in the ending of the introduction section.

- Page 3, L.15: ncor and tcor are never used in the following of the paper thus they should not be introduced. Reply: Thank you for your reminding! We have noticed the issue, and in the revised manuscript they are removed.

- Page 3, L.20: it could be also interesting to present the results for Rv as a function of the impact angle and the kinetic energy lost because lots of people are using this definition. Is it varying differently than Rn with the impact angle? Reply: In the original manuscript Rv wasn't presented because it is the square root of RE when the rotational energy isn't involved in RE. In the revised manuscript, the effect of the impact angle on Rv is also investigated, and the trend of Rv versus the impact angle is plotted as Fig. 5c.

- Page 4, l.2: ratio of kinetic energies Reply: Thank you for your reminding! In the revised manuscript it is revised.

- Page 4, l.26: 'the impact angle can influence the rebound angle': be more precise. Does the rebound angle increase or decrease when impact angle increases? Reply: Thank you a lot! In their paper, Cagnoli and Manga (2003) stated "The rebound angles are relatively larger at small and large impact angles with smaller values in between." In the original manuscript, we try to give a concise restatement while the meaning maybe unclear. In the new manuscript it has been revised as your suggestion.

Page 4, l.29: 'the kinematic coefficient of restitution Rv was more appropriate than the normal COR for use in correlations with the impact angles'. The relation between Rv and the impact angle should be also represented in this paper to check whether

this statement is also true with the present experiments. Reply: Thank you for your suggestion! In the revised manuscript, the effect of the impact angle on Rv is also investigated, and the trend of Rv versus the impact angle is plotted as Fig. 5c. In this study this statement is not valid. Various functions have considered to match data points, but no function can provide a correlation coefficient R2 more than 0.40 in terms of Rv for all options considered. Power function provides the best R2 in matching data points of Rn, which reaches 0.80.

- Page 5, l.3-4: These are poor sentences to sum up the previous results and motivate your work. Please clearly state at the end of the introduction what is missing from the previous work and requires you to do additional experiments. Reply: You are right! This is caused by the poor structure of the original manuscript. In the new manuscript, Section 1 and 2 in the original manuscript are merged together and restructured as the introduction section. And our motivation are illustrated.

- Page 5, l.14: what is the 'rebound hardness value'? Does not it have units? I think it could be more useful to give the Poisson's ratios and Young's moduli of the materials composing the impactors and the slabs. For example, people may want to use your data to compute impact forces (for example using Hertz's impact model) and compare the impact forces to the coefficients of restitution and impact angles and such computations require the Poisson's ratios and Young's moduli. Reply: Thank you for your suggestion! In the first place, the 'rebound hardness value' represent the hardness value measured by Schmidt hammer method, and it has no units. Some scholars considered the hardness as the key factor in the determination of the coefficient of restitution. In the revised manuscript, we provided the Poisson's ratios and Young's moduli for the material, and replaced "rebound hardness value" by "Schmidt Hardness R", which is a more formal name.

- Fig 6: The coefficient of restitution does not seem to depend on diameter, except 2 data points of higher value for D=10cm at low impact angle. In fact, the theory says that the coefficient of restitution should not depend on impactor size for impacts on a

[Figure]

thick block (when the thickness of the impacted slab is large compared to the size of the impactor) and that the COR decreases as the impactor size increases when the impact is on a substrate whose thickness is small compared to the impactor size (cf Farin et al. (2015) Characterization of rockfalls from seismic signal: Insights from laboratory experiments, JGR:Earth Surface). The slab you are using could be considered as thin compared to the impactor size but because the slabs seem to be a bit buried in ground, they may be considered as thick substrates, thus the coefficient of restitution does not depend on the impactor size. A comment on this could be interesting to explain the fact that the measured COR is independent of the impactor size in your experiments. Reply: Thank you for your suggestion! The law that the impact angle influence the coefficients of restitution appears independent of the sample sizes in this study. Your excellent study supports our results and we have list it as a reference. It is very interesting that in this study the sample size can affect the percentage of the total kinetic energy converted to rotational energy, but cannot affect the effect of the impact angle on the coefficients of restitution. We have checked the related literature till now, and we can't find similar works. Because more detailed information, such as the erosion caused by each impact and the impact orientation during collision, is not recorded when performing the test, the further research is absent. It is a pity and we would like to investigate this problem in the future.

- All Figures in general: Please use a larger and sans-serif font to improve figures readability. Reply: Thank you for your suggestion! In the new manuscript the figures are redrawn as your suggestion.

- Figures 6, 7 and 8: I would use the same kind of scaling law (power law) for the 3 coefficients of restitution to compare them. A 2nd order polynomial law for figure 8 makes no sense because (1) you could fit everything why it and (2) you change your mind after that and use a linear law in figure 5c because it compares better with the previous results. Reply: You are right! Considering comments by all reviewers, the best-fit curve is replaced by the mean value line for data points in the related figures

in the new manuscript. Considering the discreteness in data points, a general trend is more appropriate than a fitting cure to illustrate the effect of the impact angle on the coefficients of restitution. Although Wu (1985) suggested the linear correlation between the impact angle and Rn, Rt, a few literatures adopted the linear function to fit data points. In most literatures, data are not matched. In this study, test performed by by Chau et al. (2002), Cagnoli and Manga (2003), Asteriou et al. (2012) are selected to make a comparison. Cagnoli and Manga adopted a second-order polynomial to fit Rn, and adopted the linear function to fit Rt and RE. Asteriou adopted the power function to fit Rn and Rv. As your comment, we should pay more attention on the sence of the function adopted in fitting rather than their imitative effect. In the new manuscript, our efforts in matching data points are briefly described, and the related formula is removed. A conclusion which type of function should be recommended is not given, because the previous study and this study haven't provide sufficient evidence.

Please merge some of the figures together (e.g Fig. 2,3,4; Fig. 6,7,8; Fig. 12,13: : :) Reply: In the new manuscript the figures are merged as your suggestion and other comments.

Page 9, L.14: the sentence 'The values of Rt : : :' is unnecessary, one can read the values on the figure. Reply: In the new manuscript the sentence has been removed.

Page 10, L.5: the sentence 'The values of Re : : :' is unnecessary, one can read the values on the figure. Reply: In the new manuscript the sentence has been removed.

Page 10, L.16: Have you measured the depth of erosion created by the impacts? Maybe the largest impactor have caused more erosion of the slabs and thus lose more energy in deformation of the slab than the smallest impactors. A figure showing the energy lost as a function of the depth of erosion due to the impact could be interesting if you can do it. Reply: I am very sorry that more detailed information, such as the erosion caused by each impact and the impact orientation during collision, is not recorded when performing the test. We would like to investigate this problem in the subsequent

studies.

Page 12, L.3: 'The data points are stably located above the 45_ line until the impact angle reaches 36_' may be a clearer sentence. Reply: Thank you for your suggestion! In the new manuscript the sentence has been revised as your suggestion.

The '45_ line' is misleading because the compared variables are angles. The 'equality line' or 'y = x line' are other possibilities. Reply: Thank you for your reminding! In the new manuscript it is replaces by the "$\alpha=\beta$ line".

Page 12, l.5: 'the kinetic energy loss constituted 50-75% of the total kinetic energy' This is false: total energy also includes the rotation energy. Reply: You are right! Now the rotational energy is involved in the calculation of the kinetic energy coefficient of restitution. Therefore, the percentage is reduced. In the new manuscript, this mistake has been revised.

Page 12, l.6: 'the energy loss level cannot be assessed by comparing the rebound and impact angle': not clear Reply: Thank you for your suggestion! Results of our test shows that for a given impact angle, larger rebound angle doesn't means more kinetic energy dissipation than smaller rebound angle. The original sentence is not very clear. In the new manuscript, it has been revised as "Therefore, the ratio between the rebound angle and the impact angle cannot be directly used as a reference in estimating whether the energy loss level is high or low."

Page 12, l.18: Maybe you should directly compare your results with that of previous studies before drawing conclusions because your conclusions seem to change a bit after the comparison with the other studies (for example you say later than Rt does not depend on the impact angle and you change the scaling law for Re), thus sections 4.1, 4.2 and 5 are redundant and confusing. Reply: Thank you very much for your suggestion! Considering the purpose of this study, the paper is restructured. In the new manuscript, the results comparison between this study and the existing small scale tests follows the test results of this study, and they compose Section 3. The conclusion

is given after the comparison. "Various experimental conditions induce different results for Rn, Rt, Rv and RE, although there are certain trends that occur regardless of the test conditions. The normal coefficient of restitution Rn, kinematic coefficient of restitution Rv and kinetic energy coefficient of restitution RE all decrease with increasing in the impact angle, while the tangential coefficient of restitution Rt increases as the impact angle increases in most cases."

Page 12, L.23 to Page 13 L.2: This should be in the introduction. Reply: Thank you for your reminding! In the introduction, tests conducted by Chau et al. (2002), Cagnoli and Manga (2003), Asteriou et al. (2012) had been briefly introduced. Here, the detailed test conditions of those studies are provided in Table 2.

Page 13, L.9: what is the 'ideal state'? If you observe rebound angles larger than 1.2 times the impact angle, there is a chance that we can also observe this in nature. You should not exclude data points just because they do not compare well to the previous work. On contrary, you should keep these points and interpret why you observe such situation in your experiments and why it is not observed in the previous work. Reply: Thank you very much for your reminding! In the revised manuscript, all data points are reserved.

Table 2 and Fig. 11: please replace 'Wang 2018' by 'this study' to avoid confusion. Reply: The words in the figure have been revised as your suggestion.

Page 15, L.3: 'The minimum Rn occurred: : : erosion and particle breakage'. This explanation that stronger kinetic energy dissipation due to erosion may explain the lower Rn value for Cagnoli's experiment does not work because (1) you also observe erosion by the impacts and the Rn in your experiments are larger and (2) you state later that the normal coefficient of restitution does not correlate with kinetic energy loss: : : Reply: You are right! This study verifies that the test scales don't alter the general law regarding the effect of the impact angle on the coefficients of restitution. The reason that causes the magnitude difference is still questionable. Cagnoli suggested that "The

small Rn values in Cagnoli and Manga (2003) are due to the weak strength of pumice whose damage upon impact dissipates energy" in the comment. The existing studies and this study cannot provide sufficient evidence to determine the reason, because the test conditions are different in multiple aspects. Asteriou indicated that Rn reduces when increasing the impact velocity, and increases as the material become harder in the latest paper (Asteriou, P. and Tsiambaos, G.: Effect of impact velocity, block mass and hardness on the coefficients of restitution for rockfall analysis, Int J Rock Mech Min Sci, 106, 41-50, 2018). This problem is proposed in the ending of Section 3.

Page 15, L.9-12: the exact scaling law that describe best the data is not very important given the large scattering in the data. What matters more is if you can explain the general trend. Also, if you give a scaling law for you data, you should also try to fit the data of the previous work with the same kind of scaling law. If the scaling law works for your data and not with the other work, its usefulness is very limited: : Reply: Thank you very much for your reminding! Considering comments by all reviewers, the best-fit curve is replaced by the mean value line for data points of this study in the related figures in the new manuscript. In section 3.2, the trend line for the existing small scale tests are drawn as the original literature. The lines with data markers are the mean value lines, while those lines without data markers are fitting lines.

Page 15, L.15: The variation of the kinetic energy COR with impact angle may be better understood if you also show the rotation energy (more energy dissipated in rotation means less energy restituted in kinetic energy for the rebound). You should not remove data points just because they do not compare well with previous work. Explain the difference otherwise the same conclusions could have been drawn by just comparing the previous work together and this present work contribution is limited. Reply: Thank you very much for your suggestion! It is unreasonable to exclude those "non-ideal data points" for a better fitting curve. When the rotational energy is involved in this study, some interesting phenomenon is observed. When the impact angle is small, two sample sizes appear a clear distinction in the percentage of the total kinetic energy

converted to rotational energy. Small samples always induce bigger percentage than large samples. Considering that a higher percentage will results in a larger Rn and a lower Rt, the magnitude difference in the coefficients of restitution within the first impact angle interval is reasonable between two sample sizes.

Page 10, l.16: 'The impact velocity is an important: : : resulting coefficients of restitution': please show a figure of the CORs as a function of the impact speed (even including the previous work data) to support your conclusion. Reply: The magnitude difference in the coefficients of restitution between the tests compared was attributed to the difference in their impact velocity in the original manuscript. But, this deduction is arbitrary, considering the various test conditions. The mean value of Rn versus the impact velocity are drawn with different slope angles in the new manuscript as Fig. 7, and no determined trend is observable. The previous work data is not involved. In our opinion, to determine the effect of one specific factor on the magnitude of the coefficient of restitution using data from the tests under various test conditions together may be unreasonable. We cannot make a definitive conclusion which factor is the main reason for the magnitude difference in the coefficients of restitution between the tests compared.

Discussion section. Different things are discussed here, please add subsections to make the discussion clearer. Reply: Thank you very much for your suggestion! In the new manuscript, Discussion section is composed by three subsections.

Page 16, L. 24: I do not understand what you mean by 'with a parallel motion' Reply: I am sorry for the poor sentence. In the original paper, "with a parallel motion" means that only translational motion is involved. But we considers that this expression is also confusing. So, in the new manuscript, the related sentences are rewritten. "When the impact angle is sufficiently large to generate a rebound angle as the solid arrow, the border imposes no constraints on the rebound motion, and the sample can leave with the default rebound angle. But, when the impact angle is small and generate a default rebound angle as the dashed arrow, rotation motion must be involved to overcome the

constraint."

Page 16, L. 27: 'Therefore,ÂËŸa: : :' I do not understand the logical link with the previous sentence. If rotation speed has an important effect on rebound angle and coefficient of restitution, you should show it on Figures. Reply: Thank you very much for your suggestion! In the new manuscript, the direction transitions of translational velocities and the rotation are regarded as two consequence of the impact in Section 4. And the effect of the rotation on the coefficient of restitution is investigated. In the original paper, the logical link is not clear.

Page 17: I understand that basal roughness can lead to higher angles of rebound, but in this case, the impactors on intact slabs should have in average lower angles of rebound than impactors on eroded slabs. Can you draw a figure or give the average rebound angles on intact vs eroded slabs to support your discussion? If you measured the depth of erosion on the slabs, maybe the rebound angle could be correlated to with erosion depth (?). Reply: I am very sorry for the information isn't recorded. When one slab are too eroded, it is replaced by another one. For one specific slope angle, the data points from intact slabs and eroded slabs are mixed together, and we can't distinguish them now. And the depth of erosion is not measured for each impact. It is a pity. We would like to verify this phenomenon in the subsequent studies.

Page 20, l. 5: This conclusion does not bring anything new to the research. I believe you could draw much more results from you experimental data. Reply: Thank you very much for your encouragement! In the new manuscript, the contribution of this study is concluded as two points: (1) verified that several general laws occur when accounting for the effect of the impact angle, regardless of the test scales and conditions, (2) indicated that the rotation plays an important role in the effect of the impact angle on the coefficient of restitution. A higher percentage of kinetic energy converted to rotational energy always induces a higher normal coefficient of restitution $R_n$ and a lower tangential coefficient of restitution $R_t$.

[Figure]

Please also note the supplement to this comment:
https://www.nat-hazards-earth-syst-sci-discuss.net/nhess-2018-108/nhess-2018-108-AC1-supplement.pdf

**Supplement:**

[revised manuscript text omitted]

By introducing an angle coefficient

$$\lambda = \sin \beta / \sin \alpha \tag{6}$$

Eq. (6) can be simplified as

$$R_n = \lambda \sqrt{E_{rt} / E_i} \tag{7}$$

(a) $E_{rt}/E_i$ versus the impact angle

(b) $R_n$ versus the angle coefficient $\lambda$

**Fig. 13. The conditions of $R_n$ larger than 1.0**

$E_{rt}/E_i$, the ratio between the translational energy after impacting and the total kinetic energy before impacting, is plotted in Fig 13a with respect to the impact angle. From the perspective of the mean value, as the impact angle increases, when the

impact angles are less than 36º $E_{rt}/E$ increases, and if the impact angles beyond 36º, $E_{rt}/E_i$ decreases. While, the peak values of $E_{rt}/E$ of the four impact angle intervals fall down gradually as the impact angle increases. In this study, the values of $E_{rt}/E_i$ are located in the range (0.20, 0.60) except a few data points, which provides a standard for us to explore the conditions of $R_n$ larger than 1.0.

[revised manuscript text omitted]

When the impact angle is less than 30°, the rebound angle is more likely to exceed the impact angle, which can be attributed to the indentations and macro roughness caused by the impacts. The unexpected direction transition of translational velocity during the collision is always accompanied by the observable rotation. The percentage of kinetic energy converted to rotational energy increases as the impact angle decreases, and large samples are more likely to have a stead and small percentage than small samples. The percentage can be associated with the ratio between the rebound angle and the impact angle. For a given impact angle, larger rebound angles means more kinetic energy be converted to rotational energy during the collision. A higher percentage of kinetic energy converted to rotational energy always induces a higher normal coefficient of restitution $R_n$ and a lower tangential coefficient of restitution $R_t$, which means that the rotation motion play important role in the effect of the impact angle on the coefficient of restitution. But, no correlations are observable between the rotation motion and the other two coefficients, $R_v$ and $R_E$.

Whether the rebound angle greater than the impact angle happens is also influenced by the impact orientation, and the impact angle determines its probability. The probability increases as the impact angle decreases, which leads to higher peak values of $R_n$, as well as the extreme discreteness in the measured data points under small impact angle condition. This issue induces 
[revised manuscript text omitted]

---

## Author Comment (AC2) · 7 Jul 2018

I, on behalf of other co-authors, would like to express our gratitude for the reviewer's attitude towards the reviewing.

General comments: The article presents a laboratory study on the dependence of the coefficient of restitution regarding the impact angle, falling height etc. Based on the results a regression has been formulated to obtain normal and tangential coefficients of restitution. The R2 are not very high. This – in my opinion – has one main reason: the blocks are not spherical but have edges an corners. Their impact on the ground mains defines the rebound angle and velocity. The model itself cannot reflect this

effect because it neglects the rotational movement of the block that has a significant influence. Therefore, the model presented should be reported as being valid only for trajectory simulation codes based on point masses used to simulate the blocks. The model would not work for simulation codes that use folly shaped three-dimensional blocks. This should be stated in the introduction, handled in the discussion and be summarized in the conclusions.

Reply: Thanks you very much for your comments! In this study, the free fall tests are performed and the sample impacts the slope without rotation. When the sample leaves from the slope, the rotation is observable and the angular velocities are also recorded. Just as you indicated, the angular velocities are not involved in the previous manuscript because the assumption of a lumped-mass model is adopted in this study. Considering the comments by all reviewers, the rotation is involved in evaluating the effect of the impact angle on the coefficients of restitution. Results show that the percentage of kinetic energy converted to rotational energy increases as the impact angle decreases, and large samples are more likely to have a stead and small percentage than small samples. And a higher percentage of kinetic energy converted to rotational energy always induces a higher normal coefficient of restitution Rn and a lower tangential coefficient of restitution Rt. Although the impact orientations during impact are not involved in this study, the results may be useful for those codes based on a rigid body model when predicting the trajectory of spherical rocks with rough surface.

The main changes in manuscript: Considering all comments, the structure of this paper is rearranged. All figures are modified and rearranged. The purpose of this study is described as: (1) to verify whether the test scale influence the laws regarding the effect of the impact angle on the coefficients of restitution, (2) to determine the role of rotation in the effect of the impact angle on the coefficients of restitution. Rotation is involved in this study. As a consequence, the kinetic energy coefficient of restitution RE is recalculated, and results of the kinematic coefficient of restitution Rv is added in this study. The fitting curve are replaced by mean value lines of data points, and

the fitting formula is removed. In the original manuscript, we considered the impact velocity difference as the main reason for the magnitude difference in the coefficients of restitution between the tests compared. In the revision, we withdraw this deduction. The role of rotation in the effect of the impact angle on the coefficients of restitution is investigated. As the percentage of the total kinetic energy converted to rotational energy increases, Rn increases but Rt decreases. The percentage increases as the impact angle decreases, and large samples are more likely to have a stead and small percentage than small samples.

To special comments P1L13: Please, add short term on the kind of rock movements with or without rotation, "jumping" or vertically falling. Reply: As your suggestion, we have rephrased the sentences. "Free fall test are conducted and the velocities before and after the impact are obtained by a 3D motion capture system." The rotation is little before impact, but is observable after impact.

P2L4: Outdated references! Reply: Thanks you a lot! The simulation codes listed in the previous paper is too outdated and cannot reflect the new progress. We have inspected literatures till 2018 and some representative simulation codes are added in the revised manuscript.

P2L19: The COR is a model only. In reality it is almost zero. Example: take a spherical rock and let it fall –> it barely jumps. Reply: You are right! The coefficients of restitution is only useful for the bouncing phenomenon. When computer simulation codes are adopted in the trajectory predication, the coefficients of restitution should be input by users. Some typical values has been recommended for normal and tangential COR values according to the slope properties, such as clean hard rock, bedrock outcrops with boulders, and so on. Some summary work was listed in this paper, which may benefit some readers. As the structure of the paper is rearranged, the related sentence is removed to the end of Section 1.1, "So, Rn and Rt attracted most attentions in the previous studies, and some typical values of Rn and Rt had been summarized (Agliardi and Crosta, 2003; Heidenreich, 2004; Scioldo, 2006)."

[Figure]

P4L23, P9L14, P16L8: replace "increases in the impact" by "increasing" Reply: Thanks you for your suggestion! The expression has been revised as your suggestion.

P5L2: Glover also evaluated coefficients of restitution in http://etheses.dur.ac.uk/10968 Reply: Thanks you very much! James Glover did excellent works on how the shape of rock affect the rockfall dynamics. And we noticed that some results about the effect of the impact angle were also presented in the thesis, which has been added in the Section 1.2.

P5L12: use kg instead of g because it is doubtful that exact this weight is kept. Reply: Thanks you for your suggestion! The expression has been revised as your suggestion.

P7L5: 60fps might not be enough precise to capture the accelerations (during impact= time of the highest acceleration) there are only very small displacements that are not covered by the resolution of the cameras? Reply: Thanks you a lot! We inspected the data information and found that the accelerations are not provided by the system. In the revision the "accelerations" has been removed.

P12L8: Of course, if only translational movements are looked at. The hardness of the impact partners involved is not very relevant. The rebound is influenced mainly from the rock's edges and therefore related to its rotational movement. Reply: You are right! In the revised manuscript, the title of related figure is revised as "Direction transitions of translational velocities induced by impacts" and the related sentences is rephrased. The impact orientations, a corner contact or an edge contact, will affect the rebound motion. In the revised manuscript, the rotation is involved.

P13L7: This is a very precise weight.... Reply: Yes, it is a quite precise value. We inspected Chau's article "Coefficient of restitution and rotational motions of rockfall impacts, Int J Rock Mech Min Sci, 39, 69–77, 2002" again. And the weight is given clearly in the literature. So, we keep the value unchanged.

P16L30: "Assume" –> "Assuming" Reply: Thanks you for your suggestion! The expression has been revised as your suggestion.

P20L9: The presented concept of COR analysis an experimental/laboratory trajectory regarding the block's center of gravity. The shape of the block does not play any role as well as its rotational movement. The presented model to determine Rn therefore only works if the trajectory model simulates small mass points without rotational movements. As soon as the trajectory code aims to model spatially shaped blocks with edges and corners above data cannot be used. This consequence should be added to discussion and conclusions. Reply: In the previous paper we didn't point out that the assumption of a lumped-mass model in the study. Considering all comments, the rotation is involved in the revised paper, and its role in the effect of the impact angle is investigated. In this study the surface of samples are constituted by small artificial facets, the impact orientation can't be distinguished during the collision. Although the orientations of the block during impact are not involved in this study, the results maybe useful for those codes based on a rigid body model when simulating the trajectory of spherical rocks with rough surface.

References: Reply: Thank you very much for your reminding! We have inspected the details and formats of the references. Some new literatures are added according to the revision, and a unify format is used.

 

Please also note the supplement to this comment:
https://www.nat-hazards-earth-syst-sci-discuss.net/nhess-2018-108/nhess-2018-108-AC2-supplement.pdf

---

## Author Comment (AC3) · 7 Jul 2018

I, on behalf of other co-authors, would like to express our gratitude for the reviewer's attitude towards the reviewing.

General comments: This is a good set of experiments. I encourage the authors to take some time to improve their manuscript. Here some comments that can be useful. Reply: Thank you very much for your encouragement! To date, restrained by the measure devices, the existing laboratory test are mainly small scale tests. The initial purpose of this study is to investigate whether the existing conclusions regarding the effect of the impact angle on the coefficients of restitution are valid when the test scale

changes.Considering comments by all reviewers, the rotation is involved in this study. And, in the new manuscript, the role of the rotation in the effect of the impact angle is also investigated. Because the magnitudes of the total kinetic energy before impact varies, the percentage of the total kinetic energy converted to rotational energy is used as a reference. Results show that the percentage increases as the impact angle decreases, and large samples are more likely to have a stead and small percentage than small samples. A higher percentage always induces a higher Rn and a lower Rt. While, no clear correlations occurs between the percentage and the other two coefficients, Rv and RE. In the revised manuscript, this has been listed as another contribution of this study.

The main changes in manuscript: Considering all comments, the structure of this paper is rearranged. All figures are modified and rearranged. The purpose of this study is described as: (1) to verify whether the test scale influence the laws regarding the effect of the impact angle on the coefficients of restitution, (2) to determine the role of rotation in the effect of the impact angle on the coefficients of restitution. Rotation is involved in this study. As a consequence, the kinetic energy coefficient of restitution RE is recalculated, and results of the kinematic coefficient of restitution Rv is added in this study. The fitting curve are replaced by mean value lines of data points, and the fitting formula is removed. In the original manuscript, we considered the impact velocity difference as the main reason for the magnitude difference in the coefficients of restitution between the tests compared. In the revision, we withdraw this deduction. The role of rotation in the effect of the impact angle on the coefficients of restitution is investigated. As the percentage of the total kinetic energy converted to rotational energy increases, Rn increases but Rt decreases. The percentage increases as the impact angle decreases, and large samples are more likely to have a stead and small percentage than small samples.

To special comments LINE 4 PAGE 4. Please note that this RE value omits the rotational kinetic energy and as such, it simplifies the description of the collisions. I do

expect that your spherical polyhedrons rotated both before and after their impacts. This should be mentioned in the discussion since it affects the plot in Fig 8. For example, in our experiments (Cagnoli and Manga, 2003), our cylindrical particles did have a rotational kinetic energy but only after the collision with the target as the high-speed video camera confirmed. Reply: Thank you very much for your suggestions! Considering comments by all reviewers, the rotation is involved in the calculation of the energy coefficient of restitution RE in the new manuscript, and the role of rotation in the effect of the impact angle on the coefficients of restitution is investigated. In this study, the samples has little rotation before impact, while has the observable rotation when leaving the slope.

LINE 18 PAGE 5. I think that a drawing of the apparatus with vertical and horizontal length scales would improve the readability of the paper. Reply: Thank you very much for your suggestions! A general view of the apparatus has been added in the revised manuscript. All figures are rearranged according to their logical link, and some figures are merged.

FIG 8 PAG 11. Here, it seems to me that you felt the obligation to have to find one single best-fit curve even if your data points illustrate a much more complex situation. Rather than concave-down best-fit curves (which are truly not convincing), this plot shows two features: 1) the maximum values decrease as the impact angle increases and 2) the spread of the data points decreases as the impact angle increases. This is true for both your grain sizes. We obtained these same features as shown by Fig 4A in Cagnoli and Manga (2003). I strongly suggest to remove these concave-down curves because they are truly misleading. Reply: Thank you very much for your reminding! Considering comments by all reviewers, the best-fit curve is replaced by the mean value line for data points in the related figures in the new manuscript. Considering the discreteness in data points, a general trend is more appropriate than a fitting cure to illustrate the effect of the impact angle on the coefficients of restitution.

FIG 9 PAG 11. It would be useful to identify in this figure each experiment with its

own characteristics. Reply: Thank you very much for your reminding! In the new manuscript, two markers are adopted to represent data points for two sample sizes, respectively. And the original "45o line" is renamed as "$\alpha=\beta$ line" considering other comments.

TABLE 2 PAG 13. Please note that our cylinders are 0.89 cm long and with a basal diameter equal to 0.55 cm (Cagnoli and Manga, 2003). However, rebound angles of larger cylinders are also shown in Fig. 2A. Reply: Thank you a lot! In the new manuscript, the size of the cylinders are noted in Table 2.We noticed that the rebound angles of larger samples are presented in your excellent paper. It is not involve in the results comparison because the results comparison focuses on the effect of the impact angle on the coefficients of restitution. In Section 4.1 "Direction transitions of translational velocities", we noted that your test also observed this phenomenon.

LINE 9 PAG 13. The rebound angles can be larger than the impact angles for two reasons. First, the surface of your concrete slabs cannot be perfectly flat in particular after the target has been damaged by previous impacts. Second, the surface of your particles has a curvature that varies from place to place (i.e., they have edges and corners). In other words, the true impact angle is not known. In our Fig 2A, some rebound angles are also larger than the impact angles. Even if this seems to be a flaw of the experiments, it has to be accepted as the inevitable complexity of rock fragment collisions and it is still useful to understand this complexity. For this reason, it is not correct to exclude what you call "non-ideal data points" when computing best-fit curves. The truth is that a single best-fit curve of the entire set of data points in Fig 8 does not exist. You can plot only a trend line for the maximum values if you really want to. Reply: Thank you very much! In the revised manuscript, all data points are reserved. It is unreasonable to exclude those "non-ideal data points" for a better fitting curve. Considering comments by all reviewers, the mean value line for data points is adopted in the new manuscript.

FIG 11 PAG 14. Please, remove curves 5 from Figs 11a, 11b and 11c, because, in

nature, beta can be larger than alpha. Do Figs 11a and 11b display mean values? If yes, state this clearly. In Figure 11c, draw only curves showing the maximum RE values. Reply: Thank you for your reminding! All data points are reserved in the revised manuscript, so curve 5 is removed naturally. Considering comments by all reviewers, the mean value line is adopted to represent the trend for our study. The meaning of every trend line for the tests compared and our study in figures are stated in the new paper.

LINE 3 PAG 15. What you say here is true. However, I would rephrase the sentences. The small Rn values in Cagnoli and Manga (2003) are due to the weak strength of pumice whose damage upon impact dissipates energy. Reply: Thank you a lot! In the new manuscript, the sentence has been rewritten as your suggestion.

LINE 25 PAG 15. What do you mean with "nadir"? Please find a more appropriate word. Reply: I am very sorry for the inexact word used. Actually, in the original manuscript, we noticed that when the impact angle is less than 40o, your test provided the lowest Rt. And it increases as the impact angle increases. We considered the impact velocity as the main reason for the magnitude difference in the coefficients of restitution between different tests. However, in the new manuscript we didn't make a determined conclusion about this. Multiple factors can affect the magnitude, thus, it is unreasonable to appraise the effect of one specific factor on the magnitude of the coefficient of restitution using data from the tests under various conditions together.

LINE 30 PAG 15. As explained above, curve 1 in Fig 11c is not useful and should be removed from the plot. Reply: Thank you for your reminding! Considering comments by all reviewers, the mean value line is adopted to represent the trend for our study.

LINE 8 PAGE 16. This is not correct. Both your Fig 7 and our Fig 3B confirm that Rt increases as the impact angle increases. The problem is that the data spread is large. But this is due also to irregularity on the surfaces of target and particles, for example. Reply: You are right! One purpose of this study is to verify whether some general

laws occur when accounting for the effect of the impact angle, regardless of the test scales and conditions. The results comparison shows that the tangential coefficient of restitution Rt increases as the impact angle increases in most cases.

LINE 21 PAGE 16. This is the same explanation we have provided in our paper (see our Fig 1), but no credit is given. Reply: I am very sorry that more detailed information, such as the erosion depth caused by each impact and the impact orientation during each collision, is not recorded when performing the test. In the new manuscript, one figure of the damaged surface is provided as a credit in Discussion.

LINE 18 PAGE 18. The use of the coefficient of restitution does not provide a good description of rock fragment collisions. But credit should be given to who has already said it (e.g., Stronge, 1991). Both your and our data sets show that: 1) there is no such as thing as a single value of the coefficient of restitution, and 2) also the more informative ratio of the kinetic energy is not a constant. Reply: Thank you very much! In the revised manuscript, Stronge's conclusion has been cited as a credit, and the paper has been listed as a reference.

Please also note the supplement to this comment:
https://www.nat-hazards-earth-syst-sci-discuss.net/nhess-2018-108/nhess-2018-108-AC3-supplement.pdf

**Supplement:**

[revised manuscript text omitted]

By introducing an angle coefficient

$$\lambda = \sin \beta / \sin \alpha \tag{6}$$

Eq. (6) can be simplified as

$$R_n = \lambda \sqrt{E_{rt} / E_i} \tag{7}$$

(a) $E_{rt}/E_i$ versus the impact angle

(b) $R_n$ versus the angle coefficient $\lambda$

**Fig. 13. The conditions of $R_n$ larger than 1.0**

$E_{rt}/E_i$, the ratio between the translational energy after impacting and the total kinetic energy before impacting, is plotted in Fig 13a with respect to the impact angle. From the perspective of the mean value, as the impact angle increases, when the

impact angles are less than 36º $E_{rt}/E$ increases, and if the impact angles beyond 36º, $E_{rt}/E_i$ decreases. While, the peak values of $E_{rt}/E$ of the four impact angle intervals fall down gradually as the impact angle increases. In this study, the values of $E_{rt}/E_i$ are located in the range (0.20, 0.60) except a few data points, which provides a standard for us to explore the conditions of $R_n$ larger than 1.0.

[revised manuscript text omitted]

When the impact angle is less than 30°, the rebound angle is more likely to exceed the impact angle, which can be attributed to the indentations and macro roughness caused by the impacts. The unexpected direction transition of translational velocity during the collision is always accompanied by the observable rotation. The percentage of kinetic energy converted to rotational energy increases as the impact angle decreases, and large samples are more likely to have a stead and small percentage than small samples. The percentage can be associated with the ratio between the rebound angle and the impact angle. For a given impact angle, larger rebound angles means more kinetic energy be converted to rotational energy during the collision. A higher percentage of kinetic energy converted to rotational energy always induces a higher normal coefficient of restitution $R_n$ and a lower tangential coefficient of restitution $R_t$, which means that the rotation motion play important role in the effect of the impact angle on the coefficient of restitution. But, no correlations are observable between the rotation motion and the other two coefficients, $R_v$ and $R_E$.

Whether the rebound angle greater than the impact angle happens is also influenced by the impact orientation, and the impact angle determines its probability. The probability increases as the impact angle decreases, which leads to higher peak values of $R_n$, as well as the extreme discreteness in the measured data points under small impact angle condition. This issue induces 
[revised manuscript text omitted]

---

## Author Response (AR2)

Replies to the comments by Editor

Comments to the Author:

Dear authors,

The modifications provided to the revised manuscript have enhanced its quality. The main findings of the works are summarized and their limitations well addressed.

I consider the manuscript ready for publication if a few technical corrections are applied.

- Title: in the title, the term rockfall or free fall should be added in order to focus the object being studied

- Figures :

--> provide figures without the black / grey box around the graphs

--> do not insert the tile of the sub-figures (e.g. Fig 1, b, c...) within the figure in itself, but in the figure title. Just add (a), (b), (c), ... in the figure.

--> dimensions/sizes of the blocks and instruments on Fig. 2 and Fig. 3 should be added.

--> scale of Fig 4b should be added

--> dimension of the indentations on Fig. 12a should be added

- Figure title: most of the title are not adequate - provide a general title to each figure, and sub-title for each sub-figure

- Tables : format the tables according to the NHESS formatting ruleS.

Consider also to transfer Table 1 in an Annex.

- English : a very detailled review of the English language by a native speaker is needed. There are many grammatical errors throughout the text (too many to be summarized by myself) that need to be corrected. I give an example below (pg. 3, line 9-10):

"This definition originated from Newton's theory of particle collision, and had been used by Habib (1976), Paronuzzi (1989) and other scholars." should be rephrased in "This definition, originating from Newton's theory of particle collision, has been used by Habib (1976), Paronuzzi (1989) and other scholars."

and so on ...

Looking forward in receiving your revised version.

Best regards,

Jean-Philippe Malet

*Reply*: Thanks you very much!

The paper is revised according to your suggestion, and the main changes in manuscript are listed as follows.

(1) Title.

A new title "**Effects of the impact angle on the coefficient of restitution in rockfall analysis based on a medium-scale laboratory test**" is adopted.

(2) Figures.

All figure titles are revised to satisfy the requirement of NHESS, and the black / grey box around the graphs is removed. Some new dimensions and sizes are added on Fig. 2, Fig. 3 and Fig. 12a. Because Fig. 4b is a picture of the 3d interface in the motion analysis system, a 3d reference frame is added on Fig. 4b to illustrate the scale.

(3) Tables.

Two tables are revised to satisfy the requirement of NHESS. The meaning and unit of the notations in Table 1 are explained in the title of Table 1, which make Table 1 more concise. Therefore, Table 1 is still in the main text.

(4) English.

For proper English language, the manuscript is submitted to American Journal Experts which is a company providing language services for non-native English speakers. The editorial certificate is attached in the next page.

In the revised version, all changes are marked using red font.

Thank you again for your advices!

AMERICAN JOURNAL EXPERTS

**EDITORIAL CERTIFICATE**

This document certifies that the manuscript listed below was edited for proper English language, grammar, punctuation, spelling, and overall style by one or more of the highly qualified native English speaking editors at American Journal Experts.

**Manuscript title:**

Effects of the impact angle on the coefficient of restitution in rockfall analysis based on a medium-scale laboratory test

**Authors:**

Yanhai Wang, Wei Jiang, Shengguo Cheng, Pengcheng Song, Cong Mao

**Date Issued:**

October 9, 2018

**Certificate Verification Key:**

9CCC-F916-D8B4-1229-62DB

[Figure]

This certificate may be verified at www.aje.com/certificate. This document certifies that the manuscript listed above was edited for proper English language, grammar, punctuation, spelling, and overall style by one or more of the highly qualified native English speaking editors at American Journal Experts. Neither the research content nor the authors' intentions were altered in any way during the editing process. Documents receiving this certification should be English-ready for publication; however, the author has the ability to accept or reject our suggestions and changes. To verify the final AJE edited version, please visit our verification page. If you have any questions or concerns about this edited document, please contact American Journal Experts at support@aje.com.

American Journal Experts provides a range of editing, translation and manuscript services for researchers and publishers around the world. Our top-quality PhD editors are all native English speakers from America's top universities. Our editors come from nearly every research field and possess the highest qualifications to edit research manuscripts written by non-native English speakers. For more information about our company, services and partner discounts, please visit www.aje.com.

[revised manuscript text omitted]

---

## Author Response (AR3)

Editor Decision: Publish subject to minor revisions (review by editor) (19 Oct 2018) by Jean-Philippe Malet

Comments to the Author:

Dear authors,

Thanks for the revised version.

Technically, I do not see any red font in your file (v4) so I cannot pinpoint the changes made.

Please, provide the manuscript with the marked changes for further consideration.

Best regards

*Reply*: Thanks you very much!

The manuscript with the marked changes is attached in the Author's Response unloaded in October 10, 2018. I am very sorry that I might misunderstand some instructions in the editorial email. I have submitted the documents again.

Thank you again for your patience!

---

## Author Response (AR4)

Replies to the comments by Editor

Comments to the Author:

Dear authors,

Thanks for the revised version with the tracked changes. The manuscript is nearly ready for publication, subject to two minor changes:

I would suggest the authors to add dimensions on the Figures 3c and 3d (as already done for Figure 2)

The authors should use standard referencing for the units; for instance m.s-1 should be used and not m/s. Please verify for all units through the manuscript.

Best regards,

Jean-Philippe Malet

*Reply*: Thanks you very much for your suggestion!

The paper is revised and the main changes in manuscript are listed as follows.

(1) Dimensions are added on the Figures 3c and 3d as your suggestion.

(2) We checked the units in the manuscript, and all units using non-standard notation were modified.

In the revised version, all changes are marked using red font.

Thank you again for your advices!